# Diffusion Forcing: Next-token Prediction Meets Full-Sequence Diffusion

**Boyuan Chen**
MIT CSAIL
boyuanc@mit.edu

**Diego Martí Monsó**[*]
Technical University of Munich
diego.marti@tum.de

**Yilun Du**
MIT CSAIL
yilundu@mit.edu

**Max Simchowitz**
MIT CSAIL
msimchow@mit.edu

**Russ Tedrake**
MIT CSAIL
russt@mit.edu

**Vincent Sitzmann**
MIT CSAIL
sitzmann@mit.edu

## Abstract

This paper presents Diffusion Forcing, a new training paradigm where a diffusion model is trained to denoise a set of tokens with *independent* per-token noise levels. We apply Diffusion Forcing to sequence generative modeling by training a causal next-token prediction model to generate one or several future tokens without fully diffusing past ones. Our approach is shown to combine the strengths of next-token prediction models, such as variable-length generation, with the strengths of full-sequence diffusion models, such as the ability to guide sampling to desirable trajectories. Our method offers a range of additional capabilities, such as (1) rolling-out sequences of continuous tokens, such as video, with lengths past the training horizon, where baselines diverge and (2) new sampling and guiding schemes that uniquely profit from Diffusion Forcing's variable-horizon and causal architecture, and which lead to marked performance gains in decision-making and planning tasks. In addition to its empirical success, our method is proven to optimize a variational lower bound on the likelihoods of all subsequences of tokens drawn from the true joint distribution. Project website: `https://boyuan.space/diffusion-forcing/`

## 1  Introduction

Probabilistic sequence modeling plays a crucial role in diverse machine learning applications including natural language processing [6, 47], video prediction [31, 69] and decision making [3, 22]. Next-token prediction models in particular have a number of desirable properties. They enable the generation of sequences with varying length [32, 21, 37] (generating only a single token or an "infinite" number of tokens via auto-regressive sampling), can be conditioned on varying amounts of history [21, 37], support efficient tree search[70, 23, 25], and can be used for online feedback control [22, 3].

Current next-token prediction models are trained via *teacher forcing* [64], where the model predicts the immediate next token based on a ground truth history of previous tokens. This results in two limitations: (1) there is no mechanism by which one can guide the sampling of a sequence to minimize a certain objective, and (2) current next-token models easily become *unstable* on continuous data. For example, when attempting to auto-regressively generate a video (as opposed to text [6] or vector-quantized latents [33]) past the training horizon, slight errors in frame-to-frame predictions accumulate and the model diverges.

---

[*]Work done as a visiting student at MIT.

38th Conference on Neural Information Processing Systems (NeurIPS 2024).

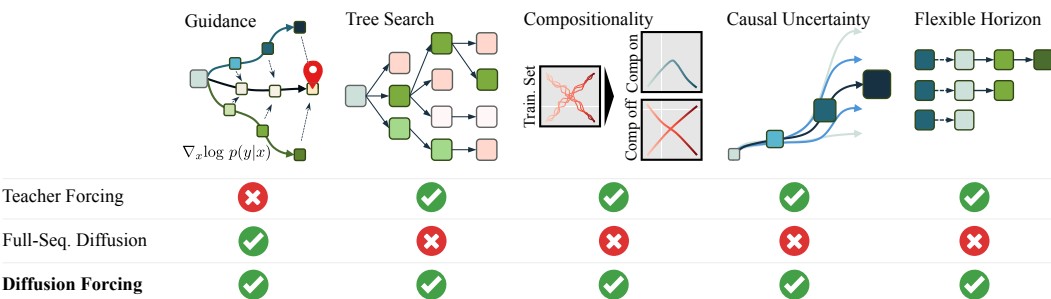

| | Guidance | Tree Search | Compositionality | Causal Uncertainty | Flexible Horizon |
|---|---|---|---|---|---|
| Teacher Forcing | ❌ | ✅ | ✅ | ✅ | ✅ |
| Full-Seq. Diffusion | ✅ | ❌ | ❌ | ❌ | ❌ |
| **Diffusion Forcing** | ✅ | ✅ | ✅ | ✅ | ✅ |

Figure 1: **Diffusion Forcing capabilities.** Today, different applications such as language modeling [6], planning [36], or video generation [31, 69] rely on *either* auto-regressive next-token prediction *or* full-sequence diffusion, according to their respective unique capabilities. The proposed Diffusion Forcing is a novel sequence generative model that enjoys key strengths of both model types.

*Full-sequence diffusion* seemingly offers a solution. Commonly used in video generation and long-horizon planning, one directly models the joint distribution of a fixed number of tokens by diffusing their concatenation [31, 1], where the noise level is identical across all tokens. They offer *diffusion guidance* [30, 16] to guide sampling to a desirable sequence, invaluable in decision-making (planning) applications [36, 34]. They further excel at generating continuous signals such as video [31]. However, full-sequence diffusion is universally parameterized via non-causal, unmasked architectures. In addition to restricting sampling to full sequences, as opposed to variable length generation, we show that this limits the possibilities for both guidance and subsequence generation (Figure 1). Further, we demonstrate that a naive attempt at combining the best of both worlds by training a next-token prediction model for full-sequence diffusion leads to poor generations, intuitively because it does not model the fact that small uncertainty in an early token necessitates high uncertainty in a later one.

In this paper, we introduce *Diffusion Forcing* (DF), a training and sampling paradigm where each token is associated with a *random, independent* noise level, and where tokens can be denoised according to arbitrary, independent, per-token schedules through a shared next-or-next-few-token prediction model. Our approach is motivated by the observation that noising tokens is a form of *partial masking*—zero noise means a token is unmasked, and complete noise fully masks out a token. Thus, DF forces the model to learn to "unmask" any collection of variably noised tokens (Figure 2). Simultaneously, by parameterizing predictions as a composition of next-token prediction models, our system can flexibly generate varying length sequences as well as compositionally generalize to new trajectories (Figure 1).

We implement DF for sequence generation as *Causal Diffusion Forcing* (CDF), in which future tokens depend on past ones via a causal architecture. We train the model to denoise all tokens of a sequence at once, with an independent noise level per token. During sampling, CDF gradually denoises a sequence of Gaussian noise frames into clean samples where different frames may have different noise levels at each denoising step. Like next-token prediction models, CDF can generate variable-length sequences; unlike next-token prediction, it does so stably from the immediate next token to thousands of tokens in the future – even for continuous tokens. Moreover, like full-sequence diffusion it accepts guidance towards high-reward generations. Synergistically leveraging causality, flexible horizon, and variable noise schedules, CDF enables a new capability, Monte Carlo Guidance (MCG), that dramatically improves the sampling of high-reward generations compared to non-causal full-sequence diffusion models. Fig. 1 overviews these capabilities.

In summary, our contributions are: (1) We propose Diffusion Forcing, a new probabilistic sequence model that has the flexibility of next-token prediction models while being able to perform long-horizon guidance like full-sequence diffusion models. (2) Taking advantage of Diffusion Forcing's unique capabilities, we introduce a novel decision-making framework that allows us to use Diffusion Forcing as simultaneously a *policy* ([10]) and as a *planner* ([36]). (3) We formally prove that, under appropriate conditions, optimizing our proposed training objective maximizes a lower bound on the likelihood of the joint distribution of *all sub-sequences* observed at training time. (4) We empirically evaluate CDF across diverse domains such as video generation, model-based planning, visual imitation learning, and time series prediction, and demonstrate CDF's unique capabilities,

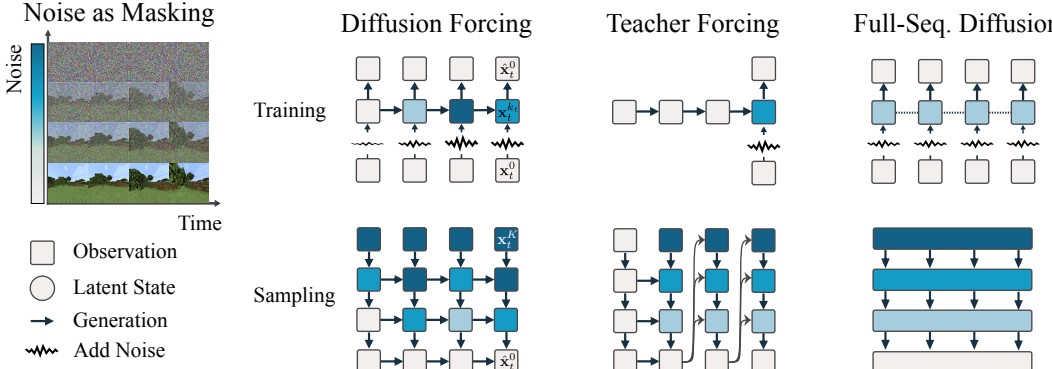

Figure 2: **Method Overview.** Diffusion Forcing trains causal sequence neural networks (such as an RNN or a masked transformer) to denoise flexible-length sequences where each frame of the sequence can have a *different* noise level. In contrast, next-token prediction models, common in language modeling, are trained to predict a single next token from a *ground-truth* sequence (teacher forcing [64]), and full-sequence diffusion, common in video generation, train non-causal architectures to denoise all frames in a sequence at once with the *same* noise level. Diffusion Forcing thus *interleaves* the time axis of the sequence and the noise axis of diffusion, unifying strengths of both alternatives and enabling completely new capabilities (see Secs. 3.2,3.4).

such as stabilizing long-rollout autoregressive video generation, composing sub-sequences of those observed at training time with user-determined memory horizon, Monte Carlo Guidance, and more.

## 2 Related Work and Preliminaries

We discuss related work and preliminaries for our core application, sequence generative modeling; see Appendix D for further literature review.

Our method unifies two perspectives on sequence modeling: Bayesian filtering along the time axis, denoted by subscript $t$, and diffusion along an "uncertainty" (or noise level) axis denoted by superscript $k$. In the following, we denote observations as $\mathbf{x} \in \mathcal{X}$ and latent states as $\mathbf{z} \in \mathcal{Z}$.

**Bayesian Filtering.** Given a Hidden Markov Model (HMM) defined by latent states $\mathbf{z}_t$ and observations $\mathbf{x}_t$, a Bayes filter is a probabilistic method for estimating latent states recursively over time from incoming observations. A prior model $p(\mathbf{z}_{t+1}|\mathbf{z}_t)$ infers a belief over the next state given only the current state, and an observation model infers a belief over the next observation given the current latent state $p(\mathbf{x}_t|\mathbf{z}_t)$. When a new observation is made, a posterior model $p(\mathbf{z}_{t+1}|\mathbf{z}_t, \mathbf{x}_{t+1})$ provides an updated estimation of the next latent state $\mathbf{z}_{t+1}$. When trained end-to-end with neural networks [22, 23], latent states are not an estimate of any physical quantity, but a sufficiently expressive latent that summarizes past observations for predicting future observations $(\mathbf{x}_{t'})_{t'>t}$ in the sequence.

**Diffusion Models.** Diffusion models [56, 28] have proven to be highly expressive and reliable generative models. We review their essentials here. Let $q(\mathbf{x})$ denote a data distribution of interest, and let $\mathbf{x}^0 \equiv \mathbf{x} \sim q$. We consider a forward diffusion process that gradually adds Gaussian noise to a data point over a series of time steps. This process is modeled as a Markov chain, where the data at each step $k$ is noised incrementally:

$$q(\mathbf{x}^k|\mathbf{x}^{k-1}) = \mathcal{N}(\mathbf{x}^k; \sqrt{1-\beta_k}\mathbf{x}^{k-1}, \beta_k\mathbf{I}) \tag{2.1}$$

where $\mathcal{N}$ is the normal distribution and $\beta_k$ is the variance of the noise added at each step controlled by a schedule $\{\beta_k \in (0,1)\}_{k=1}^K$. The process continues until the data is converted into pure noise at $\mathbf{x}^K$. The reverse process is also a Markov chain and attempts to recreate the original data from the noise with a parameterized model $p_\theta$:

$$p_\theta(\mathbf{x}^{k-1}|\mathbf{x}^k) = \mathcal{N}(\mathbf{x}^{k-1}; \boldsymbol{\mu}(\mathbf{x}^k, k), \gamma_k\mathbf{I}), \tag{2.2}$$

where the mean $\boldsymbol{\mu}$ is a model with a neural network, and where it is shown [29] that one can set the covariance to the identity scaled by a fixed constant $\gamma_k$ depending on $k$. Adopting the standard

exposition, we reparametrize the mean $\boldsymbol{\mu}$ in terms of noise prediction $\boldsymbol{\epsilon} = (\sqrt{1 - \bar{\alpha}_t})^{-1}\mathbf{x}_t^{k_t} - \sqrt{\bar{\alpha}_t}\boldsymbol{\mu}$. This leads [28] to the following least squares objective:

$$\mathcal{L}(\theta) = \mathbb{E}_{k,\mathbf{x}^0,\epsilon}\left[\|\boldsymbol{\epsilon}^k - \boldsymbol{\epsilon}_\theta(\mathbf{x}^k, k)\|^2\right], \tag{2.3}$$

where $\mathbf{x}^k = \sqrt{\bar{\alpha}_k}\mathbf{x}^0 + \sqrt{1 - \bar{\alpha}_k}\boldsymbol{\epsilon}^k$ and $\boldsymbol{\epsilon}^k \sim \mathcal{N}(0, \mathbf{I})$ . One can then sample from this model via Langevin dynamics $\mathbf{x}^{k-1} \leftarrow \frac{1}{\sqrt{\alpha_k}}(\mathbf{x}_t^k - \frac{1-\alpha_k}{\sqrt{1-\bar{\alpha}_k}}\boldsymbol{\epsilon}_\theta(\mathbf{x}_t^k, k) + \sigma_k\mathbf{w})$ [28].

**Guidance of Diffusion Models.** Guidance [30, 16] allows biasing diffusion generation towards desirable predictions at sampling time. We focus on classifier guidance [16]: given a classifier $c(y|\mathbf{x}^k)$ of some desired $y$ (e.g. class or success indicator), one modifies the Langevin sampling [29] gradient $\boldsymbol{\epsilon}_\theta(\mathbf{x}^k, k)$ to be $\boldsymbol{\epsilon}_\theta(\mathbf{x}^k, k) - \sqrt{1 - \bar{\alpha}_k}\nabla_{x^k}\log c(y|\mathbf{x}^k)$. This allows sampling from the joint distribution of $\mathbf{x}$ and class label $y$ without the need to train a conditional model. Other energies such as a least-squares objective comparing the model output to a desirable ground truth have been explored in applications such as decision making [16, 36].

**Next-Token Prediction Models.** Next-token prediction models are sequence models that predict the next frame $\mathbf{x}_{t+1}$ given past frames $\mathbf{x}_{1:t}$. At training time, one feeds a neural network with $\mathbf{x}_{1:t}$ and minimizes $||\hat{\mathbf{x}} - \mathbf{x}||^2$ for continuous data or a cross-entropy loss for discrete data [64]. At sampling time, one samples the next frame $\hat{\mathbf{x}}_{t+1}$ following $p(\mathbf{x}_{t+1}|\mathbf{x}_{1:t})$. If one treats $\hat{\mathbf{x}}_{t+1}$ as $\mathbf{x}_{t+1}$, one can use the same model to predict $\mathbf{x}_{t+2}$ and repeat until a full sequence is sampled. Unlike full-sequence diffusion models, next-token models do not accept multi-step guidance, as prior frames must be fully determined to sample future frames.

**Diffusion Sequence Models.** Diffusion has been widely used in sequence modeling. [43] use full-sequence diffusion models to achieve controllable text generation via guidance, such as generating text following specified parts of speech. [31] trains full-sequence diffusion models to synthesize short videos and uses a sliding window to roll out longer conditioned on previously generated frames. [36] uses full-sequence diffusion models as planners in offline reinforcement learning. This is achieved by training on a dataset of interaction trajectories with the environment and using classifier guidance at sampling time to sample trajectories with high rewards towards a chosen goal. [49] modifies auto-regressive models to denoise the next token conditioned on previous tokens. It trains with teacher forcing [64] and samples next-token auto-regressively for time series data. Most similar to our work is AR-Diffusion [65], which trains full-sequence text diffusion with a causal architecture with linearly dependent noise level along the time axis. We provide a detailed comparision between this approach and ours in Appendix D.

## 3 Method

### 3.1 Noising as partial masking

Recall that *masking* is the practice of occluding a subset of data, such as patches of an image [26] or timesteps in a sequence [15, 48], and training a model to recover unmasked portions. Without loss of generality, we can view any collection of tokens, sequential or not, as an ordered set indexed by $t$. Training next-token prediction with teacher forcing can then be interpreted as masking each token $\mathbf{x}_t$ at time $t$ and making predictions from the past $\mathbf{x}_{1:t-1}$. Restricted to sequences, we refer to all these practices as *masking along the time axis*. We can also view full-sequence forward diffusion, i.e., gradually adding noise to the data $\mathbf{x}_{1:T}^0 \equiv \mathbf{x}_{1:T}$, as a form of *partial masking*, which we refer to as *masking along the noise axis*. Indeed, after $K$ steps of noising, $\mathbf{x}_{1:T}^K$ is (approximately) pure white noise without information about the original data.

We establish a unified view along both axes of masking (see Fig. 2). We denote $\mathbf{x}_{1:T}$ for a sequence of tokens, where the subscript indicates the time axis. As above, $\mathbf{x}_t^{k_t}$ denotes $\mathbf{x}_t$ at noise level $k_t$ under the forward diffusion process (2.1); $\mathbf{x}_t^0 = \mathbf{x}$ is the unnoised token, and $\mathbf{x}_t^K$ is white noise $\mathcal{N}(0, \mathbf{I})$. Thus, $(\mathbf{x}_t^{k_t})_{1 \leq t \leq T}$ denotes a sequence of noisy observations where each token has a *different* noise level $k_t$, which can be seen as the degree of *partial masking* applied to each token through noising.

### 3.2 Diffusion Forcing: different noise levels for different tokens

*Diffusion Forcing* (DF) is a framework for training and sampling arbitrary sequence lengths of noisy tokens $(\mathbf{x}_t^{k_t})_{1 \leq t \leq T}$, where critically, *the noise level $k_t$ of each token can vary by time step*. In this

| **Algorithm 1** Diffusion Forcing Training | **Algorithm 2** DF Sampling with Guidance |
|---|---|
| 1: **loop** | 1: **Input:** Model $\theta$, scheduling matrix $\mathcal{K}$, initial latent $\mathbf{z}_0$, guidance cost $c(\cdot)$. |
| 2:    Sample tajectory of observations $(\mathbf{x}_1, ..., \mathbf{x}_T)$. | 2: **Initialize** $\mathbf{x}_1, \ldots, \mathbf{x}_T \sim \mathcal{N}(0, \sigma_K^2 I)$. |
| 3:    **for** $t = 1, ..., T$ **do** | 3: **for** row $m = M - 1, ..., 0$ **do** |
| 4:        Sample independent noise level $k_t \in \{0, 1, ..., K\}$ | 4:    **for** $t = 1, \ldots, T$ **do** |
| 5:        $\mathbf{x}_t^{k_t} = \text{ForwardDiffuse}(\mathbf{x}_t, k_t)$ | 5:        $\mathbf{z}_t^{\text{new}} \sim p_\theta(\mathbf{z}_t \mid \mathbf{z}_{t-1}, \mathbf{x}_t, \mathcal{K}_{m+1, t})$. |
| 6:        Define $\epsilon_t = \frac{\mathbf{x}_t^{k_t} - \sqrt{\bar{\alpha}_{k_t}} \mathbf{x}_t}{\sqrt{1 - \bar{\alpha}_{k_t}}}$ | 6:        $k \leftarrow \mathcal{K}_{m,t}, \mathbf{w} \sim \mathcal{N}(0, \mathbf{I})$. |
| 7:        Update $\mathbf{z}_t \sim p_\theta(\mathbf{z}_t \mid \mathbf{z}_{t-1}, \mathbf{x}_t^{k_t}, k_t)$. | 7:        $\mathbf{x}_t^{\text{new}} \leftarrow \frac{1}{\sqrt{\alpha_k}}(\mathbf{x}_t - \frac{1-\alpha_k}{\sqrt{1-\bar{\alpha}_k}} \epsilon_\theta(\mathbf{z}_t^{\text{new}}, \mathbf{x}_t, k)) + \sigma_k \mathbf{w}$ |
| 8:        Set $\hat{\epsilon}_t = \epsilon_\theta(\mathbf{z}_{t-1}, \mathbf{x}_t^{k_t}, k_t)$ | 8:        **Update** $\mathbf{z}_t \leftarrow \mathbf{z}_t^{\text{new}}$. |
| 9:    **end for** | 9:    **end for** |
| 10:   $L = \text{MSELoss}([\hat{\epsilon}_1, ..., \hat{\epsilon}_n], [\epsilon_1, ..., \epsilon_n])$ | 10:   $\mathbf{x}_{1:H} \leftarrow \text{AddGuidance}(\mathbf{x}_{1:H}^{\text{new}}, \nabla_\mathbf{x} \log c(\mathbf{x}_{1:H}^{\text{new}}))$ |
| 11:   Backprop with $L$ and update $\theta$ | 11: **end for** |
| 12: **end loop** | 12: **Return** $\mathbf{x}_{1:T}$. |

paper, we focus on time series data, and thus instantiate Diffusion Forcing with causal architectures (where $\mathbf{x}_t^{k_t}$ depends only on past noisy tokens), which we call *Causal Diffusion Forcing* (CDF). For simplicity, we focus on a minimal implementation with a vanilla Recurrent Neural Network (RNN) [11]. Potential transformer implementation of Diffusion Forcing is also possible but we defer its discussion to Appendix C.1.

The RNN with weights $\theta$ maintains latents $\mathbf{z}_t$ capturing the influence of past tokens, and these evolve via dynamics $\mathbf{z}_t \sim p_\theta(\mathbf{z}_t | \mathbf{z}_{t-1}, \mathbf{x}_t^{k_t}, k_t)$ with a recurrent layer. When an incoming noisy observation $\mathbf{x}_t^{k_t}$ is made, the hidden state is updated in a Markovian fashion $\mathbf{z}_t \sim p_\theta(\mathbf{z}_t | \mathbf{z}_{t-1}, \mathbf{x}_t^{k_t}, k_t)^2$. When $k_t = 0$, this is the posterior update in Bayes filtering; whereas when $k_t = K$ (and $\mathbf{x}_t^K$ is pure noise and thus uninformative), this is equivalent to modeling the "prior distribution" $p_\theta(\mathbf{z}_t | \mathbf{z}_{t-1})$ in Bayes filtering. Given latent $\mathbf{z}_t$, an observation model $p_\theta(\mathbf{x}_t^0 | \mathbf{z}_t)$ predicts $\mathbf{x}_t$.

**Training.** The dynamics model $p_\theta(\mathbf{z}_t | \mathbf{z}_{t-1}, \mathbf{x}_t^{k_t}, k_t)$ and the observation model $p_\theta(\mathbf{x}_t^0 | \mathbf{z}_t)$ together form a RNN unit. Such unit has the same input-output behavior as a standard conditional diffusion model, using a conditioning variable $\mathbf{z}_{t-1}$ and a noisy token $\mathbf{x}_t^{k_t}$ as input to predict the noise-free $\mathbf{x}_t = \mathbf{x}_t^0$ and thus, indirectly, the noise $\epsilon^{k_t}$ via affine reparametrization [29]. We can thus directly train (Causal) Diffusion Forcing with the conventional diffusion training objective. We parameterize the aforementioned unit in terms of noise prediction $\epsilon_\theta(\mathbf{z}_{t-1}, \mathbf{x}_t^{k_t}, k_t)$. We then find parameters $\theta$ by minimizing the loss

$$\underset{\substack{k_t, \mathbf{x}_t, \epsilon_t \\ \mathbf{z}_t \sim p_\theta(\mathbf{z}_t | \mathbf{z}_{t-1}, \mathbf{x}_t^{k_t}, k_t)}}{\mathbb{E}} \sum_{t=1}^{T} \Big[ \| \epsilon_t - \epsilon_\theta(\mathbf{z}_{t-1}, \mathbf{x}_t^{k_t}, k_t) \|^2 \Big], \tag{3.1}$$

where we sample $k_{1:T}$ uniformly from $[K]^T$, $\mathbf{x}_{1:T}$ from our training data, and $\epsilon_t \sim \mathcal{N}(0, \sigma_{k_t}^2 I)$ in accordance with the forward diffusion process (see Algorithm 1 for pseudocode). Importantly, the loss (3.1) captures essential elements of Bayesian filtering and conditional diffusion. In Appendix B.1, we further re-derive common techniques in diffusion model training for Diffusion Forcing, which proves extremely useful for video prediction experiments. In Appendix C.2, we discuss the need of sampling $k_{1:T}$ uniformly. Finally, we prove the validity of this objective stated informally in the following Theorem 3.1 in Appendix A.

**Theorem 3.1** (Informal). *The Diffusion Forcing training procedure (Algorithm 1) optimizes a reweighting of an Evidence Lower Bound (ELBO) on the expected log-likelihoods $\ln p_\theta((\mathbf{x}_t^{k_t})_{1 \leq t \leq T})$, where the expectation is averaged over noise levels $k_{1:T} \sim [K]^T$ and $\mathbf{x}_t^{k_t}$ noised according to the forward process. Moreover, under appropriate conditions, optimizing (3.1) also maximizes a lower bound on the likelihood for* all *sequences of noise levels, simultaneously.*

---

[2]We implement $\mathbf{z}_t = p_\theta(\mathbf{z}_t | \mathbf{z}_{t-1}, \mathbf{x}_t^{k_t}, k_t)$ to be deterministic, with $\mathbf{z}_t$ representing a distribution over beliefs rather than a sample from it. This allows training by backpropogating through the latent dynamics in Eq.(3.1).

We remark that a special case of 'all sequences of noise levels' are those for which either $k_t = 0$ or $k_t = K$; thus, one can mask out *any prior token* and DF will learn to sample from the correct conditional distribution, modeling the distribution of all possible sub-sequences of the training set.

**Sampling.** Diffusion Forcing sampling is depicted in Algorithm 2 and is defined by prescribing a noise schedule on a 2D $M \times T$ grid $\mathcal{K} \in [K]^{M \times T}$; columns correspond to time step $t$ and rows indexed by $m$ determine noise-level. $\mathcal{K}_{m,t}$ represents the desired noise level of the time-step $t$ token for row $m$. To generate a whole sequence of length $T$, initialize the tokens $\mathbf{x}_{1:T}$ to be white noise, corresponding to noise level $k = K$. We iterate down the grid row-by-row, denoising left-to-right across columns to the noise levels prescribed by $\mathcal{K}$. By the last row $m = 0$, the tokens are clean, i.e. their noise level is $\mathcal{K}_{0,t} \equiv 0$. Appendix B.5 discusses corner cases of this scheme; the hyperparameters $(\alpha_k, \bar{\alpha}_k, \sigma_k)$ are set to their standard values [29]. The matrix $\mathcal{K}$ specifies how fast each token gets denoised at every step of sequence diffusion. Since Diffusion Forcing is trained to denoise tokens of all sequences of noise levels, $\mathcal{K}$ can be designed to flexibly achieve different behaviors without re-training the model.

### 3.3 New Capabilities in Sequence Generation

We now explain the new capabilities this flexible sampling paradigm has to offer.

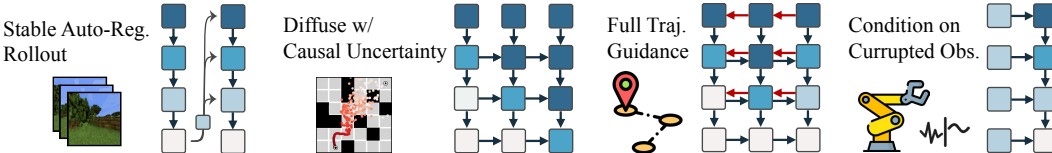

**Stabilizing autoregressive generation.** For high-dimensional, continuous sequences such as video, auto-regressive architectures are known to diverge, especially when sampling past the training horizon. In contrast, Diffusion Forcing can stably roll out long sequences even beyond the training sequence length by updating the latents using the previous latent associated with slightly "noisy tokens" for some small noise level $0 < k \ll K$. Our experiments (Sec. 4.1) illustrates the resulting marked improvements in long-horizon generation capabilities; App. C.4 provides further intuition.

**Keeping the future uncertain.** Beginning from a sequence of white noise tokens $[\mathbf{x}_1^K, \mathbf{x}_2^K, \mathbf{x}_3^K]^\top$, we may denoise the first token fully and the second token partially, yielding $[\mathbf{x}_1^0, \mathbf{x}_2^{K/2}, \mathbf{x}_3^K]^\top$, then $[\mathbf{x}_1^0, \mathbf{x}_2^0, \mathbf{x}_3^{K/2}]^\top$, and finally denoising all tokens fully to $[\mathbf{x}_1^0, \mathbf{x}_2^0, \mathbf{x}_3^0]^\top$. Interpreting the noise level as uncertainty, this "zig-zag" sampling scheme intuitively encodes the immediate future as more certain than the far future. Sec. 3.4 describes how this leads to more effective sequence guidance.

**Long-horizon Guidance.** In Line 10 of Algorithm 2, one may add guidance to the partially diffused trajectory $\mathbf{x}_{1:T}$ as in Sec. 2. Due to the dependency of future tokens on the past, guidance gradients from future tokens can propagate backwards in time. The unique advantage of Diffusion Forcing is that, because we can diffuse future tokens without fully diffusing the past, the gradient guides the sampling of *past* tokens, thereby achieving long-horizon guidance while respecting causality. We elaborate on implementation details in Appendix C.3. As we show in Section 4.2, planning in this manner significantly outperforms guided full-sequence diffusion models.

### 3.4 Diffusion Forcing for Flexible Sequential Decision Making

The capabilities offered by Diffusion Forcing motivate our novel framework for sequential decision making (SDM), with key applications to robotics and autonomous agents. Consider a Markov Decision Process defined by an environment with dynamics $p(\mathbf{s}_{t+1}|\mathbf{s}_t, \mathbf{a}_t)$, observation $p(\mathbf{o}_t|\mathbf{s}_t)$ and reward $p(\mathbf{r}_t|\mathbf{s}_t, \mathbf{a}_t)$. The goal is to train a policy $\pi(\mathbf{a}_t|\mathbf{o}_{1:t})$ such that the expected cumulative reward of a trajectory $\mathbb{E}[\sum_{t=1}^T \mathbf{r}_t]$ is maximized. We assign tokens $\mathbf{x}_t = [\mathbf{a}_t, \mathbf{r}_t, \mathbf{o}_{t+1}]$. A trajectory is a sequence $\mathbf{x}_{1:T}$, possibly of variable length; training is conducted as in Algorithm 1. At each step $t$ of execution, past (noise-free) tokens $\mathbf{x}_{1:t-1}$ are summarized by a latent $\mathbf{z}_{t-1}$. Conditioned on this latent, we sample, via Algorithm 2, a plan $\hat{\mathbf{x}}_{t:t+H}$, with $\hat{\mathbf{x}}_t = [\hat{\mathbf{a}}_t, \hat{\mathbf{r}}_t, \hat{\mathbf{o}}_{t+1}]^\top$ containing predicted actions, rewards and observations. $H$ is a look-ahead window, analogous to future predictions in model predictive control [20]. After taking planned action $\hat{\mathbf{a}}_t$, the environment produces a reward $\mathbf{r}_t$

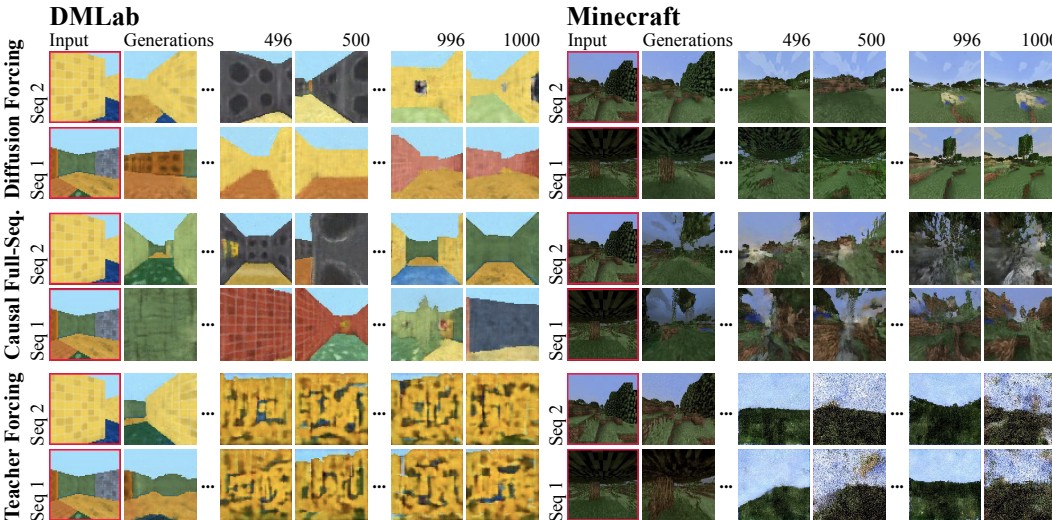

Figure 3: **Video Generation.** Among tested methods, Diffusion Forcing generations are uniquely temporally consistent and do not diverge even when rolling out well past the training horizon. Please see the project website for video results.

and next observation $\mathbf{o}_{t+1}$, yielding next token $\mathbf{x}_t = [\hat{\mathbf{a}}_t, \mathbf{r}_t, \mathbf{o}_{t+1}]^\top$. The latent is updated according to the posterior $p_\theta(\mathbf{z}_t | \mathbf{z}_{t-1}, \mathbf{x}_t, 0)$. Our framework enables functionality as both *policy* and *planner*:

**Flexible planning horizon.** Diffusion Forcing (a) can be deployed on *tasks of variable horizon*, because each new action is selected sequentially, and (b) its lookahead window $H$ can be shortened to lower latency (using Diffusion Forcing as a *policy*), or lengthened to perform long-horizon *planning* (via guidance described below), without re-training or modifications of the architecture. Note that (a) is not possible for full-sequence diffusion models like Diffuser [36] with full-trajectory generation horizons, whereas diffusion policies [10] need fixed, small lookahead sizes, precluding (b).

**Flexible reward guidance.** As detailed in Appendix C.3, Diffusion Forcing can plan via guidance using any reward (in place of $\log c$) specified over future steps: this includes dense per-time step rewards on the entire trajectory $\sum_{t=1}^T \mathbf{r}_t$, dense rewards on a future lookahead $\sum_{t'=t}^{t+H} \mathbf{r}_t$, and sparse rewards indicating goal completion $-\|\mathbf{o}_T - \mathbf{g}\|^2$. Per-time step policies cannot take advantage of this latter, longer horizon guidance.

**Monte Carlo Guidance (MCG), future uncertainty.** Causal Diffusion Forcing allows us to influence the generation of a token $\mathbf{x}_t^k$ by guidance on the whole distribution of future $\mathbf{x}_{t+1:T}$. Instead of drawing a single trajectory sample to calculate this guidance gradient, we can draw multiple samples of the future and average their guidance gradients. We call this Monte Carlo Guidance. In the spirit of so-called shooting methods like MPPI [63], $\mathbf{x}_t^k$ is then guided by the expected reward over the distribution of all future outcomes instead of one particular outcome. The effect of MCG is enhanced when combined with sampling schedules that keep the noise level of future tokens high when denoising immediate next tokens (e.g. the zig-zag schedule described in Sec. 3.3), accounting for greater uncertainty farther into the future. Appendix C.5 further justifies the significance of MCG, and why Diffusion Forcing uniquely takes advantage of it.

## 4 Experiments

We extensively evaluate Diffusion Forcing's merits as a generative sequence model across diverse applications in video and time series prediction, planning, and imitation learning. Please find the dataset and reproducibility details in the Appendix, as well as video results on the project website.

### 4.1 Video Prediction: Consistent, Stable Sequence Generation and Infinite Rollout.

We train a convolutional RNN implementation of Causal Diffusion Forcing for video generative modeling on videos of Minecraft gameplay [68] and DMLab navigation [68]. At sampling time, we

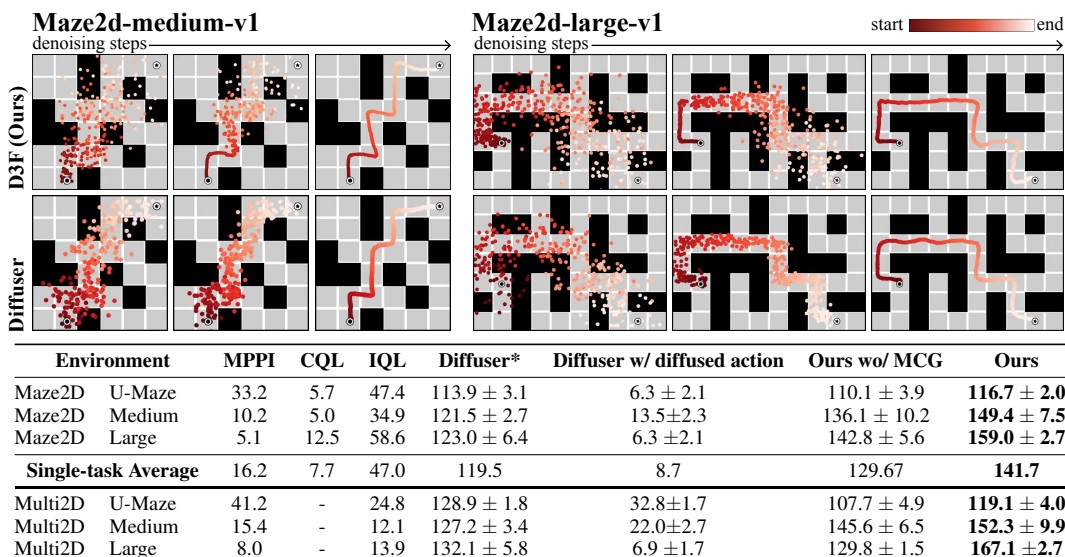

| Environment | | MPPI | CQL | IQL | Diffuser* | Diffuser w/ diffused action | Ours wo/ MCG | Ours |
|---|---|---|---|---|---|---|---|---|
| Maze2D | U-Maze | 33.2 | 5.7 | 47.4 | $113.9 \pm 3.1$ | $6.3 \pm 2.1$ | $110.1 \pm 3.9$ | $\mathbf{116.7 \pm 2.0}$ |
| Maze2D | Medium | 10.2 | 5.0 | 34.9 | $121.5 \pm 2.7$ | $13.5 \pm 2.3$ | $136.1 \pm 10.2$ | $\mathbf{149.4 \pm 7.5}$ |
| Maze2D | Large | 5.1 | 12.5 | 58.6 | $123.0 \pm 6.4$ | $6.3 \pm 2.1$ | $142.8 \pm 5.6$ | $\mathbf{159.0 \pm 2.7}$ |
| **Single-task Average** | | 16.2 | 7.7 | 47.0 | 119.5 | 8.7 | 129.67 | **141.7** |
| Multi2D | U-Maze | 41.2 | - | 24.8 | $128.9 \pm 1.8$ | $32.8 \pm 1.7$ | $107.7 \pm 4.9$ | $\mathbf{119.1 \pm 4.0}$ |
| Multi2D | Medium | 15.4 | - | 12.1 | $127.2 \pm 3.4$ | $22.0 \pm 2.7$ | $145.6 \pm 6.5$ | $\mathbf{152.3 \pm 9.9}$ |
| Multi2D | Large | 8.0 | - | 13.9 | $132.1 \pm 5.8$ | $6.9 \pm 1.7$ | $129.8 \pm 1.5$ | $\mathbf{167.1 \pm 2.7}$ |
| **Multi-task Average** | | 21.5 | - | 16.9 | 129.4 | 20.6 | 127.7 | **146.2** |

Table 1: **Diffusion Forcing for Planning.** (**top**) During sampling, Diffusion Forcing allows each time step to be denoised on different noise schedules, enabling us to account for causal uncertainty during guided planning. Diffusion Forcing keeps the far future more uncertain than the near future while Diffuser [36] puts them at the same noise level during sampling. (**bottom**) Quantitatively, Diffusion Forcing achieves the highest average reward across runs. Diffuser fails dramatically when executing the actually generated actions, requiring a hand-crafted PD controller (indicated by the asterisk) and ignoring generated actions.

perform auto-regressive rollout with stabilization proposed in Sec. 3.3. We consider two baselines, both leveraging the same exact RNN architecture: a next-frame diffusion baseline trained with teacher forcing [64] as well as a causal full-sequence diffusion model. Figure 3 displays qualitative results of roll-outs generated by Diffusion Forcing and baselines starting from unseen frames for both datasets. While Diffusion Forcing succeeds at stably rolling out even far beyond its training horizon (e.g. 1000 frames), teacher forcing and full-sequence diffusion baselines diverge quickly. Further, within the training horizon, we observe that full-sequence diffusion suffers from frame-to-frame discontinuity where video sequences jump dramatically, while Diffusion Forcing roll-outs show ego-motion through a consistent 3D environment. This highlights the ability of Diffusion Forcing to stabilize rollouts of high-dimensional sequences without compounding errors.

## 4.2 Diffusion Planning: MCG, Causal Uncertainty, Flexible Horizon Control.

Decision-making uniquely benefits from Diffusion Forcing's capabilities. We evaluate our proposed decision-making framework in a standard offline RL benchmark, D4RL [18]. Specifically, we benchmark Diffusion Forcing on a set of 2D maze environments with sparse reward. An agent is tasked with reaching a designated goal position starting from a random starting position. In Appendix E.5 we provide a detailed description of the environment. The benchmark provides a dataset of *random walks* through mazes (thus stochastic). We train one model per maze.

We benchmark the proposed decision-making framework 3.4 with state-of-the-art offline RL methods and the recently introduced Diffuser [36], a diffusion planning framework. See Fig. 1 for qualitative and quantitative results: DF outperforms Diffuser and all baselines across all 6 environments.

**Benefit of Monte Carlo Guidance.** The typical goal for an RL problem is to find actions that maximize the *expected* future rewards, which we achieve through MCG. Full-sequence diffusion models such as Diffuser do not support sampling to maximize expected reward, as we formally derive in Appendix C.5. To understand MCG's importance, we ablate it in Table 1. Removing MCG guidance degrades our performance, though Diffusion Forcing remains competitive even then.

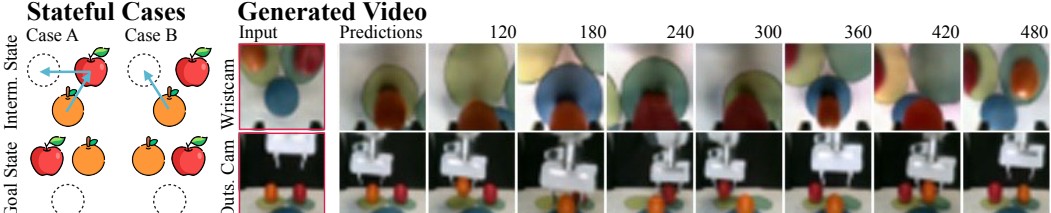

Figure 4: In our real robot task, a robot arm is asked to swap the slots of two fruits using a third slot. Since the fruits are input in random slots at the beginning, one cannot determine the next steps from a single observation without knowledge of the initial placement of the fruits. As illustrated in (a) and (b), the upper observation is the same but the desired outcome illustrated below can vary—the task thus requires remembering the initial configuration. In addition, as shown in (c), the same model that generates actions also synthesizes realistic video from just a single frame.

**Benefit of Modeling Causality.** Unlike pure generative modeling, sequential decision-making takes actions and receives feedback. Due to compounding uncertainty, the immediate next actions are more important than those in the far future. Though Diffuser and subsequent models are trained to generate sequences of action-reward-state tuples $[\mathbf{a}_t, \mathbf{r}_t, \mathbf{o}_t]$, directly executing the actions will lead to a trajectory that deviates significantly from the generated states. In other words, the generated states and actions are not causally consistent with each other. To address this shortcoming, Diffuser's implementation ignores the generated actions and instead relies on a hand-crafted PD controller to infer actions from generated states. In Table 1, we see that Diffuser's performance drops dramatically when directly executing generated actions. In contrast, Diffusion Forcing's raw action generations are self-consistent, outperforming even actions selected by combining Diffuser's state predictions with a handcrafted PD controller.

**Benefit of Flexible Horizon.** Many RL tasks have a fixed horizon, requiring the planning horizon to shrink as an agent makes progress in the task. Diffusion Forcing accomplishes this by design, while full-sequence models like Diffuser perform poorly even with tweaks, as we explain in Appendix C.6.

### 4.3 Controllable Sequential Compositional Generation

We demonstrate that by only modifying the sampling scheme, we can flexibly compose sub-sequences of sequences observed at training time. We consider a dataset of trajectories on a 2D, square plane, where all trajectories start from one corner and end up in the opposite corner, forming a cross shape. As shown in Fig. 1, when no compositional behavior is desired, one can let DF keep full memory, replicating the cross-shaped distribution. When one desires compositionality, one can let the model generate shorter plans without memory using MPC, leading to the stitching of the cross's sub-trajectories, forming a V-shaped trajectory. Due to limited space, we defer the result to Appendix E.2.

### 4.4 Robotics: Long horizon imitation learning and robust visuomotor control

Finally, we illustrate that Diffusion Forcing (DF) opens up new opportunities in the visuomotor control of real-world robots. Imitation learning [10] is a popular technique in robotic manipulation where one learns an observation-to-action mapping from expert demonstrations. However, the lack of memory often prevents imitation learning from accomplishing long-horizon tasks. DF not only alleviates this shortcoming but also provides a way to make imitation learning robust.

**Imitation Learning with Memory.** We collect a dataset of videos and actions by teleoperating with a Franka robot. In the chosen task, one needs to swap the position of an apple and an orange, using a third slot. See Fig. 4 for an illustration. The initial positions of the fruits are randomized such that there are two possible goal states. As illustrated in Fig. 4, when one fruit is in the third slot, the desired outcome cannot be inferred from the current observation—a policy must remember the initial configuration to determine which fruit to move. In contrast to common behavior cloning methods, DF naturally incorporates memory in its latent state. We found that DF achieves $80\%$ success rate while diffusion policy [10], a state-of-the-art imitation learning algorithm without memory, fails.

**Robustness to missing or noisy observations.** Because it incorporates principles from Bayes filtering, Diffusion Forcing can perform imitation learning while being robust to noisy or missing observations. We demonstrate this by adding visual distractions and even fully occluding the camera during execution. DF allows us to easily indicate these observations as "noisy" by using $k > 0$, in which case DF relies heavily on its prior model to predict actions. Consequently, the success rate is only lowered by $4\%$ to $76\%$. In contrast, a next-frame diffusion model baseline attains a success rate of $48\%$: it must treat perturbed observations as ground truth and suffers out-of-distribution error.

**Potential for pre-training with video.** Finally, in parallel to generating actions, Fig. 4 illustrates that Diffusion Forcing is capable of generating a video of the robot performing the task given only an initial frame, unifying diffusion policy/imitation learning and video generative modeling and paving the way to pre-training on unlabeled video.

### 4.5 Time Series Forecasting: Diffusion Forcing is a Good General-purpose Sequence Model

In Appendix E, we show that DF is competitive with prior diffusion [49] and transformer-based [50] work on multivariate time series forecasting, following the experimental setup of [53].

## 5 Discussion

**Limitations.** Our current causal implementation is based on an RNN. Applications to higher-resolution video or more complex distributions likely require large transformer models following instructions in Appendix C.1. We do not investigate the scaling behavior of Diffusion Forcing to internet-scale datasets and tasks.

**Conclusion.** In this paper, we introduced Diffusion Forcing, a new training paradigm where a model is trained to denoise sets of tokens with independent, per-token noise levels. Applied to time series data, we show how a next-token prediction model trained with Diffusion Forcing combines the benefits of both next-token models and full-sequence diffusion models. We introduced new sampling and guidance schemes that lead to dramatic performance gains when applied to tasks in sequential decision making. Future work may investigate the application of Diffusion Forcing to domains other than time series generative modeling, and scale up Diffusion Forcing to larger datasets.

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

# A Theoretical Justification

In this section, we provide theoretical justification for the train of Diffusion Forcing. The main contributions can be summarized as follows:

- We show that our training methods optimize a reweighting of the Evidence Lower Bound (ELBO) on the average log-likelihood of our data. We first establish this in full generality (Theorem A.1), and then specialize to the form of Gaussian diffusion (Corollary A.2). We show that the resulting terms decouple in such a fashion that, in the limit of a fully expressive latent and model, makes the reweighting terms immaterial.

- We show that the expected likelihood over *any* distribution over sequences of noise levels can be lower bounded by a sum over nonnegative terms which, when reweighted, correspond to the terms optimized in the Diffusion Forcing training objective maximizes. Thus, for a fully expressive network that can drive all terms to their minimal value, Diffusion Forcing optimizes a valid surrogate of the likelihood of *all sequences of noise levels simultaneously.*

We begin by stating an ELBO for general Markov forward processes $q(\cdot)$, and generative models $p_{\boldsymbol{\theta}}(\cdot)$, and then specialize to Gaussian diffusion, thereby recovering our loss. We denote our Markov forward process $q(\cdot)$ as

$$q(\mathbf{x}^{1:K} \mid \mathbf{x}^0) = \prod_{k=1}^{K} q(\mathbf{x}^k \mid \mathbf{x}^{k-1}), \tag{A.1}$$

and a parameterized probability model

$$p_{\boldsymbol{\theta}}(((\mathbf{x}_t^k)_{1 \leq k \leq K}, \mathbf{z}_t)_{t \geq 1}) \tag{A.2}$$

We assume that $p_{\boldsymbol{\theta}}$ satisfies the *Markov property* that

$$p_{\boldsymbol{\theta}}(\mathbf{z}_t, \mathbf{x}_t^{k_t} \mid \mathbf{z}_{1:t-1}, (\mathbf{x}_s^{k_s})_{1 \leq s < t}) = p_{\boldsymbol{\theta}}(\mathbf{z}_t, \mathbf{x}^{k_t} \mid \mathbf{z}_{t-1}) \tag{A.3}$$

that is, the latent codes $\mathbf{z}_{t-1}$ is a sufficient statistic for $\mathbf{x}^{k_t}$ given the history. We say that $p_{\boldsymbol{\theta}}$ has *deterministic latents* if $p_{\boldsymbol{\theta}}(\mathbf{z}_t \mid \mathbf{z}_{1:t-1}, (\mathbf{x}_s^{k_s})_{1 \leq s < t}, \mathbf{x}_t^{k_t})$ is a Dirac delta.

*Remark* 1. In order for $p_{\boldsymbol{\theta}}$ to have deterministic latents and correspond to a valid probability distribution, we need to view the latents $\mathbf{z}_t$ not as individual variables, but as a collection of variables $\mathbf{z}_t(k_{1:t})$ indexed by $t \in [T]$ and the *history* of noise levels $k_{1:t} \in \{0, 1, \ldots, K\}^t$. In this case, simply setting $\mathbf{z}_t(k_{1:t}) = (k_{1:t}, (\mathbf{x}_s^{k_s})_{1 \leq s \leq t})$ tautologically produces deterministic latents. The reason for indexing $\mathbf{z}_t(k_{1:t})$ with $k_{1:t}$ then arises because, otherwise, $p_{\boldsymbol{\theta}}(\mathbf{z}_t \mid ((\mathbf{x}_s^{k_s})_{1 \leq s \leq t}, (\mathbf{x}_s^{k_s'})_{1 \leq s \leq t})$ would be ill-defined unless $k_s = k_s'$ for all $1 \leq s \leq t$, and thus, $p_{\boldsymbol{\theta}}$ would not correspond to a joint probability measure. The exposition and theorem that follows allow $\mathbf{z}_t(k_{1:t})$ to be indexed on past noise levels $k_{1:t}$ but suppresses dependence on $k_{1:t}$ to avoid notational confusion.

## A.1 Main Results

We can now state our main theorem, which provides an evidence lower bound (ELBO) on the expected log-likelihood of partially-noised sequences $(\mathbf{x}_t^{k_t})_{1 \leq t \leq T}$, under uniformly sampled levels $k_t$ and $\mathbf{x}_t^{k_t}$ obtained by noising according to $q(\cdot)$ as in (A.1). Notice that this formulation does not require an explicit for of $q(\cdot)$ or $p_{\boldsymbol{\theta}}$, but we will specialize to Gaussian diffusion in the following section.

**Theorem A.1.** *Fix* $\mathbf{x}_{1:T}^0$. *Define the expectation over the forward process with random noise level* $k_{1:T}$ *as*

$$\mathbb{E}_{\text{forward}}[\cdot] := \mathbb{E}_{k_1, \ldots, k_T \overset{\text{unif}}{\sim} [K]} \mathbb{E}_{\mathbf{x}_s^{k_s} \sim q(\mathbf{x}_s^{k_s} \mid \mathbf{x}_s^0), 1 \leq s \leq T}[\cdot], \tag{A.4}$$

*and the expectation over the latents under* $p_{\boldsymbol{\theta}}(\cdot)$ *conditioned on* $k_{1:T}, (\mathbf{x}_s^{k_t})_{1 \leq t \leq T}$ *as*

$$\mathbb{E}_{p, \mathbf{z}_{1:T}}[\cdot] := \mathbb{E}_{\mathbf{z}_s \sim p(\mathbf{z}_s \mid \mathbf{z}_{s-1}, \mathbf{x}_s^{k_s}), s \leq T}\left[\cdot \mid k_{1:T}, (\mathbf{x}_t^{k_t})_{1 \leq t \leq T}\right] \tag{A.5}$$

*Then, as long as $p_{\boldsymbol{\theta}}$ satisfies the Markov property,*

$$\mathbb{E}_{\text{forward}}[\ln p_{\boldsymbol{\theta}}((\mathbf{x}_t^{k_t})_{1\leq t\leq T})] \geq C(\mathbf{x}_{1:T}^0)$$

$$+ \mathbb{E}_{\text{forward}}\mathbb{E}_{p,\mathbf{z}_{1:T}}\left[\sum_{t=1}^{T}\left(\frac{1}{K+1}\ln p_{\boldsymbol{\theta}}(\mathbf{x}_t^0 \mid \mathbf{x}_t^1, \mathbf{z}_{t-1}) + \sum_{j=2}^{K}\frac{j}{K+1}D_{\mathbb{KL}}\left(q(\mathbf{x}_t^{j-1} \mid \mathbf{x}_t^j, \mathbf{x}_t^0) \parallel p_{\boldsymbol{\theta}}(\mathbf{x}_t^j \mid \mathbf{x}_t^{j-1}, \mathbf{z}_{t-1})\right)\right)\right],$$

*where $C(\mathbf{x}_{1:T}^0)$ is a constant depending only on $\mathbf{x}_{1:T}^0$ (the unnoised data). Moreover, if the latents are deterministic (i.e. $p_{\boldsymbol{\theta}}(\mathbf{z}_t \mid \mathbf{z}_{t-1}, \mathbf{x}_t^{k_t})$ is a Dirac distribution), then the inequality holds with inequality if and only if $q(\mathbf{x}_t^{k_t+1:T} \mid \mathbf{x}_t^{k_t}) \equiv p_{\boldsymbol{\theta}}(\mathbf{x}_t^{k_t+1:T} \mid \mathbf{x}_t^{k_t}, \mathbf{z}_{t-1})$, i.e. the variational approximation is exact.*

The proof of the above theorem is given in Appendix A.2. Remarkably, it involves *only two* inequalities! The first holds with equality under deterministic latents and the second holds if and only if variational approximation is exact: $q(\mathbf{x}_t^{k_t+1:T} \mid \mathbf{x}_t^{k_t}) \equiv p_{\boldsymbol{\theta}}(\mathbf{x}_t^{k_t+1:T} \mid \mathbf{x}_t^{k_t}, \mathbf{z}_{t-1})$. This tightness of the ELBO suggests that the expression in Theorem A.1 is a relatively strong surrogate objective for optimizing the likelihoods.

### A.1.1 Specializing to Gaussian diffusion

We now special Theorem A.1 to Gaussian diffusion. For now, we focus on the "$\mathbf{x}$-prediction" formulation of diffusion, which is the one used in our implementation. The "$\epsilon$-prediction" formalism, used throughout the main body of the text, can be derived similarly (see Section 2 of [7] for a clean exposition). The following theorem follows directly by apply standard likelihood and KL-divergence computations for the DDPM [28, 7] to Theorem A.1.

**Corollary A.2.** *Let*

$$q(\mathbf{x}^{k+1} \mid \mathbf{x}_t^k) = \mathcal{N}(\mathbf{x}^k; \sqrt{1-\beta_k}\mathbf{x}^{k-1}, \beta_k\mathbf{I}), \tag{A.6}$$

*and define $\alpha_k = (1-\beta_k)$, $\bar{\alpha}_k = \prod_{j=1}^{k}\alpha_j$. Suppose that we parameterize $p_{\boldsymbol{\theta}}(\mathbf{x}_t^j \mid \mathbf{x}_t^{j+1}, \mathbf{z}_{t-1}) = \mathcal{N}(\mu_{\boldsymbol{\theta}}(\mathbf{x}_t^{j+1}, \mathbf{z}_{t-1}, j), \sigma_j^2)$, where further,*

$$\mu_{\boldsymbol{\theta}}(\mathbf{x}_t^j, \mathbf{z}_{t-1}, j) = \frac{(1-\bar{\alpha}_{j-1})\sqrt{\alpha_j}}{1-\bar{\alpha}_j}\mathbf{x}_t^j + \frac{(1-\alpha_j)\sqrt{\bar{\alpha}_{j-1}}}{1-\bar{\alpha}_j}\hat{\mathbf{x}}_{\boldsymbol{\theta}}(\mathbf{x}_t^j, \mathbf{z}_{t-1}, j), \quad \sigma_j^2 := \frac{(1-\alpha_j)(1-\sqrt{\bar{\alpha}_{j-1}})}{1-\bar{\alpha}_j}.$$

*Then, as long as $p_{\boldsymbol{\theta}}$ satisfies the Markov property, we obtained*

$$\mathbb{E}_{\text{forward}}[\ln p_{\boldsymbol{\theta}}((\mathbf{x}_t^{k_t})_{1\leq t\leq T})] + C(\mathbf{x}_{1:T}^0) \geq \mathbb{E}_{\text{forward}}\mathbb{E}_{p,\mathbf{z}_{1:T}}\left[\sum_{t=1}^{T}\frac{j}{K+1}\sum_{j=1}^{K}c_j\|\hat{\mathbf{x}}_{\boldsymbol{\theta}}^0(\mathbf{x}_t^j, \mathbf{z}_{t-1}, j) - \mathbf{x}_t^0\|^2\right]$$

$$= \mathbb{E}_{\text{forward}}\mathbb{E}_{p,\mathbf{z}_{1:T}}\left[\sum_{t=1}^{T}\mathbf{1}\{k_t \geq 1\}\cdot k_t c_{k_t}\|\hat{\mathbf{x}}_{\boldsymbol{\theta}}^0(\mathbf{x}_t^{k_t}, \mathbf{z}_{t-1}, k_t) - \mathbf{x}_t^0\|^2\right],$$

*where above, we define $c_j = \frac{(1-\alpha_j)^2\bar{\alpha}_{j-1}}{2\sigma^2(1-\bar{\alpha}_j)^2}$.*

*Proof.* The first inequality follows from the standard computations for the "$\mathbf{x}$-prediction" formulation of Diffusion (see Section 2.7 of [7] and references therein). The second follows by replacing the sum over $j$ with an expectation over $k_t \overset{\text{unif}}{\sim} \{0, 1, \ldots, K\}$. $\qquad\square$

We make a couple of remarks:

- As noted above, Corollary A.2 can also be stated for $\epsilon$-prediction, or the so-called "$\mathbf{v}$-prediction" formalism, as all are affinely related.

- Define an idealized latent $\tilde{\mathbf{z}}_{t-1}$ consisting of all past tokens $(\mathbf{x}_t^{k_t})$ as well as of their noise levels $k_t$. This is a sufficient statistic for $\mathbf{z}_{t-1}$, and thus we can always view

$\hat{\mathbf{x}}_{\boldsymbol{\theta}}^0(\mathbf{x}_t^{k_t}, \mathbf{z}_{t-1}, k_t) = \hat{\mathbf{x}}_{\boldsymbol{\theta}}^0(\mathbf{x}_t^{k_t}, \bar{\mathbf{z}}_{t-1}, k_t)$, where $\mathbf{z}_{t-1}$ is just compressing $\bar{\mathbf{z}}_{t-1}$. When applying the expectation of $\mathbf{x}_{1:T} \sim q$ to both sides of the bound in Corollary A.2, and taking an infimum over possible function approximator $\hat{\mathbf{x}}_{\theta}^0$, we obtain

$$\inf_{p_\theta} \mathop{\mathbb{E}}_q \mathop{\mathbb{E}}_{\text{forward}} \mathop{\mathbb{E}}_{p,\mathbf{z}_{1:T}} \|\hat{\mathbf{x}}_{\boldsymbol{\theta}}^0(\mathbf{x}_t^{k_t}, \mathbf{z}_{t-1}, k_t) - \mathbf{x}_t^0\|^2 = \inf_{p_\theta} \mathop{\mathbb{E}}_q \mathop{\mathbb{E}}_{\text{forward}} \mathop{\mathbb{E}}_{p,\mathbf{z}_{1:T}} \|\hat{\mathbf{x}}_{\boldsymbol{\theta}}^0(\mathbf{x}_t^{k_t}, \bar{\mathbf{z}}_{t-1}) - \mathbf{x}_t^0\|^2$$
$$= \mathbf{Var}_q[\mathbf{x}_t^0 \mid (\mathbf{x}_s^{k_s})_{1 \le s \le t}, k_1, \ldots, k_t].$$

This leads to a striking finding: with expressive enough latents and $p_\theta$, we can view the maximization of each term in Corollary A.2 separately across time steps. The absence of this coupling means that the weighting terms are immaterial to the optimization, and thus can be ignored.

- Given the above remarks, we can optimize the ELBO by taking gradients through the objective specified by Corollary A.2, and are free to drop any weighting terms (or rescale them) as desired. Backpropagation through $\mathbb{E}_{p,\mathbf{z}_{1:T}}$ is straightforward due to deterministic latents. This justifies the correctness of our training objective (3.1) and protocol Algorithm 1.

### A.1.2 Capturing all subsequences

Theorem A.1 stipulates that, up to reweighting, the Diffusion Forcing objective optimizes a valid ELBO on the expected log-likelihoods over uniformly sampled noise levels. The following theorem can be obtained by a straightforward modification of the proof of Theorem A.1 generalizes this to arbitrary (possibly temporally correlated) sequences of noise.

**Theorem A.3.** *Let $\mathcal{D}$ be an arbitrary distribution over $[K]^T$, and define $P_t(j \mid k_{1:t-1}) := \Pr_{\mathcal{D}}[k_t = j \mid k_{1:t-1}]$. Fix $\mathbf{x}_{1:T}^0$. Define the expectation over the forward process with random noise level $k_{1:T}$ as*

$$\mathop{\mathbb{E}}_{\text{forward},\mathcal{D}}[\cdot] := \mathop{\mathbb{E}}_{k_1,\ldots,k_T \sim \mathcal{D}} \mathop{\mathbb{E}}_{\mathbf{x}_s^{k_s} \sim q(\mathbf{x}_s^{k_s} \mid \mathbf{x}_s^0), 1 \le s \le T}[\cdot], \tag{A.7}$$

*and the expectation over the latent under $p_{\boldsymbol{\theta}}(\cdot)$ conditioned on $k_{1:T}, (\mathbf{x}_s^{k_t})_{1 \le t \le T}$ as*

$$\mathop{\mathbb{E}}_{p,\mathbf{z}_{1:T}}[\cdot] := \mathop{\mathbb{E}}_{\mathbf{z}_s \sim p(\mathbf{z}_s \mid \mathbf{z}_{s-1}, \mathbf{x}_s^{k_s}), s \le T}\left[\cdot \mid k_{1:T}, (\mathbf{x}_t^{k_t})_{1 \le t \le T}\right] \tag{A.8}$$

*Then, as long as $p_\theta$ satisfies the Markov property,*

$$\mathop{\mathbb{E}}_{\text{forward},\mathcal{D}}[\ln p_{\boldsymbol{\theta}}((\mathbf{x}_t^{k_t})_{1 \le t \le T})] \ge C(\mathbf{x}_{1:T}^0) + \mathop{\mathbb{E}}_{\text{forward},\mathcal{D}} \mathop{\mathbb{E}}_{p,\mathbf{z}_{1:T}}\left[\sum_{t=1}^T \Xi_t\right], where$$

$$\Xi_t := \left(P_t(1 \mid k_{1:t-1}) \ln p_{\boldsymbol{\theta}}(\mathbf{x}_t^0 \mid \mathbf{x}_t^1, \mathbf{z}_{t-1}) + \sum_{j=2}^K j P_t(j \mid k_{1:t-1}) \mathrm{D}_{\mathrm{KL}}\left(q(\mathbf{x}_t^{j-1} \mid \mathbf{x}_t^j, \mathbf{x}_t^0) \parallel p_{\boldsymbol{\theta}}(\mathbf{x}_t^j \mid \mathbf{x}_t^{j-1}, \mathbf{z}_{t-1})\right)\right),$$

*where $C(\mathbf{x}_{1:T}^0)$ is a constant depending only on $\mathbf{x}_{1:T}^0$ (the noise-free data), and where the inequality is an* equality *under the conditions that (a) $p_{\boldsymbol{\theta}}(\mathbf{z}_t \mid \mathbf{z}_{t-1}, \mathbf{x}_t^{k_t})$ is a Dirac distribution (deterministic latents), and (b) $q(\mathbf{x}_t^{k_t+1:T} \mid \mathbf{x}_t^{k_t}) \equiv p_{\boldsymbol{\theta}}(\mathbf{x}_t^{k_t+1:T} \mid \mathbf{x}_t^{k_t}, \mathbf{z}_{t-1})$, i.e. the variational approximation is sharp.*

*In particular, in the Gaussian case of Corollary A.2, we have*

$$\mathop{\mathbb{E}}_{\text{forward},\mathcal{D}}[\ln p_{\boldsymbol{\theta}}((\mathbf{x}_t^{k_t})_{1 \le t \le T})] + C(\mathbf{x}_{1:T}^0) \ge \mathop{\mathbb{E}}_{\text{forward},\mathcal{D}} \mathop{\mathbb{E}}_{p,\mathbf{z}_{1:T}}\left[\sum_{t=1}^T \mathbf{1}\{k_t \ge 1\} k_t c_{k_t} \|\hat{\mathbf{x}}_{\boldsymbol{\theta}}^0(\mathbf{x}_t^{k_t}, \mathbf{z}_{t-1}, k_t) - \mathbf{x}_t^0\|^2\right],$$

The most salient case for us is the restriction of $\mathcal{D}$ to fixed sequences of noise $k_1, \ldots, k_T$ (i.e. Dirac distributions on $[K]^T$). In this case, $P_t(j \mid k_{1:t-1}) = 0$ for all but $j = k_t$, and thus our training objective need not be a lower bound on $\mathbb{E}_{\text{forward},\mathcal{D}}[\ln p_{\boldsymbol{\theta}}((\mathbf{x}_t^{k_t})_{1 \le t \le T})]$. However, the terms in the lower bound are, up to reweighting, an *subset* of those terms optimized in the training objective. Thus, in light of the remarks following Corollary A.2, a fully expressive network can optimize all the terms in the loss simultaneously. We conclude that, for a fully expressive neural network, optimizing the training objective (3.1) is a valid surrogate for maximizing the likelihood of all possible noise sequences.

## A.2 Proof of Theorem A.1

Define $\mathbb{E}_{<t}[\cdot]$ as shorthand for $\mathbb{E}_{k_{1:s} \overset{\text{unif}}{\sim} [K]} \mathbb{E}_{\mathbf{x}_s^{k_s} \sim q(\mathbf{x}_s^{k_s} | \mathbf{x}_s^0), 1 \le s \le t-1} \mathbb{E}_{\mathbf{z}_s \sim p(\mathbf{z}_s | \mathbf{z}_{s-1}, \mathbf{x}_s^{k_s}), s \le t}[\cdot]$. We begin with the following claim

**Claim 1** (Expanding the latents). *The following lower bound holds:*

$$\mathbb{E}_{\text{forward}}[\ln p_{\boldsymbol{\theta}}((\mathbf{x}_t^{k_t})_{1 \le t \le T})] \ge \sum_{t=1}^{T} \mathbb{E}_{<t} \mathbb{E}_{k_t \overset{\text{unif}}{\sim} \{0,1,\ldots,K\}} \mathbb{E}_{\mathbf{x}_t^{k_t} \sim q(\mathbf{x}_t^{k_t} | \mathbf{x}_t^0)} \left[ \ln p_{\boldsymbol{\theta}}(\mathbf{x}_t^{k_t} \mid \mathbf{z}_{t-1}) \right], \quad \text{(A.9)}$$

*Moreover, this lower bound holds with equality if $\mathbf{z}_s \sim p(\mathbf{z}_s \mid \mathbf{z}_{s-1}, \mathbf{x}_s^{k_s})$ is a Dirac distribution (i.e., deterministic latents).*

*Proof.* Let's fix a sequence $k_{1:T}$. It holds that

$$
\begin{aligned}
p_{\boldsymbol{\theta}}((\mathbf{x}_t^{k_t})_{1 \le t \le T}) &= \int_{\mathbf{z}_{1:T}} \prod_{t=1}^{T} p(\mathbf{x}_t^{k_t}, \mathbf{z}_t \mid (\mathbf{x}_s^{k_s}, \mathbf{z}_s)_{s<t}) \\
&= \int_{\mathbf{z}_{1:T}} \prod_{t=1}^{T} p(\mathbf{x}_t^{k_t}, \mathbf{z}_t \mid \mathbf{z}_{t-1}) && \text{(Markov Property)} \\
&= \int_{\mathbf{z}_{1:T}(\mathbf{k})} \prod_{t=1}^{T} p(\mathbf{z}_t \mid \mathbf{z}_{t-1}, \mathbf{x}_t^{k_t}) p_{\boldsymbol{\theta}}(\mathbf{x}_t^{k_t} \mid \mathbf{z}_{t-1}) \\
&= \mathbb{E}_{\mathbf{z}_s \sim p(\mathbf{z}_s | \mathbf{z}_{s-1}, \mathbf{x}_s^{k_s}), s \le T} \prod_{t=1}^{T} p_{\boldsymbol{\theta}}(\mathbf{x}_t^{k_t} \mid \mathbf{z}_{t-1}). && \text{(Importance Sampling)}
\end{aligned}
$$

Thus, by Jensen's inequality,

$$\ln p_{\boldsymbol{\theta}}((\mathbf{x}_t^{k_t})_{1 \le t \le T}) \ge \mathbb{E}_{\mathbf{z}_s \sim p(\mathbf{z}_s | \mathbf{z}_{s-1}, \mathbf{x}_s^{k_s}), s \le T} \sum_{t=1}^{T} \ln p_{\boldsymbol{\theta}}(\mathbf{x}_t^{k_t} \mid \mathbf{z}_{t-1}) = \mathbb{E}_{p, \mathbf{z}_{1:T}} \left[ \sum_{t=1}^{T} \ln p_{\boldsymbol{\theta}}(\mathbf{x}_t^{k_t} \mid \mathbf{z}_{t-1}) \right],$$

where the inequality is and equality when $p_{\boldsymbol{\theta}}(\mathbf{z}_s \mid \mathbf{z}_{s-1}, \mathbf{x}_s^{k_s})$ is a Dirac distribution. By applying $\mathbb{E}_{\text{forward}}$ to both sides of the above display, and invoking the Markov property of the latents, we conclude that

$$
\begin{aligned}
\mathbb{E}_{\text{forward}}[\ln p_{\boldsymbol{\theta}}((\mathbf{x}_t^{k_t})_{1 \le t \le T})] &\ge \mathbb{E}_{\text{forward}} \mathbb{E}_{p, \mathbf{z}_{1:T}} \left[ \sum_{t=1}^{T} \ln p_{\boldsymbol{\theta}}(\mathbf{x}_t^{k_t} \mid \mathbf{z}_{t-1}) \right] \\
&= \sum_{t=1}^{T} \mathbb{E}_{<t} \mathbb{E}_{k_t \overset{\text{unif}}{\sim} \{0,1,\ldots,K\}} \mathbb{E}_{\mathbf{x}_t^{k_t} \sim q(\mathbf{x}_t^{k_t} | \mathbf{x}_t^0)} \left[ \ln p_{\boldsymbol{\theta}}(\mathbf{x}_t^{k_t} \mid \mathbf{z}_{t-1}) \right].
\end{aligned}
$$

$\square$

We now unpack the terms obtained from the preceding claim.

**Claim 2** (ELBO w.r.t. $q$). *It holds that*

$$\mathbb{E}_{\mathbf{x}_t^{k_t} \sim q(\mathbf{x}_t^{k_t} | \mathbf{x}_t^0)} \left[ \ln p_{\boldsymbol{\theta}}(\mathbf{x}_t^{k_t} \mid \mathbf{z}_{t-1}) \right] \ge C_1(\mathbf{x}_0, k_t) + \left[ \mathbb{E}_{\mathbf{x}_t^{k_t:K} \sim q(\mathbf{x}_t^{k_t:K} | \mathbf{x}_t^0)} \ln \frac{p_{\boldsymbol{\theta}}(\mathbf{x}_t^{k_t:K} \mid \mathbf{z}_{t-1})}{q(\mathbf{x}_t^{k_t+1:K} \mid \mathbf{x}_t^0)} \right].$$

*where $C_1(\mathbf{x}_0, k_t)$ is a constant depending only on $\mathbf{x}_0$ and $k_t$, and where the inequality holds with equality if and only if $q(\mathbf{x}_t^{k_t+1:T} \mid \mathbf{x}_t^{k_t}) \equiv p_{\boldsymbol{\theta}}(\mathbf{x}_t^{k_t+1:T} \mid \mathbf{x}_t^{k_t}, \mathbf{z}_{t-1})$.*

*Proof.* We have that

$$\mathop{\mathbb{E}}_{\mathbf{x}_t^{k_t} \sim q(\mathbf{x}_t^{k_t} | \mathbf{x}_t^0)} \left[ \ln p_{\boldsymbol{\theta}}(\mathbf{x}_t^{k_t} \mid \mathbf{z}_{t-1}) \right]$$

$$= \mathop{\mathbb{E}}_{\mathbf{x}_t^{k_t} \sim q(\mathbf{x}_t^{k_t} | \mathbf{x}_t^0)} \left[ \ln \int p_{\boldsymbol{\theta}}(\mathbf{x}_t^{k_t:K} \mid \mathbf{z}_{t-1}) \mathrm{d}\mathbf{x}_t^{k_t+1:K} \right]$$

$$= \mathop{\mathbb{E}}_{\mathbf{x}_t^{k_t} \sim q(\mathbf{x}_t^{k_t} | \mathbf{x}_t^0)} \left[ \ln \left( \mathop{\mathbb{E}}_{\mathbf{x}_t^{k_t+1:K} \sim q(\mathbf{x}_t^{k_t+1:K} | \mathbf{x}_t^{k_t})} \left[ \frac{p_{\boldsymbol{\theta}}(\mathbf{x}_t^{k_t:K} \mid \mathbf{z}_{t-1})}{q(\mathbf{x}_t^{k_t+1:K} \mid \mathbf{x}_t^{k_t})} \right] \right) \right]$$

$$\geq \mathop{\mathbb{E}}_{\mathbf{x}_t^{k_t} \sim q(\mathbf{x}_t^{k_t} | \mathbf{x}_t^0)} \left[ \mathop{\mathbb{E}}_{\mathbf{x}_t^{k_t+1:K} \sim q(\mathbf{x}_t^{k_t+1:K} | \mathbf{x}_t^{k_t})} \left[ \ln \frac{p_{\boldsymbol{\theta}}(\mathbf{x}_t^{k_t:K} \mid \mathbf{z}_{t-1})}{q(\mathbf{x}_t^{k_t+1:K} \mid \mathbf{x}_t^{k_t})} \right] \right] \qquad \text{((Jensen's inequality))}$$

$$= \mathop{\mathbb{E}}_{\mathbf{x}_t^{k_t:K} \sim q(\mathbf{x}_t^{k_t:K} | \mathbf{x}_t^0)} \left[ \ln \frac{p_{\boldsymbol{\theta}}(\mathbf{x}_t^{k_t:K} \mid \mathbf{z}_{t-1})}{q(\mathbf{x}_t^{k_t+1:K} \mid \mathbf{x}_t^{k_t})} \right] \qquad \text{(Markov property of } q(\cdot)\text{)}$$

$$= C_1(\mathbf{x}_0, k_t) + \left[ \mathop{\mathbb{E}}_{\mathbf{x}_t^{k_t:K} \sim q(\mathbf{x}_t^{k_t:K} | \mathbf{x}_t^0)} \ln \frac{p_{\boldsymbol{\theta}}(\mathbf{x}_t^{k_t:K} \mid \mathbf{z}_{t-1})}{q(\mathbf{x}_t^{k_t+1:K} \mid \mathbf{x}_t^0)} \right],$$

where the constant $C_1(\mathbf{x}_0, k_t) = \mathbb{E}_{\mathbf{x}_t^{k_t:K} \sim q(\mathbf{x}_t^{k_t:K} | \mathbf{x}_t^0)} \left[ \ln \frac{q(\mathbf{x}_t^{k_t+1:K} | \mathbf{x}_t^0)}{q(\mathbf{x}_t^{k_t+1:K} | \mathbf{x}_t^{k_t})} \right]$ depends only on $\mathbf{x}_0$ and $k_t$. To check the conditions for equality, note that if $q(\mathbf{x}_t^{k_t+1:T} \mid \mathbf{x}_t^{k_t}) \equiv p_{\boldsymbol{\theta}}(\mathbf{x}_t^{k_t+1:T} \mid \mathbf{x}_t^{k_t}, \mathbf{z}_{t-1})$, then

$$\mathop{\mathbb{E}}_{\mathbf{x}_t^{k_t+1:K} \sim q(\mathbf{x}_t^{k_t+1:K} | \mathbf{x}_t^{k_t})} \left[ \ln \frac{p_{\boldsymbol{\theta}}(\mathbf{x}_t^{k_t:K} \mid \mathbf{z}_{t-1})}{q(\mathbf{x}_t^{k_t+1:K} \mid \mathbf{x}_t^{k_t})} \right] = \ln p_{\boldsymbol{\theta}}(\mathbf{x}_t^{k_t} \mid \mathbf{z}_{t-1}) + \mathop{\mathbb{E}}_{\mathbf{x}_t^{k_t+1:K} \sim q(\mathbf{x}_t^{k_t+1:K} | \mathbf{x}_t^{k_t})} \left[ \ln p_{\boldsymbol{\theta}}(\mathbf{x}_t^{k_t+1:K} \mid \mathbf{z}_{t-1}, \mathbf{x}_t^{k_t}) \right]$$

Since $\ln(\cdot)$ is strictly concave, $\mathbb{E}_{\mathbf{x}_t^{k_t+1:K} \sim q(\mathbf{x}_t^{k_t+1:K} | \mathbf{x}_t^{k_t})} \left[ \ln p_{\boldsymbol{\theta}}(\mathbf{x}_t^{k_t} \mid \mathbf{z}_{t-1}) \right] = 0$ if and only if $p_{\boldsymbol{\theta}}(\mathbf{x}_t^{k_t+1:K} \mid \mathbf{z}_{t-1}, \mathbf{x}_t^{k_t}) = q(\mathbf{x}_t^{k_t+1:K} \mid \mathbf{x}_t^{k_t})$. $\qquad\square$

**Claim 3** (Computing the expected ELBO).

$$\mathop{\mathbb{E}}_{\mathbf{x}_t^{k_t:K} \sim q(\mathbf{x}_t^{k_t:K} | \mathbf{x}_t^0)} \ln \frac{p_{\boldsymbol{\theta}}(\mathbf{x}_t^{k_t:K} \mid \mathbf{z}_{t-1})}{q(\mathbf{x}_t^{k_t+1:K} \mid \mathbf{x}_t^0)}$$

$$= C_3(\mathbf{x}_0, k_t) + \mathbf{1}\{k_t = 0\} \ln p_{\boldsymbol{\theta}}(\mathbf{x}_t^0 \mid \mathbf{x}_t^1, \mathbf{z}_{t-1}) + \sum_{j=1}^{K-1} \mathbf{1}\{j \geq k_t\} \mathrm{D}_{\mathbb{KL}} \left( q(\mathbf{x}_t^j \mid \mathbf{x}_t^{j+1}, \mathbf{x}_t^0) \,\|\, p_{\boldsymbol{\theta}}(\mathbf{x}_t^j \mid \mathbf{x}_t^{j+1}, \mathbf{z}_{t-1}) \right),$$

*where $C_2(\mathbf{x}_0, k_t)$ is some other constant depending on $\mathbf{x}_0$ and $k_t$.*

*Proof.* The proof invokes similar manipulations to the standard ELBO derivation for diffusion, but with a few careful modifications to handle the fact that we only noise to level $k_t$. As is standard, we require the identity

$$q(\mathbf{x}_t^j \mid \mathbf{x}_t^{j-1}, \mathbf{x}_t^0) = q(\mathbf{x}_t^{j-1} \mid \mathbf{x}_t^j, \mathbf{x}_t^0) \cdot \frac{q(\mathbf{x}_t^j \mid \mathbf{x}_t^0)}{q(\mathbf{x}_t^{j-1} \mid \mathbf{x}_t^0)}. \qquad\qquad \text{(A.10)}$$

**Part 1: Expanding the likelihood ratios** . Using the above identity, we obtain

$$\ln \frac{p_{\boldsymbol{\theta}}(\mathbf{x}_t^{k_t:K} \mid \mathbf{z}_{t-1})}{q(\mathbf{x}_t^{k_t+1:K} \mid \mathbf{x}_t^0)}$$

$$= \ln p(\mathbf{x}_t^K \mid \mathbf{z}_{t-1}) + \ln \frac{p_{\boldsymbol{\theta}}(\mathbf{x}_t^{k_t} \mid \mathbf{x}_t^{k_t+1}, \mathbf{z}_{t-1})}{q(\mathbf{x}_t^{k_t+1} \mid \mathbf{x}_t^0)} + \sum_{j=k_t+2}^{K} \ln \frac{p_{\boldsymbol{\theta}}(\mathbf{x}_t^{j-1} \mid \mathbf{x}_t^j, \mathbf{z}_{t-1})}{q(\mathbf{x}_t^j \mid \mathbf{x}_t^{j-1}, \mathbf{x}_t^0)}$$

$$\overset{(i)}{=} \ln p(\mathbf{x}_t^K \mid \mathbf{z}_{t-1}) + \ln \frac{p_{\boldsymbol{\theta}}(\mathbf{x}_t^{k_t} \mid \mathbf{x}_t^{k_t+1}, \mathbf{z}_{t-1})}{q(\mathbf{x}_t^{k_t+1} \mid \mathbf{x}_t^0)} + \sum_{j=k_t+2}^{K} \left( \ln \frac{p_{\boldsymbol{\theta}}(\mathbf{x}_t^{j-1} \mid \mathbf{x}_t^j, \mathbf{z}_{t-1})}{q(\mathbf{x}_t^{j-1} \mid \mathbf{x}_t^j, \mathbf{x}_t^{k_t})} + \ln \frac{q(\mathbf{x}_t^{j-1} \mid \mathbf{x}_t^0)}{q(\mathbf{x}_t^j \mid \mathbf{x}_t^0)} \right)$$

$$\overset{(ii)}{=} \ln p(\mathbf{x}_t^K \mid \mathbf{z}_{t-1}) + \ln \frac{p_{\boldsymbol{\theta}}(\mathbf{x}_t^{k_t} \mid \mathbf{x}_t^{k_t+1}, \mathbf{z}_{t-1})}{q(\mathbf{x}_t^{k_t+1} \mid \mathbf{x}_t^0)} + \ln \frac{q(\mathbf{x}_t^{k_t+1} \mid \mathbf{x}_t^{k_t})}{q(\mathbf{x}_t^K \mid \mathbf{x}_t^{k_t})} + \sum_{j=k_t+1}^{K-1} \ln \frac{p_{\boldsymbol{\theta}}(\mathbf{x}_t^j \mid \mathbf{x}_t^{j+1}, \mathbf{z}_{t-1})}{q(\mathbf{x}_t^j \mid \mathbf{x}_t^{j+1}, \mathbf{x}_t^0)}$$

$$= \frac{\ln p(\mathbf{x}_t^K \mid \mathbf{z}_{t-1})}{q(\mathbf{x}_t^K \mid \mathbf{x}_t^{k_t})} + \ln p_{\boldsymbol{\theta}}(\mathbf{x}_t^{k_t} \mid \mathbf{x}_t^{k_t+1}, \mathbf{z}_{t-1}) + \sum_{j=k_t+1}^{K-1} \ln \frac{p_{\boldsymbol{\theta}}(\mathbf{x}_t^j \mid \mathbf{x}_t^{j+1}, \mathbf{z}_{t-1})}{q(\mathbf{x}_t^j \mid \mathbf{x}_t^{j+1}, \mathbf{x}_t^0)}$$

$$= \ln\left( q(\mathbf{x}_t^{k_t} \mid \mathbf{x}_t^{k_t+1})^{\mathbf{1}\{k_t \geq 1\}} \right) + \ln \frac{p(\mathbf{x}_t^K \mid \mathbf{z}_{t-1})}{q(\mathbf{x}_t^K \mid \mathbf{x}_t^{k_t})} + \ln \frac{p_{\boldsymbol{\theta}}(\mathbf{x}_t^{k_t} \mid \mathbf{x}_t^{k_t+1}, \mathbf{z}_{t-1})}{q(\mathbf{x}_t^{k_t} \mid \mathbf{x}_t^{k_t+1})^{\mathbf{1}\{k_t \geq 1\}}} + \sum_{j=k_t+1}^{K-1} \ln \frac{p_{\boldsymbol{\theta}}(\mathbf{x}_t^j \mid \mathbf{x}_t^{j+1}, \mathbf{z}_{t-1})}{q(\mathbf{x}_t^j \mid \mathbf{x}_t^{j+1}, \mathbf{x}_t^0)},$$

where $(i)$ uses A.10, $(ii)$ invokes a cancellation in the telescoping sum, and the final display follows from the computation

$$q(\mathbf{x}_t^{k_t} \mid \mathbf{x}_t^{k_t+1})^{\mathbf{1}\{k_t \geq 1\}} = \begin{cases} 1 & k_t = 0 \\ q(\mathbf{x}_t^{k_t} \mid \mathbf{x}_t^{k_t+1}) & k_t \geq 1 \end{cases}. \tag{A.11}$$

Observe that, because we don't parameterize $p(\mathbf{x}_t^K \mid \mathbf{z}_{t-1})$, $\ln\left( q(\mathbf{x}_t^{k_t} \mid \mathbf{x}_t^{k_t+1})^{\mathbf{1}\{k_t \geq 1\}} \right) +$ $\frac{\ln p(\mathbf{x}_t^K | \mathbf{z}_{t-1})}{q(\mathbf{x}_t^K | \mathbf{x}_t^{k_t})}$ can be regarded as some constant $C'(\mathbf{x}_t^{k_t}, \mathbf{x}_t^{k_t+1}, \mathbf{x}_t^K)$. Thus,

$$\ln \frac{p_{\boldsymbol{\theta}}(\mathbf{x}_t^{k_t:K} \mid \mathbf{z}_{t-1})}{q(\mathbf{x}_t^{k_t+1:K} \mid \mathbf{x}_t^0)} = C'(\mathbf{x}_t^{k_t}, \mathbf{x}_t^{k_t+1}, \mathbf{x}_t^K) + \ln \frac{p_{\boldsymbol{\theta}}(\mathbf{x}_t^{k_t} \mid \mathbf{x}_t^{k_t+1}, \mathbf{z}_{t-1})}{q(\mathbf{x}_t^{k_t} \mid \mathbf{x}_t^{k_t+1})^{\mathbf{1}\{k_t \geq 1\}}} + \sum_{j=k_t+1}^{K-1} \ln \frac{p_{\boldsymbol{\theta}}(\mathbf{x}_t^j \mid \mathbf{x}_t^{j+1}, \mathbf{z}_{t-1})}{q(\mathbf{x}_t^j \mid \mathbf{x}_t^{j+1}, \mathbf{x}_t^0)}$$
$$\tag{A.12}$$

**Part 2: Taking expecations.** We can now simplify to taking expectations. Observe that

$$\mathbb{E}_{\mathbf{x}_t^{k_t:K} \sim q(\mathbf{x}_t^{k_t:K} | \mathbf{x}_t^0)} \ln \frac{p_{\boldsymbol{\theta}}(\mathbf{x}_t^j \mid \mathbf{x}_t^{j+1}, \mathbf{z}_{t-1})}{q(\mathbf{x}_t^j \mid \mathbf{x}_t^{j+1}, \mathbf{x}_t^0)} = D_{\mathbb{KL}}\left( q(\mathbf{x}_t^j \mid \mathbf{x}_t^{j+1}, \mathbf{x}_t^0) \,\|\, p_{\boldsymbol{\theta}}(\mathbf{x}_t^j \mid \mathbf{x}_t^{j+1}, \mathbf{z}_{t-1}) \right),$$

and similarly,

$$\mathbb{E}_{\mathbf{x}_t^{k_t:K} \sim q(\mathbf{x}_t^{k_t:K} | \mathbf{x}_t^0)} \ln \frac{p_{\boldsymbol{\theta}}(\mathbf{x}_t^{k_t} \mid \mathbf{x}_t^{k_t+1}, \mathbf{z}_{t-1})}{q(\mathbf{x}_t^{k_t} \mid \mathbf{x}_t^{k_t+1})^{\mathbf{1}\{k_t \geq 1\}}} = \begin{cases} \ln p_{\boldsymbol{\theta}}(\mathbf{x}_t^0 \mid \mathbf{x}_t^1, \mathbf{z}_{t-1}) & k_t = 0 \\ D_{\mathbb{KL}}\left( q(\mathbf{x}_t^{k_t} \mid \mathbf{x}_t^{k_t+1}, \mathbf{x}_t^0) \,\|\, p_{\boldsymbol{\theta}}(\mathbf{x}_t^{k_t} \mid \mathbf{x}_t^{j+1}, \mathbf{z}_{t-1}) \right) & k_t \geq 1. \end{cases}$$

Finally, $\mathbb{E}_{\mathbf{x}_t^{k_t:K} \sim q(\mathbf{x}_t^{k_t:K} | \mathbf{x}_t^0)} C'(\mathbf{x}_t^{k_t}, \mathbf{x}_t^{k_t+1}, \mathbf{x}_t^K)$ is a constant $C_2(k_t, \mathbf{x}_0)$ depending only on $k_t, \mathbf{x}_0$. Thus, from (A.12)

$$\mathbb{E}_{\mathbf{x}_t^{k_t:K} \sim q(\mathbf{x}_t^{k_t:K} | \mathbf{x}_t^0)} \ln \frac{p_{\boldsymbol{\theta}}(\mathbf{x}_t^{k_t:K} \mid \mathbf{z}_{t-1})}{q(\mathbf{x}_t^{k_t+1:K} \mid \mathbf{x}_t^0)}$$

$$= C_2(k_t, \mathbf{x}_0) + \mathbf{1}\{k_t = 0\} \ln p_{\boldsymbol{\theta}}(\mathbf{x}_t^0 \mid \mathbf{x}_t^1, \mathbf{z}_{t-1}) + \sum_{j=\max\{1, k_t\}}^{K-1} D_{\mathbb{KL}}\left( q(\mathbf{x}_t^j \mid \mathbf{x}_t^{j+1}, \mathbf{x}_t^0) \,\|\, p_{\boldsymbol{\theta}}(\mathbf{x}_t^j \mid \mathbf{x}_t^{j+1}, \mathbf{z}_{t-1}) \right)$$

$$= C_2(k_t, \mathbf{x}_0) + \mathbf{1}\{k_t = 0\} \ln p_{\boldsymbol{\theta}}(\mathbf{x}_t^0 \mid \mathbf{x}_t^1, \mathbf{z}_{t-1}) + \sum_{j=1}^{K-1} \mathbf{1}\{j \geq k_t\} D_{\mathbb{KL}}\left( q(\mathbf{x}_t^j \mid \mathbf{x}_t^{j+1}, \mathbf{x}_t^0) \,\|\, p_{\boldsymbol{\theta}}(\mathbf{x}_t^j \mid \mathbf{x}_t^{j+1}, \mathbf{z}_{t-1}) \right).$$

$$\square$$

**Completing the proof of the ELBO.** We are now ready to complete the proof. By combining the previous two claims, we have

$$\underset{\mathbf{x}_t^{k_t} \sim q(\mathbf{x}_t^{k_t}|\mathbf{x}_t^0)}{\mathbb{E}} \left[ \ln p_{\boldsymbol\theta}(\mathbf{x}_t^{k_t} \mid \mathbf{z}_{t-1}) \right]$$

$$\geq C_3(\mathbf{x}_0, k_t) + \mathbf{1}\{k_t = 0\} \ln p_{\boldsymbol\theta}(\mathbf{x}_t^0 \mid \mathbf{x}_t^1, \mathbf{z}_{t-1}) + \sum_{j=1}^{K-1} \mathbf{1}\{j \geq k_t\} D_{\mathbb{KL}}\left( q(\mathbf{x}_t^j \mid \mathbf{x}_t^{j+1}, \mathbf{x}_t^0) \,\|\, p_{\boldsymbol\theta}(\mathbf{x}_t^j \mid \mathbf{x}_t^{j+1}, \mathbf{z}_{t-1}) \right),$$

where $C_3(\mathbf{x}_0, k_t) = C_1(\mathbf{x}_0, k_t) + C_2(\mathbf{x}_0, k_t)$ and where again, the above is an equality when $q(\mathbf{x}_t^{k_t+1:T} \mid \mathbf{x}_t^{k_t}) \equiv p_{\boldsymbol\theta}(\mathbf{x}_t^{k_t+1:T} \mid \mathbf{x}_t^{k_t}, \mathbf{z}_{t-1})$. Taking an expectation over $k_t \overset{\text{unif}}{\sim} \{0, 1, \ldots, K\}$, we have

$$\underset{k_t \overset{\text{unif}}{\sim} \{0,1,\ldots,K\}}{\mathbb{E}} [\mathbf{1}\{k_t = 0\}] = \frac{1}{K+1}, \qquad \underset{k_t \overset{\text{unif}}{\sim} \{0,1,\ldots,K\}}{\mathbb{E}} \mathbf{1}\{j \geq k_t\} = \frac{j+1}{K+1}. \qquad (\text{A.13})$$

and consequently,

$$\underset{k_t \overset{\text{unif}}{\sim} \{0,1,\ldots,K\}}{\mathbb{E}} \underset{\mathbf{x}_t^{k_t} \sim q(\mathbf{x}_t^{k_t}|\mathbf{x}_t^0),1\leq t\leq T}{\mathbb{E}} \ln p_{\boldsymbol\theta}((\mathbf{x}_t^{k_t})_{1\leq t\leq T})$$

$$\geq C_4(\mathbf{x}_t^0) + \frac{1}{K+1} \ln p_{\boldsymbol\theta}(\mathbf{x}_t^0 \mid \mathbf{x}_t^1, \mathbf{z}_{t-1}) + \sum_{j=1}^{K-1} \frac{j+1}{K+1} D_{\mathbb{KL}}\left( q(\mathbf{x}_t^j \mid \mathbf{x}_t^{j+1}, \mathbf{x}_t^0) \,\|\, p_{\boldsymbol\theta}(\mathbf{x}_t^j \mid \mathbf{x}_t^{j+1}, \mathbf{z}_{t-1}) \right)$$

Invoking Claim 1,

$$\underset{\text{forward}}{\mathbb{E}} [\ln p_{\boldsymbol\theta}((\mathbf{x}_t^{k_t})_{1\leq t\leq T})]$$

$$\geq \sum_{t=1}^{T} \underset{<t}{\mathbb{E}} \underset{k_t \overset{\text{unif}}{\sim} \{0,1,\ldots,K\}}{\mathbb{E}} \underset{\mathbf{x}_t^{k_t} \sim q(\mathbf{x}_t^{k_t}|\mathbf{x}_t^0)}{\mathbb{E}} \left[ \ln p_{\boldsymbol\theta}(\mathbf{x}_t^{k_t} \mid \mathbf{z}_{t-1}) \right]$$

$$= \sum_{t=1}^{T} \underset{<t}{\mathbb{E}} \left[ C_4(\mathbf{x}_t^0) + \frac{1}{K+1} \ln p_{\boldsymbol\theta}(\mathbf{x}_t^0 \mid \mathbf{x}_t^1, \mathbf{z}_{t-1}) + \sum_{j=1}^{K-1} \frac{j+1}{K+1} D_{\mathbb{KL}}\left( q(\mathbf{x}_t^j \mid \mathbf{x}_t^{j+1}, \mathbf{x}_t^0) \,\|\, p_{\boldsymbol\theta}(\mathbf{x}_t^j \mid \mathbf{x}_t^{j+1}, \mathbf{z}_{t-1}) \right) \right]$$

We conclude by observing that $\sum_{t=1}^{T} \mathbb{E}_{<t} \left[ C_4(\mathbf{x}_t^0) \right]$ is a constant $C(\mathbf{x}_{1:T}^0)$, and that

$$\underset{<t}{\mathbb{E}} \left[ \ln p_{\boldsymbol\theta}(\mathbf{x}_t^0 \mid \mathbf{x}_t^1, \mathbf{z}_{t-1}) \right] = \underset{\text{forward } p, \mathbf{z}_{1:T}}{\mathbb{E}} \left[ \ln p_{\boldsymbol\theta}(\mathbf{x}_t^0 \mid \mathbf{x}_t^1, \mathbf{z}_{t-1}) \right]$$

$$\underset{<t}{\mathbb{E}} \left[ D_{\mathbb{KL}}\left( q(\mathbf{x}_t^j \mid \mathbf{x}_t^{j+1}, \mathbf{x}_t^0) \,\|\, p_{\boldsymbol\theta}(\mathbf{x}_t^j \mid \mathbf{x}_t^{j+1}, \mathbf{z}_{t-1}) \right) \right]$$

$$= \underset{\text{forward } p, \mathbf{z}_{1:T}}{\mathbb{E}} \left[ D_{\mathbb{KL}}\left( q(\mathbf{x}_t^j \mid \mathbf{x}_t^{j+1}, \mathbf{x}_t^0) \,\|\, p_{\boldsymbol\theta}(\mathbf{x}_t^j \mid \mathbf{x}_t^{j+1}, \mathbf{z}_{t-1}) \right) \right],$$

since both terms only depend on $k_{1:t-1}, (\mathbf{x}_s^{k_s})_{1\leq s\leq t-1}$ and $\mathbf{z}_{1:t-1}$. We conclude then that

$$\underset{\text{forward}}{\mathbb{E}} [\ln p_{\boldsymbol\theta}((\mathbf{x}_t^{k_t})_{1\leq t\leq T})] \geq C(\mathbf{x}_{1:T}^0)$$

$$+ \underset{\text{forward } p, \mathbf{z}_{1:T}}{\mathbb{E}} \left[ \sum_{t=1}^{T} \left( \frac{1}{K+1} \ln p_{\boldsymbol\theta}(\mathbf{x}_t^0 \mid \mathbf{x}_t^1, \mathbf{z}_{t-1}) + \sum_{j=1}^{K-1} \frac{j+1}{K+1} D_{\mathbb{KL}}\left( q(\mathbf{x}_t^j \mid \mathbf{x}_t^{j+1}, \mathbf{x}_t^0) \,\|\, p_{\boldsymbol\theta}(\mathbf{x}_t^j \mid \mathbf{x}_t^{j+1}, \mathbf{z}_{t-1}) \right) \right) \right],$$

as needed. Lastly, we recall that the above is an *equality* under the conditions that
(a) $p_{\boldsymbol\theta}(\mathbf{z}_t \mid \mathbf{z}_{t-1}, \mathbf{x}_t^{k_t})$ is a Dirac distribution, and (b) $q(\mathbf{x}_t^{k_t+1:T} \mid \mathbf{x}_t^{k_t}) \equiv p_{\boldsymbol\theta}(\mathbf{x}_t^{k_t+1:T} \mid \mathbf{x}_t^{k_t}, \mathbf{z}_{t-1})$, and we reindex $j \leftarrow j+1$ to ensure consistency with indexing in standard expositions of the diffusion ELBO.

$\square$

# B  Additional Method Details

## B.1  Fused SNR reweighting

SNR reweighting [24] is a widely used technique to accelerate the convergence of image diffusion models. In short, it reweighs the diffusion loss proportional to the signal-to-noise ratio (SNR) of

noisy $\mathbf{x}^k$. In Diffusion Forcing, conditioning variable $\mathbf{z}_{t-1}$ can also contain a non-trivial amount of information about $\mathbf{x}_t$, in addition to $\mathbf{x}_t^{k_t}$. For example, in a deterministic markovian system, if $\mathbf{x}_{t-1}^{k_{t-1}}$ has its noise level $k_{t-1} = 0$, the posterior state $\mathbf{z}_{t-1}$ contains all the information needed to predict $\mathbf{x}_t^0$ regardless of the noise level of $\mathbf{x}_t^{k_t}$.

Therefore we **re-derive** SNR reweighting to reflect this change in Diffusion Forcing. We call this technique Fused SNR reweighting. We follow the intuition of original SNR reweighting to loosely define SNR in a sequence with independent levels of noises at different time steps. Denote $S_t$ as the normalized SNR reweighting factor for $\mathbf{x}_t^{k_t}$ following its normal derivation in diffusion models. For example, if one uses min snr strategy [24], its reweighting factor will always fall between $[0, C]$ which we divide by $C$ to get $S_t \in [0, 1]$. Define signal decay factor $0 < \gamma < 1$, measuring what proportion of signal in $\mathbf{x}_{t-1}^{k_{t-1}}$ contribute to denoising $\mathbf{x}_t^{k_t}$. This is the simple exponential decay model of sequential information. Now, define cumulated SNR recursively as the running mean of $S_t$: $\bar{S}_t = \gamma \bar{S}_{t-1} + (1-\gamma) S_t$ to account for signals contributed by the entire noisy history to the denoising at time step $t$. The other factor that contributes to the denoising is $S_t$ of noisy observation $\mathbf{x}_t^{k_t}$. To combine them, we use a simplified model for independent events. Notice $S_t$ and $\bar{S}_t$ always falls in range $[0, 1]$, and therefore can be reinterpreted as probabilities of having all the signal one needs to perfect denoise $\mathbf{x}_t^{k_t}$. Since the noise level at $t$ is independent of prior noise levels, we can view $S_t$ and $\bar{S}_{t-1}$ as probabilities of independent events and thus can composed to define a joint probability $S_t' = 1 - (1 - S_t)(1 - \bar{S}_{t-1})$, and we use this $S_t'$ as our fused SNR reweighting factor for diffusion training.

In our experiments, we choose to follow the min-SNR reweighting strategy [24] to derive the $S$. Our Fused SNR reweighting proves extremely useful to accelerate the convergence of video prediction, while we didn't observe a boost on non-image domains so we didn't use it there.

## B.2 Architecture

**Video Diffusion** We choose both the raw image $\mathbf{x}$ and latent state $\mathbf{z}$ to be 2D tensors with channel, width, and height. For simplicity, we use the same width and height for $\mathbf{x}$ and $\mathbf{z}$. We then implement the transition model $p(\mathbf{x}_t^{k_t} | \mathbf{z}_{t-1})$ with a typical diffusion U-net [46]. We use the output of the U-net as the input to a gated recurrent unit (GRU) and use $\mathbf{z}_{t-1}$ as the hidden state feed into a GRU. The output of GRU is treated as $\mathbf{z}_t$. For observation model $p(\mathbf{x}_t | \mathbf{z}_t)$, we use a 1-layer resnet [27] followed by a conv layer. We combine these two models to create an RNN layer, where the latent of a particular time step is $\mathbf{z}_{t-1}$, input is $\mathbf{x}_t^{k_t}$ and output is $\hat{\mathbf{x}}$. One can potentially obtain better results by training Diffusion Forcing with a causal transformer architecture. However, since RNN is more efficient for online decision-making, we also stick with it for video prediction and it already gives us satisfying results.

We choose the number of channels in $\mathbf{z}$ to be 16 for DMlab and 32 for Minecraft. In total, our Minecraft model consists of 36 million parameters and our DMlab model consists of 24 million parameters. We can potentially obtain a better Minecraft video prediction model with more parameters, but we defer that to future works to keep the training duration reasonable ($< 1$ day). In maze planning, the number of total parameters is $4.33$ million.

**Non-Video Diffusion** For non-spatial $\mathbf{x}$ that is not video nor images, we use residue MLPs [59] instead of Unet as the backbone for the dynamics model. Residue MLP is basically the ResNet [27] equivalent for MLP. Similar to video prediction, we feed the output of resMLP into a GRU along with $\mathbf{z}_{t-1}$ to get $\mathbf{z}_t$. Another ResMLP serves as the observation model.

## B.3 Diffusion parameterization

In diffusion models, there are three equivalent prediction objectives, $\mathbf{x}_0$, $\epsilon$ [28], and $v$ parameterization [52]. Different objectives lead to different reweighting of loss at different noise levels, together with SNR reweighting. For example, $\epsilon$ parameterization and $v$ parameterization are essential in generating pixel data that favors high-frequency details.

In our experiments, we use $v$ parameterization for video prediction and found it essential to both convergence speed and quality.

We observe that $\mathbf{x}_0$ parameterization is strongly favorable in planning and imitation learning, likely because they don't favor an artificial emphasis on high-frequency details. We observe the benefits of v-parameterization in time-series prediction.

## B.4 Noise schedule

We use sigmoid noise schedule [9] for video prediction, linear noise schedule for maze planning, and cosine schedule for everything else.

## B.5 Implementation Details of Sampling with Guidance

**Corner case of sampling noise** In our sampling algorithm, due to the flexibility of the scheduling matrix $\mathcal{K}$, there are corner cases when $\mathbf{x}_t^{k_t}$ is required to stay at its same noise level during a sampling step. The core question of this corner case is whether we should update $\mathbf{x}_t^{k_t}$ at all. One option is just copying over the old value. The other option is to run a backward diffusion followed by a forward diffusion back to its old noise level to resample under the diffusion process. While we conclude this can be an open question, we prefer the later approach, resampling, and use it in Monte Carlo Guidance to generate multiple samples. We note that even if one takes the first approach, the guidance gradient can still flow back in the time steps before $t$ as the dynamics model $p(\mathbf{z}_t|\mathbf{x}_t^{k_t}, \mathbf{z}_{t-1})$ can still propagate the guidance gradient to $\mathbf{z}_{t-1}$.

Other than Monte Carlo Guidance, this corner case only happens when $k_t = 0$ or $k_t = K$ throughout our experiments. That is, we chose our $\mathcal{K}$ such that once any token gets diffused slightly, it will keep diffusing. In the case of $k_t = K$, keeping $\mathbf{x}_t^{k_t}$ at the same noise level implies it will stay as white noise, and we don't even need to sample another white noise. In case $k_t = 0$, the time step is already completely diffused either approach should give us the same result so we just opt for copying over for simplicity.

**Guidance for maze planning** In maze planning, our main baseline Diffuer [36] discards the reward from the dataset and directly plans with the goal position and velocity. We adopt the same convention for Diffusion Forcing. One can perform guidance on goal position using log-likelihood $||\mathbf{p}_T - \mathbf{g}||$, but a flexible horizon model should not require users to manually specify a $T$ to reach its goal, instead we want it to try to reach the goal for any possible horizon. Therefore we use the reward model $\sum_t ||\mathbf{p}_T - \mathbf{g}||$ so any time step can be the final step to reach the goal. This objective is challenging due to the non-convex nature of 2D maze, but we found Diffusion Forcing can still reliably find plans without bumping into walls. However, we also observe that the agent tend to leave the goal location due to the nature of the provided dataset - the goal location is just one possible waypoint for the robot to pass through, and there are no trajectories that simply stay at the goal. We also tried this reward for guidance with Diffuser, but it didn't work even with a good amount of tuning.

## B.6 Performance Optimization

Accelerating the diffusion sampling of Diffusion Forcing is similar to that of normal diffusion models. We adopt DDIM [57] sampling for the diffusion of each token. While we use $K = 1000$ steps of diffusion, we sample with only $100$ DDIM for video prediction and $50$ for non-video domains.

While Diffusion Forcing can be implemented with transformers, we use an RNN as the backbone for Diffusion Forcing experiments it's widely used in decision-making for its flexibility and efficiency in online decision-making systems. To further reduce training time and GPU memory usage, we use frame-stacking to stack multiple observed images as a single $\mathbf{x}$. This is due to the fact that adjacent tokens can be very similar - e.g. recording the same motion at higher fps can lead to this. We deem that it's wasteful if we roll out the dynamics model multiple times to generate almost identical tokens. For video datasets, we manually examine how many time steps it takes to require a minimal level of prediction power instead of copying frames over. There is another reason why we use frame stacking - many diffusion model techniques such as different noise schedules are designed to model $\mathbf{x}$ with correlated elements or redundancy. Low-dimensional systems may need drastically different hyperparameters when they lack the data redundancy these techniques are tested on. Frame stacking is thus also helpful for our non-image experiments so we can start with canonical hyperparameters of diffusion models. We use a frame stack of $4$ for DMlab video prediction, $8$ for Minecraft, and $10$ for maze planning.

At sampling time, we also have a design choice to reduce compute usage, as reflected in line 8 of Algorithm 2. In line 8, we directly assign $\mathbf{z}_t^{\text{new}}$ to $\mathbf{z}_t$, instead of recalculating $\mathbf{z}_t$ with posterior model $p(\mathbf{z}_t|\mathbf{z}_{t-1}, \mathbf{x}_t^{\text{new}}, k-1)$. Since the model is trained to condition on $\mathbf{z}_t$ estimated from arbitrary noisy history, we recognize that both are valid approaches. The reason why the choose line 8 is twofold. First, it cuts the compute by half, avoiding computing posterior every step. Second, this happens to be what we want for stabilization - $\mathbf{z}_t^{\text{new}}$ already contains the information of the clean $\mathbf{x}_t^{\text{new}}$ under our simplified observation model, and happens to be estimated with $k = k_t$, a noise level higher than that of $\mathbf{x}_t^{\text{new}}$. This happens to implement the behavior we want for stabilization.

## B.7 Sampling schedule for causal uncertainty

Inference is depicted in Algorithm 2 and Figure 2. In Equation B.1, we illustrate a specific instantiation of the $\mathcal{K}$ matrix we used for causal planning. For simplicity, we denote the case where a latent $\mathbf{z}_0$ is given and aim to generate $\mathbf{x}_{1:H+1}$.

$$\mathcal{K}^{\text{pyramid}} = \begin{bmatrix} K & K & K & ... & K \\ K-1 & K & K & ... & K \\ K-2 & K-1 & K & ... & K \\ \vdots & \vdots & \vdots & \ddots & \vdots \\ 1 & 2 & 3 & ... & H \\ 0 & 1 & 2 & ... & H-1 \\ \vdots & \vdots & \vdots & \ddots & \vdots \\ 0 & 0 & 0 & ... & 1 \\ 0 & 0 & 0 & ... & 0 \end{bmatrix} \tag{B.1}$$

Diffusion Forcing begins by sampling our sequences as white noise with noise level $K$. It then denoises along each row $m = 1, \ldots, M$ of $\mathcal{K}$ in decreasing order. It does so by proceeding sequentially through frames $t = 1, \ldots, T$, updating the latent (Line 5 of Algorithm 2), and then partially applying the backward process to noise level $k = \mathcal{K}_{m,t}$ dictated by the scheduling matrix $\mathcal{K}$ (Line 6-7 of Algorithm 2). We call a $\mathcal{K}$ like this pyramid scheduling, as the tokens in the far future are kept at higher noise level than near future.

## B.8 Metrics for Maze Planning

We report the episode reward of Diffusion Forcing for different maze planning environments in Table 1. However, we found that the episode reward isn't necessarily a good metric: Intuitively, maze planning should reward smart agents that can find the fastest route to the goal, not a slow-walking agent that goes there at the end of the episode. The dataset never contains data on the behavior of staying at the goal, so agents are supposed to walk away after reaching the goal with sequence planning methods. Diffuser may had an unfair advantage of just generating slow plans, which happens to let the agent stay in the neighborhood of the goal for more steps and get a very high reward as a result. This metric seems to exploit flaws in the environment design - a good design would involve a penalty of longer time taken to reach the goal. Therefore, in future works based on our paper, we encourage alternative metrics like the time it takes to reach the goal for the first time, which Diffusion Forcing excels at.

## B.9 Implementation Details of Timeseries Regression

We follow the implementation of pytorch-ts, where the validation set is a random subset of the training set with the same number of sequences as the test set. We use early stopping when validation crps-sum hasn't increased for 6 epochs. We leverage the same architecture (1 mlp and 4 grus) as well as a batch size of 32.

## B.10 Compute Resources

All of our experiments use $fp16$ mixed precision training. Time series, maze planning, compositionally, and visual imitation experiments can be trained with a single $2080Ti$ with 11GB of memory. We tune the batch size such that we fully use the memory of GPUs. This translates to a batch size of

2048 for maze planning and compositional experiments, and 32 for visual imitation learning. While we use early stopping on the validation set for time series experiments, we did not carefully search for the minimal number of training steps required, though the model usually converges between 50k to $100k$ steps. The above environments thus usually take $4 - 8$ hours to train although there is without doubt a significant potential for speed up.

Video prediction is GPU intensive. We use $8$ A100 GPUs for both video prediction datasets. We train for $50K$ steps with a batch size of $8 \times 16$. It usually takes 12 hours to converge at $40K$ steps of training (occasional validation time also included).

## C    Additional Intuitions and Explainations

### C.1    Extension to transformer backbone

While this paper focuses on a causal implementation of Diffusion Forcing with RNNs, it's easy to adopt Diffusion Forcing with modern architectures like transformers. One can simply modify a transformer-based sequence diffusion model to train with independent noise levels across tokens and follow the techniques listed in Section B.1. A strict implementation of causal Diffusion Forcing would involve a causal attention mask on the transformer. However, Diffusion Forcing's fractional masking can do something more interesting: Consider the scenario that we use a transformer without a causal mask. We can still implement causality by controlling noise. By labeling the future as full white noise, there is no information leaked into the past tokens. By labeling future tokens as free of noise, we make the model completely non-causal. By labeling the future tokens as noisy, a slight amount of information about the future is provided for the prediction of past tokens. This effectively states that one only needs a non-causal architecture, but controlling fractional noise of the future, to achieve partial or complete causality. These extensions are beyond the scope of this paper, but we already verified their effectiveness and thus provide them as intuitions for future works.

### C.2    The need for independent noise levels

When training Diffusion Forcing, we choose to sample per-token noise level following i.i.d uniform distribution from $[1, 2...K]$. One may wonder about the necessity of this choice. Here we discuss the unique abilities of independent noise and the compute overhead added by it.

The use of independent noise confers a number of special capabilities in our model, including stabilization of autoregressive rollout 3.3, modeling causal uncertainty 3.3, and removing the need for expensive reconstruction guidance when conditioning on context C.6. None of these capabilities can be achieved by full-sequence diffusion. AR-diffusion [66] and Rolling Diffusion [51] can only achieve the first and third one. There are more sampling-time applications such as flexible frame interpolation. Finally, we also saw the practical benefits of using independent noise in hyperparameter tuning. One can simply try different sampling schemes to figure out the most effective one for their applications. All these capabilities only require training the model once with Diffusion Forcing. In contrast, any tuning of the sampling scheme would require re-training the model for AR-diffusion and Rolling Diffusion.

On the other hand, we didn't observe much computing overhead when comparing Diffusion Forcing to full-sequence diffusion, as soon as one closely follows our training techniques like B.1. The empirical evidence is based on our experiments with an experimental transformer implementation of Diffusion Forcing and is thus not fully consistent with the main paper. However, we present the high-level descriptions below for readers interested in more insights: The complexity added by independent noise levels is in the temporal dimension. Therefore, we first adopt a standard technique for video diffusion models - image pre-training, to abstract away the complexity of the image pixels themselves. Then the complexity left is temporal prediction only. We then take the pre-trained image-only model and continue training it on video data. It turns out the sampling result of Diffusion Forcing with fewer training steps in this second stage is already better than that of full-sequence diffusion at convergence. We speculate that the better result is due to the same data-augmentation effect described in prior works [39]. This shows that the overhead added by independent noise is well-warranted when considering the overall training compute (including image pre-training).

## C.3 Guidance as planning

As stated in Section 2, one can use the gradient of the logarithmic of a classifier $\log c(y|\mathbf{x}_t^k)$ to guide the sampling process of diffusion model towards samples with a desired attribute $y$. For example, $y$ can refer to the indicator of a success event. However, we can consider the logarithmic of a more general *energy function* $c(\mathbf{x}_t^k)$. This has the interpretation as $\Pr(y|\mathbf{x}_t^k)$, where $\Pr\left[y = 1 \mid \mathbf{x}_t^k\right] = e^{c(\mathbf{x}_t^k)}$. Some popular candidate energies include

$$c(\mathbf{x}_t^k) = \mathbb{E}\left[\sum_{t'>t} \mathbf{r}'(\mathbf{x}_{t'}^{k_{t'}}) \mid \mathbf{x}_t^k\right], \tag{C.1}$$

corresponding to a cost-to-go; we can obtain unbiased estimates of this gradient by using cumulative reward $\tilde{c}(\mathbf{x}_t^k) = \sum_{t'>t} \mathbf{r}(\mathbf{x}_{t'}^{k_{t'}})$. We can also use goal distance $c = -\|\mathbf{x}_T^{k_T} - \mathbf{g}\|^2$ as a terminal reward. We provide details about the guidance function deployed in the maze2d planning experiment in Appendix B.5.

## C.4 Noising and stabilizing long-horizon generations

Here, we explain in detail how we use noising to stabilize long-horizon generation. At each time $t$, during the denoising, we maintain a latent $\mathbf{z}_{t-1}^{k_{\text{small}}}$ from the previous time step, with $0 < k_{\text{small}} \ll K$ corresponding to some small amount of noise. We then do *next token* diffusion to diffuse the token $\mathbf{x}_t$ across noise levels $\mathbf{x}_t^K, \mathbf{x}_t^{K-1}, \ldots, \mathbf{x}_t^0$ (corresponding to Algorithm 2 with horizon $T = 1$, initial latent $\mathbf{z}_{t-1}^k$, and noise schedule $\mathcal{K}_{m,1} = m$); this process also produces latents $\mathbf{z}_t^K, \mathbf{z}_t^{K-1}, \ldots, \mathbf{z}_t^0$ associated with each noise level. From these, we use the latent $\mathbf{z}_t^{k_{\text{small}}}$ to repeat the process. This noised latent can be interpreted as an implementation of conditioning on $\mathbf{x}_t^{k_{\text{small}}}$ in an autoregressive process. In a potential transformer implementation of Diffusion Forcing as we discussed in Appendix C.1, one can instead run a forward diffusion on a fully diffused token to achieve stabilization.

It is widely appreciated that adding noise to data ameliorates long-term compounding error in behavior cloning applications [38, 42], and even induces robustness to non-sequential adversarial attacks [13]. In autoregressive video generation, the noised $\mathbf{x}_t^{k_{\text{small}}}$ is in-distribution for training, because Diffusion Forcing trains from noisy past observation in its training objective. Hence, this method can be interpreted as a special case of the DART algorithm for behavior cloning [42], where the imitiator (in our case, video generator) is given actions (in our case, next video frames) from noisy observations (in our case, noised previous frames). Somewhat more precisely, because we use both tokens at training time to train Diffusion Forcing, and using slightly noised tokens for autoregression at test time, our approach inherits the theoretical guarantee of the HINT algorithm [5].

## C.5 Why Monte Carlo Guidance relies on Diffusion Forcing

Monte Carlo Guidance provides substantial variance reduction in our estimate of cost-to-go guidance (C.1). This technique crucially relies on the ability to roll out future tokens from current ones to use these sample rollouts to get Monte Carlo estimates for gradients. This is not feasible with full-sequence diffusion, because this requires denoising all tokens in tandem; thus, for a given fixed noise level, there is no obvious source of randomness to use for the Monte Carlo estimate. It may be possible to achieve variable horizon via the trick proposed in the following subsection to simulate future rollouts, but to our knowledge, this approach is nonstandard.

## C.6 Does the replacement technique lead to flexible horizons in full-sequence diffusion?

A naive way to obtain flexible horizon generation in full-sequence diffusion is via the "replacement trick": consider a full sequence model trained to diffuse $\mathbf{x}_{1:T}$, which we partition into $\mathbf{x}_{1:t-1}, \mathbf{x}_{t:T}]$. Having diffused tokens $\mathbf{x}_{1:t-1}$, we can attempt to denoise tokens of the form $[\tilde{\mathbf{x}}_{1:t-1}^k, \mathbf{x}_{t:T}^k]$, where we *fix* $\tilde{\mathbf{x}}_{1:t-1}^k = \mathbf{x}_{1:t-1}$ to be the previously generated token, and only have score gradients update the remaining $\mathbf{x}_{t:T}^k$. One clear disadvantage of this method is inefficiency - one still needs to diffuse the whole sequence even when there is one step left at $t = T - 1$. What's more, [31] points out that this approach of conditioning, named "conditioning by replacement", is both mathematically unprincipled

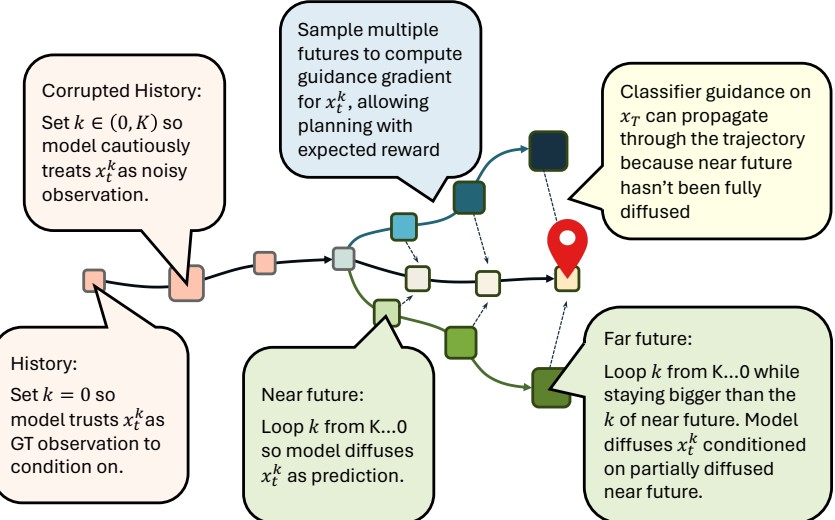

Figure 5: Diffusion Forcing is trained on independent level of noises at different timesteps. As a result, we can control the noise level $k$ to achieve different effects on conditioning and prediction.

and can lead to inconsistency in the generated sequence. The best fix proposed by [31] incorporates an additional gradient term with respect to $\mathbf{x}_{t:T}$ at every diffusion step; this is still an incomplete fix and suffers the computation cost of an extra backward propagation for every sampling step.

### C.7 Further connection to Bayesian filtering

The core idea of Diffusion Forcing can be interpreted as using diffusion to construct an interpolation between prior distribution and posterior distribution of a Bayes filter. Consider the hybrid distribution $p(\mathbf{z}_t|\mathbf{z}_{t-1}, \mathbf{x}_t^k)$. When $k = 0$, this hybrid distribution becomes the posterior $p(\mathbf{z}_t|\mathbf{z}_{t-1}, \mathbf{x}_t)$. On the other hand, when $k = K$, the hybrid distribution becomes $p(\mathbf{z}_t|\mathbf{z}_{t-1}, \mathbf{n})$ for $\mathbf{n} \sim \mathcal{N}(0, \mathbf{I})$. Since the independent Gaussian noise term $\mathbf{n}$ contains no information about $\mathbf{z}$, this is exactly the prior distribution $p(\mathbf{z}_t|\mathbf{z}_{t-1})$. By varying $k$ between $K$ and 0, the same neural network can parameterize everything between prior and posterior.

### C.8 Connection to other sequence training schemes

Noise as masking provides a unified view of different sequence training schemes. The following exposition uses a length 3 sequence as an example: We always start with fully masked sequence $[\mathbf{x}_1^K, \mathbf{x}_2^K, \mathbf{x}_3^K]$ with the goal of denoising it a "clean sequence" of zero noise. $[\mathbf{x}_1^0, \mathbf{x}_2^0, \mathbf{x}_3^0]$. Assume all diffusions are sampled with 3-step DDIM.

**Autoregressive.** In teacher forcing, one trains a model to predict the next token conditioned on prior observations. One can train next-token diffusion models with teacher forcing such as [49]: feed neural network with past observations as well as a current observation and ask it to predict clean current observation. A typical training pair can have the input of $[\mathbf{x}_1^0, \mathbf{x}_2^0, \mathbf{x}_3^K]^\top$ and target of $[\mathbf{x}_1^0, \mathbf{x}_2^0, \mathbf{x}_3^0]^\top$.

At sampling time, one fully diffuses the next token before adding the diffused observation to history to perform an autoregressive rollout. The diffusion process would thus look like

$$[\mathbf{x}_1^K, \mathbf{x}_2^K, \mathbf{x}_3^K]^\top$$
$$[\mathbf{x}_1^{K//2}, \mathbf{x}_2^K, \mathbf{x}_3^K]^\top,$$
$$[\mathbf{x}_1^0, \mathbf{x}_2^K, \mathbf{x}_3^K]^\top,$$
$$[\mathbf{x}_1^0, \mathbf{x}_2^{K//2}, \mathbf{x}_3^K]^\top$$
$$[\mathbf{x}_1^0, \mathbf{x}_2^0, \mathbf{x}_3^K]^\top,$$
$$[\mathbf{x}_1^0, \mathbf{x}_2^0, \mathbf{x}_3^{K//2}]^\top,$$
$$[\mathbf{x}_1^0, \mathbf{x}_2^0, \mathbf{x}_3^0]^\top.$$

Notably, Diffusion Forcing can also perform this sampling scheme at sampling time for applications like imitation learning, when one wants to diffuse the next action as fast as possible.

**Full Sequence Diffusion.** Full sequence diffusion models accept a noisy sequence and denoises level-by-level

$$[\mathbf{x}_1^K, \mathbf{x}_2^K, \mathbf{x}_3^K]^\top$$
$$[\mathbf{x}_1^{K//2}, \mathbf{x}_2^{K//2}, \mathbf{x}_3^{K//2}]^\top,$$
$$[\mathbf{x}_1^0, \mathbf{x}_2^0, \mathbf{x}_3^0]^\top.$$

Notably, Diffusion Forcing can also perform this sampling scheme at sampling time.

**Diffusion Forcing with causal uncertainty** As shown in Figure 2, to model causal uncertainty, Diffusion Forcing keeps the far future more uncertain than the near future by having a larger noise level $k$, at any time of diffusion. An example pattern looks like this:

$$[\mathbf{x}_1^K, \mathbf{x}_2^K, \mathbf{x}_3^K]^\top$$
$$[\mathbf{x}_1^{K//2}, \mathbf{x}_2^K, \mathbf{x}_3^K]^\top,$$
$$[\mathbf{x}_1^0, \mathbf{x}_2^{K//2}, \mathbf{x}_3^K]^\top,$$
$$[\mathbf{x}_1^0, \mathbf{x}_2^0, \mathbf{x}_3^{K//2}]^\top$$
$$[\mathbf{x}_1^0, \mathbf{x}_2^0, \mathbf{x}_3^0]^\top$$

Notable, [65] is the first one to propose such a linear uncertainty sampling scheme for causal diffusion models, although Diffusion Forcing provides a generalization of such scheme in combination with other abilities.

**Diffusion Forcing with stablization** Previously we introduced the autoregressive sampling scheme that Diffusion Forcing can also do. However, such a scheme can accumulate single-step errors because it treats predicted $\mathbf{x}$ as ground truth observation. Diffusion Forcing addresses this problem by telling the model that generated images should be treated as noisy ground truth, as shown in 2.

It first fully diffuses the first token,

$$[\mathbf{x}_1^K, \mathbf{x}_2^K, \mathbf{x}_3^K]^\top$$
$$[\mathbf{x}_1^{K//2}, \mathbf{x}_2^K, \mathbf{x}_3^K]^\top,$$
$$[\mathbf{x}_1^0, \mathbf{x}_2^K, \mathbf{x}_3^K]^\top$$

Then, it feed the diffused $\mathbf{x}_1^0$ into the model but tell it is of a slightly higher noise level, as $\mathbf{x}_1^1$ to diffuse $\mathbf{x}_2$.

$$[\mathbf{x}_1^1, \mathbf{x}_2^{K//2}, \mathbf{x}_3^K]^\top$$
$$[\mathbf{x}_1^1, \mathbf{x}_2^0, \mathbf{x}_3^K]^\top$$

Then, it feeds the diffused $\mathbf{x}_2^0$ into the model but tells it is of a higher noise level, as $\mathbf{x}_2^1$.

$$[\mathbf{x}_1^1, \mathbf{x}_2^1, \mathbf{x}_3^{K//2}]^\top,$$
$$[\mathbf{x}_1^1, \mathbf{x}_2^1, \mathbf{x}_3^0]^\top.$$

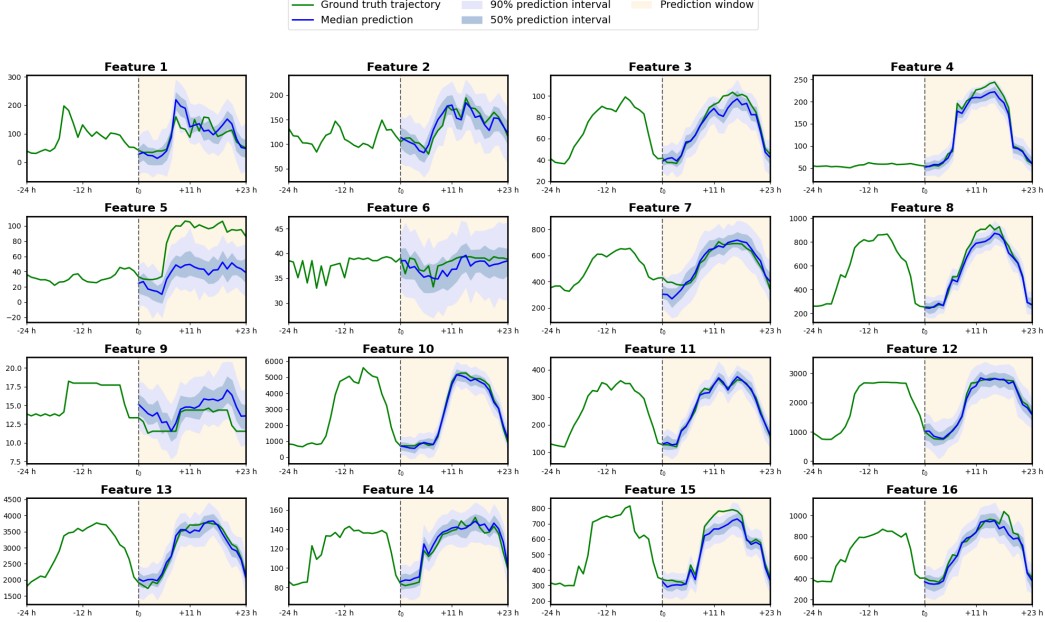

Figure 6: Prediction intervals of Diffusion Forcing for the first prediction window of the test set in the Electricity time series dataset. Only the first 16 features out of 370 are plotted.

## D  Extended Related Work

**Reconstructing masked tokens.** Masked Autoencoders for images [26] and videos [17] are a popular method for representation learning in pixel space. They have been extended to perform diffusion to generate masked patches conditioned on unmasked ones [62, 19].

**Casting Image Generation as Sequence Generation.** [60, 8] show that even generative modeling of non-sequential data, such as images, can be fruitfully cast as sequence generative modeling.

**Non-Diffusion Probabilistic Sequence Models.** [12] parameterize token-to-token transitions via a variational auto-encoder. This makes them probabilistic, but does not directly maximize the joint probability of sequences, but rather, enables sampling from the distribution of single-step transitions.

**Sequence Diffusion with Varying Noise Levels.** Most similar to our work is AR-Diffusion [65] which similarly aims to train next-token prediction models for sequence diffusion. Key differences are that AR-Diffusion proposes a noise level that is *linearly* dependent on the position of each word in the sequence, while our critical contribution is to have each noise level be *independent*, as this uniquely enables our proposed sampling schemes, such as stabilizing auto-regressive generation and conditioning on corrupted observations. Further, AR-Diffusion only explores language modeling and does not explore guidance, while we investigate Diffusion Forcing as a broadly applicable sequence generative model with particular applications to sequential decision-making. In particular, we introduce Monte-Carlo Guidance as a novel guidance mechanism. Another closely related work is Rolling Diffusion [51], which proposes to diffuse a sequence with near future more certain and far future more uncertain, resembling the causally uncertain sampling scheme of Diffusion Forcing. Like AR-Diffussion, Rolling Diffusion's training noise levels are linearly dependent on the positions of tokens and must use the exact same noise level scheme at sampling time. It, therefore, shares the aforementioned limitations of AR-Diffusion as well.

## E  Additional Experiment Results

### E.1  Multivariate Probabilistic Time Series Forecasting

To illustrate Diffusion Forcing's new training objective does not degrade it as a generic sequence model, we evaluate Diffusion Forcing on high-dimensional and long-horizon sequence prediction

tasks in time series prediction. We adopt multiple time series datasets with real-world applications from GluonTS [2] and evaluate Diffusion Forcing with strong baselines with standard metrics in this domain. In this section, we mainly focus on the results and analysis. For a detailed description of datasets and the metric, we refer the reader to Appendix F.4.

**Problem Formulation** Let $\boldsymbol{X} = \{\mathbf{x}_t\}_{t=1}^T$ be a sequence (multivariate time series) of $D$-dimensional observations $\mathbf{x}_t \in \mathbb{R}^D$ of some underlying dynamical process, sampled in discrete time steps $t \in \{1, \ldots, T\}$, where $T \in \mathbb{N}$. In the problem setting of probabilistic time series forecasting, the sequence $\boldsymbol{X} = \{\boldsymbol{X}_c, \boldsymbol{X}_p\}$ is split into two subsequences at time step $t_0 \in \mathbb{N}$ with $1 < t_0 \leq T$: the context window $\boldsymbol{X}_c := \{\mathbf{x}_t\}_{t=1}^{t_0-1}$ (also called history or evidence) of length $t_0 - 1$, and the prediction window $\boldsymbol{X}_p := \{\mathbf{x}_t\}_{t=t_0}^T$ of length $T - t_0 + 1$ (also known as the prediction horizon). Then, the task is to model the conditional joint probability distribution

$$q(\mathbf{x}_{t_0:T} \mid \mathbf{x}_{1:t_0-1}) := \prod_{t=t_0}^T q(\mathbf{x}_t \mid \mathbf{x}_{1:t-1}) \tag{E.1}$$

over the samples in the prediction window. If we know the distribution in (E.1), we can sample forecast prediction sequences given some initial context from the evidence sequence. However, most time-dependent data generation processes in nature have complex dynamics and no tractable formulation of $q(\mathbf{x}_{t_0:T} \mid \mathbf{x}_{1:t_0-1})$. Instead, we construct a statistical model that approximates the generative process in (E.1) and estimates quantiles via Monte Carlo sampling of simulated trajectories. In this way, confidence levels or uncertainty measures can be calculated, and point forecasts can be produced as the mean or median trajectory [35].

Table 2: Results for time series forecasting. We report the test set CRPS$_{\mathbf{sum}}$ (the lower, the better) of comparable methods on six time series datasets. We measure the mean and standard deviation of our method from five runs trained with different seeds.

| Method | Exchange | Solar | Electricity | Traffic | Taxi | Wikipedia |
|---|---|---|---|---|---|---|
| VES [35] | $0.005 \pm 0.000$ | $0.900 \pm 0.003$ | $0.880 \pm 0.004$ | $0.350 \pm 0.002$ | - | - |
| VAR [44] | $0.005 \pm 0.000$ | $0.830 \pm 0.006$ | $0.039 \pm 0.001$ | $0.290 \pm 0.001$ | - | - |
| VAR-Lasso [44] | $0.012 \pm 0.000$ | $0.510 \pm 0.006$ | $0.025 \pm 0.000$ | $0.150 \pm 0.002$ | - | $3.100 \pm 0.004$ |
| GARCH [61] | $0.023 \pm 0.000$ | $0.880 \pm 0.002$ | $0.190 \pm 0.001$ | $0.370 \pm 0.001$ | - | - |
| DeepAR [54] | - | $0.336 \pm 0.014$ | $0.023 \pm 0.001$ | $0.055 \pm 0.003$ | - | $0.127 \pm 0.042$ |
| LSTM-Copula [53] | $0.007 \pm 0.000$ | $0.319 \pm 0.011$ | $0.064 \pm 0.008$ | $0.103 \pm 0.006$ | $0.326 \pm 0.007$ | $0.241 \pm 0.033$ |
| GP-Copula [53] | $0.007 \pm 0.000$ | $0.337 \pm 0.024$ | $0.025 \pm 0.002$ | $0.078 \pm 0.002$ | $0.208 \pm 0.183$ | $0.086 \pm 0.004$ |
| KVAE [40] | $0.014 \pm 0.002$ | $0.340 \pm 0.025$ | $0.051 \pm 0.019$ | $0.100 \pm 0.005$ | - | $0.095 \pm 0.012$ |
| NKF [14] | - | $0.320 \pm 0.020$ | $0.016 \pm 0.001$ | $0.100 \pm 0.002$ | - | $0.071 \pm 0.002$ |
| Transformer-MAF [50] | $0.005 \pm 0.003$ | $0.301 \pm 0.014$ | $0.021 \pm 0.000$ | $0.056 \pm 0.001$ | $0.179 \pm 0.002$ | $0.063 \pm 0.003$ |
| TimeGrad [49] | $0.006 \pm 0.001$ | $0.287 \pm 0.020$ | $0.021 \pm 0.001$ | $0.044 \pm 0.006$ | $0.114 \pm 0.020$ | $0.049 \pm 0.002$ |
| ScoreGrad sub-VP SDE [67] | $0.006 \pm 0.001$ | $\mathbf{0.256 \pm 0.015}$ | $\mathbf{0.019 \pm 0.001}$ | $0.041 \pm 0.004$ | $0.101 \pm 0.004$ | $\mathbf{0.043 \pm 0.002}$ |
| **Ours** | $\mathbf{0.003 \pm 0.001}$ | $0.289 \pm 0.002$ | $0.023 \pm 0.001$ | $\mathbf{0.040 \pm 0.004}$ | $\mathbf{0.075 \pm 0.002}$ | $0.085 \pm 0.007$ |

**Results.** We evaluate the effectiveness of Diffusion Forcing as a sequence model on the canonical task of multivariate time series forecasting by following the experiment setup of [53, 50, 49, 58, 67] Concretely, we benchmark Diffusion Forcing on the datasets Solar, Electricity, Traffic, Taxi, and Wikipedia. These datasets have different dimensionality, domains, and sampling frequencies, and capture seasonal patterns of different lengths. The features of each dataset are detailed in Table 3. We access the datasets from GluonTS [2], and set the context and prediction windows to the same length for each dataset. Additionally, we employ the same covariates as [49]. We evaluate the performance of the model quantitatively by estimating the Summed Continuous Ranked Probability Score CRPS$_{\text{sum}}$ via quantiles. As a metric, CRPS$_{\text{sum}}$ measures how well a forecast distribution matches the ground truth distribution. We provide detailed descriptions of the metric in Appendix F.4. We benchmark with other diffusion-based methods in time series forecastings, such as TimeGrad [49] and the transformer-based Transformer-MAF [50]. In particular, the main baseline of interest, TimeGrad [49], is a next-token diffusion sequence model trained with teacher forcing. We track the CRPS$_{\text{sum}}$ metric on the validation set and use early stopping when the metric has not improved for 6 consecutive epochs, while all epochs are fixed to 100 batches across datasets. We then measure the CRPS$_{\text{sum}}$ on the test set at the end of the training, which we report in Table 2. We use the exact same architecture and hyperparameters for all time series datasets and experiments. Diffusion Forcing outperforms all prior methods except for [67] with which Diffusion Forcing is overall tied, except for the Wikipedia dataset, on which Diffusion Forcing takes fourth place. Note that time series is not the core application

of Diffusion Forcing, and that we merely seek to demonstrate that the Diffusion Forcing objective is applicable to diverse domains with no apparent trade-off in performance over baseline objectives.

## E.2 Additional results in compositional generation

Since Diffusion Forcing models the joint distribution of any subset of a sequence, we can leverage this unique property to achieve compositional behavior - i.e., Diffusion Forcing can sample from the distribution of *subsets* of the trajectory and compose these sub-trajectories into new trajectories.

In particular, we show that we can also have flexible control over how compositional Diffusion Forcing is. As shown in 7, consider a dataset of trajectories on a 2D, square plane, where all trajectories start from one corner and end up in the opposite corner, forming a cross shape. When no compositional behavior is desired, one can let the models replicate the cross-shaped distribution by allowing full memory of the HMM model. When one desires compositional such as generating a V-shaped trajectory, which stitches two sub-trajectories together, one can let the model generate shorter plans with no-memory context using MPC. (Add figures).

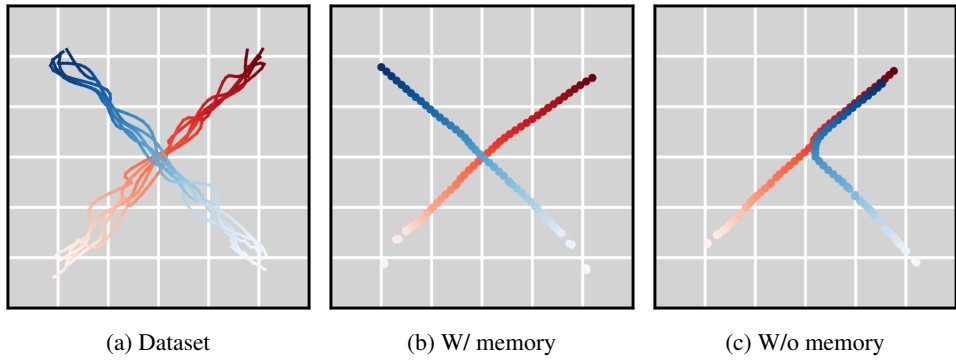

|          (a) Dataset          |          (b) W/ memory          |          (c) W/o memory          |

Figure 7: Given a dataset of trajectories (a), Diffusion Forcing models the joint distribution of all subsequences of arbitrary length. At sampling time, we can sample from the trajectory distribution by sampling Diffusion Forcing with full horizon (b) or recover Markovian dynamics by disregarding previous states (c).

## E.3 Additional results in video prediction (wo/ cherry picking)

**Infinite Rollout without sliding window** Diffusion Forcing can rollout longer than maximum training horizon*without sliding window*. That is, we run Diffusion Forcing's RNN continuously without ever reinitializing $\mathbf{z}_0$. This is a surprising effect we observed from the rollout stabilization property of Diffusion Forcing. In Figure 8, 10, we use Diffusion Forcing to generate video sequences of length 180 and visualize subsampled sequences. Notably, Diffusion Forcing used in these visualizations is trained with a maximum length of 72 frames for Minecraft and 36 frames for DMLab, illustrating it can rollout 2x-5x times longer than it's trained on *without sliding window*. In addition, we also tried rolling these models out for 2000 frames and without seeing the model blowing up on both datasets. There are occasional cases where the Minecraft agent gets stuck and the entire screen is the "dirt" block, but this is more of a dataset issue E.5 and the agent is able to recover after it turns around.

**Consistency** We also present additional results where we only generate within our maximum training length. As shown in figure 13 12, Diffusion Forcing can generate consistent videos. Results are not cherry-picked.

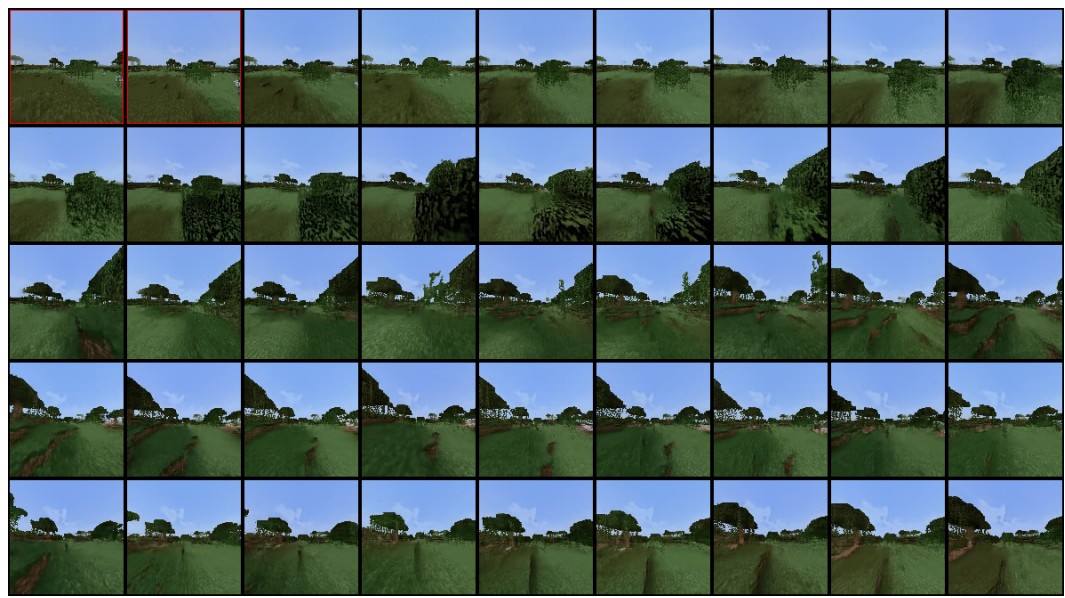

Figure 8: Visualization shows Diffusion Forcing trained on 72 frames is able to rollout 180 frames on Minecraft dataset *without sliding window*. The visualization shows a non-cherry-picked subsampling of these 180 frames, although Diffusion Forcing can roll out much longer (such as 2000 frames) on this dataset.

### E.4  Additional results in planning

We provide some additional visualizations of causal planning in  15. We also present additional visualization of Diffusion Forcing performing model predictive control in action.  As shown in figure 14, Diffusion Forcing can generate plans of shorter horizons since it's flexible horizon.

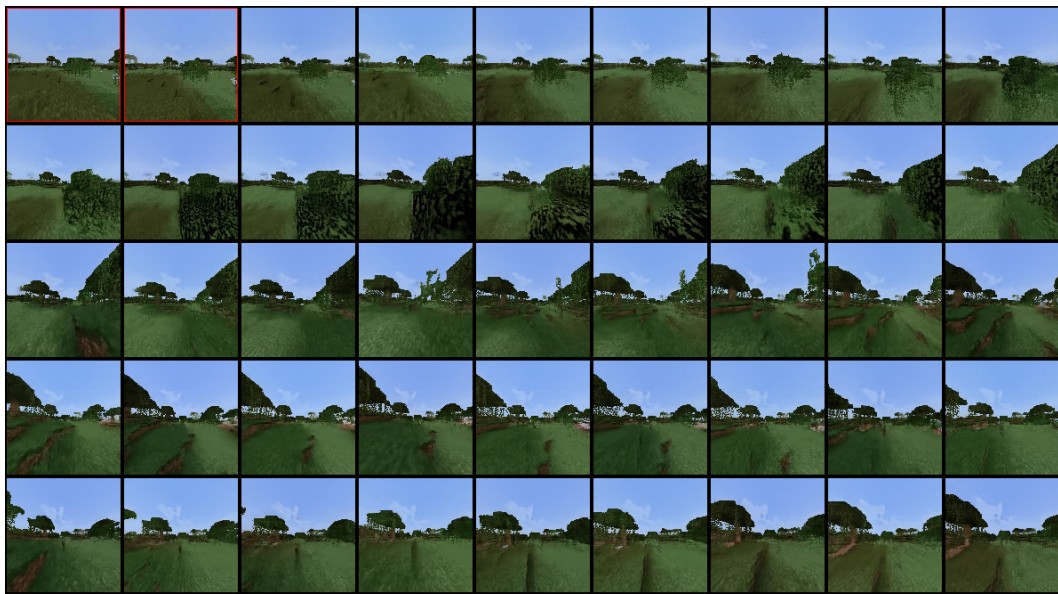

Figure 9: Diffusion Forcing trained on 72 frames is able to rollout 180 frames on Minecraft dataset *without sliding window*. The visualization shows a non-cherry-picked subsampling of these 180 frames, although Diffusion Forcing can roll out much longer (such as 2000 frames) on this dataset. The first few frames marked in red are the ground truth images of the dataset used for conditioning.

## E.5 Real robot experiment setup

In Figure 16 we visualize our robot experiment setup with corruption on observation. The dataset is collected when the target bag isn't present, while we test with such a bag in the scene zero-shot for the imitation learning experiment with observation corruption. The typical failure mode is when the robot no longer reacts to the visual clues of the randomized location of objects. We didn't observe the robot act wildly due to visual distractors.

# F  Additional details about datasets

## F.1  Dataset for video diffusion

We adopt the video prediction dataset Minecraft and DMlab used by TECO[68].

**Minecraft Navigation**  The Minecraft navigation dataset consists of first-person-view videos of random walks in the Minecraft 'swamp' biome. The agent walks via a technique called 'sprint jump' which allows it to jump across blocks without getting stuck at 1 block obstacles. The agent walks straight most of the time, with small chances of turning left or right. The height and width of the video is 128 pixels and we trim long videos to subsequences of 72 frames. The dataset comes with paired action data but we discard them to bring more stochasticity to the prediction task. Due to limited compute, we only train on about 10% of the total subsequences.

One problem we noticed about the dataset is when the agent runs into obstacles with a height of 2 blocks or more. In this case, the agent will get stuck and the entire video sequence will consist of grey granite patterns or brown dirty patterns. This leads to a huge amount of frames with these patterns, making video models predict meaningless frames. Yet, we deem this as a problem of this dataset itself.

**DMLab Navigation**  Deepmind Lab navigation dataset consists of random walks in a 3D maze environment. For DMLab, the resolution is 64 pixels and we use subsequences of 48 frames. We also disregard the provided actions due to training.

We note that the VQ-VAE latent that stable video diffusion [4] diffuses is also only $128 \times 128 \times 3$, indicating Diffusion Forcing has the potential to scale up to higher resolution images with pre-trained

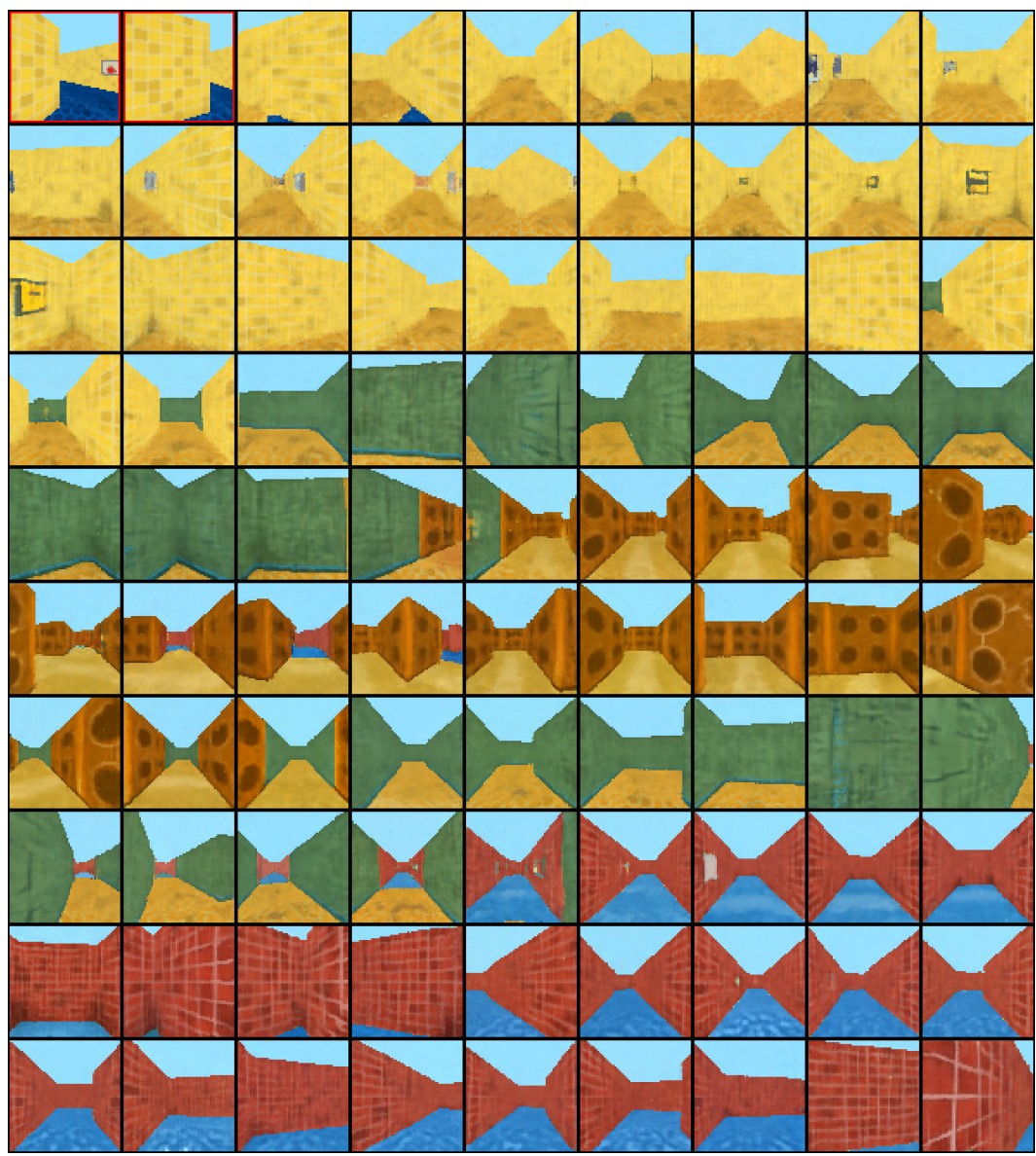

Figure 10: Visualization shows Diffusion Forcing trained on 36 frames is able to rollout 180 frames on DMLab dataset *without sliding window*. The visualization shows a non-cherry-picked subsampling of these 180 frames, although Diffusion Forcing can roll out almost infinitely on this dataset. The first few frames marked in red are the ground truth images of the dataset used for conditioning.

image encoder and decoders. Due to the sheer size of the datasets, we only use about $10\%$ of the total data sequences for training due to limited computing, as we observe that doing so already allows us to make good generations from initial frames from the test set.

## F.2   Dataset for planning

D4RL [18] is a standard offline RL benchmark featuring a wide range of reinforcement learning environments. Each environment is associated with a provided dataset of offline interactions with the environment featuring state, action, and reward trajectories.

Like Diffuer [36], we choose the 3 maze environments as they are challenging long-horizon, multi-modal, sparse reward problems uniquely suited for visualization and evaluating planning algorithms. The IDs for the 3 used environments are "maze2d-medium-v1", "maze2d-large-v1", "maze2d-umaze-

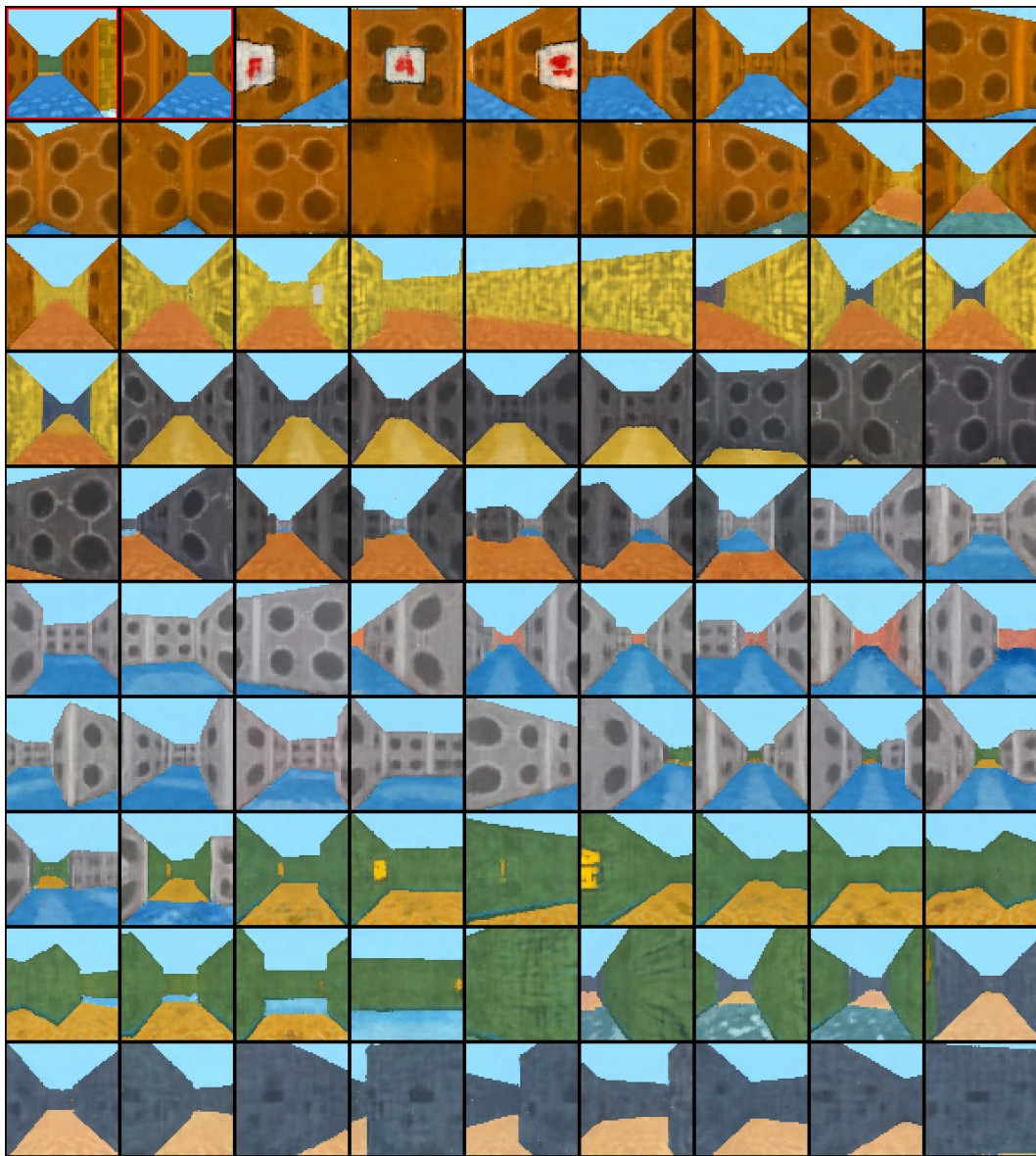

Figure 11: Visualization shows Diffusion Forcing trained on 36 frames is able to rollout 180 frames on DMLab dataset *without sliding window*. The visualization shows a non-cherry-picked subsampling of these 180 frames, although Diffusion Forcing can roll out almost infinitely on this dataset. The first few frames marked in red are the ground truth images of the dataset used for conditioning.

v1". In each environment, one controls the acceleration of a robot to walk it towards a goal. The observation space is 4 dimensional, featuring 2D location and velocity. The action space is 2D acceleration. The agent always receives a random start location and the goal is to reach a fixed goal position for each maze. The agent receives a reward of 1 if it is within a circle of radius 0.5 centered at the goal state, and 0 otherwise.

The offline RL dataset for the maze environments consists of random walks in the maze. Specifically, the authors first designate all intersections and turn in the maze as waypoints and code an agent to navigate between waypoints with some randomization. As a result, the random walks are generated in a way that the path is collision-free with the walls. The random walks introduce stochasticity to the dataset, as trajectories in the dataset are never towards a specific goal.

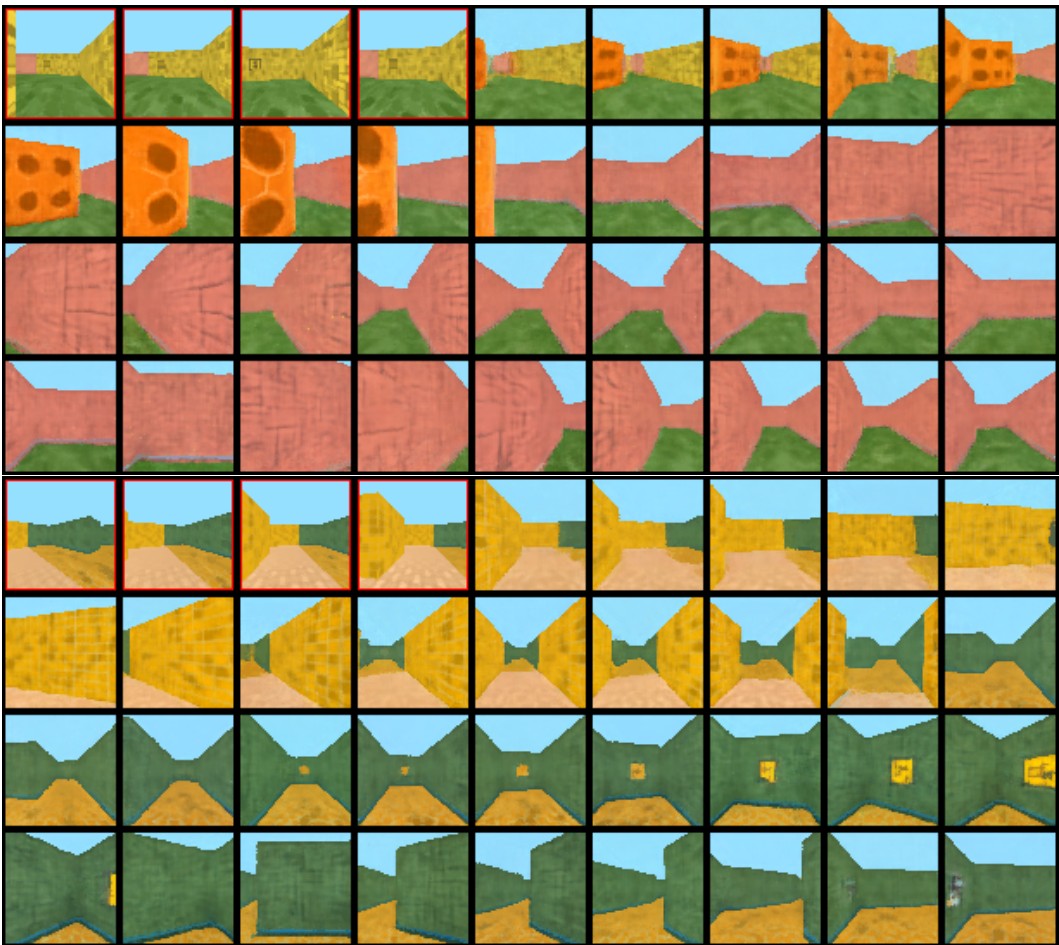

Figure 12: Additional non-cherry-picked video prediction results on DMLab dataset, generated within maximum training length. The first few frames marked in red are the ground truth images of the dataset used for conditioning.

There are a few choices adopted from our main baseline Diffuser [36]: we disregard the reward in the dataset and plan with goals only. We also evaluate a multi-goal variant of each environment (labeled as "multi" in Table 1), where the goal is randomized just like the starting position.

### F.3 Dataset for robot learning

We choose a long horizon robotic manipulation task as described in Section 4.4: Consider a tabletop with three slots where we can place objects. One places an apple at slot A or slot B randomly, and then places an orange at the other slot between A and B. A robot is challenged to swap the position of two fruits using the third slot C. That is, it can only move a fruit to an empty slot at a time. For example, when the apple is at slot A and the orange is at slot B, it may move the apple to slot C, leaving slot A empty. Then move the orange to slot A and finally move the apple from slot C to slot B. In figure 4, we illustrate the non-markovian property of the task: When the apple is at slot B and the orange is at slot C, one cannot tell what the immediate action is without knowing the initial positions of objects.

We put stickers on the table indicating a circular region occupied by any slot. Each circular region is designed to be about double the diameter of a fruit. To make sure the task requires visual feedback, we also randomize the location of a fruit inside the slot. We collected 150 expert demonstrations of a Franka robot performing the task using VR teleoperation and impedance control. Among them, each initial slot configuration makes up half of the dataset. We record videos from two camera views, one from a hand camera and one in the front capturing all three slots. Each demonstration

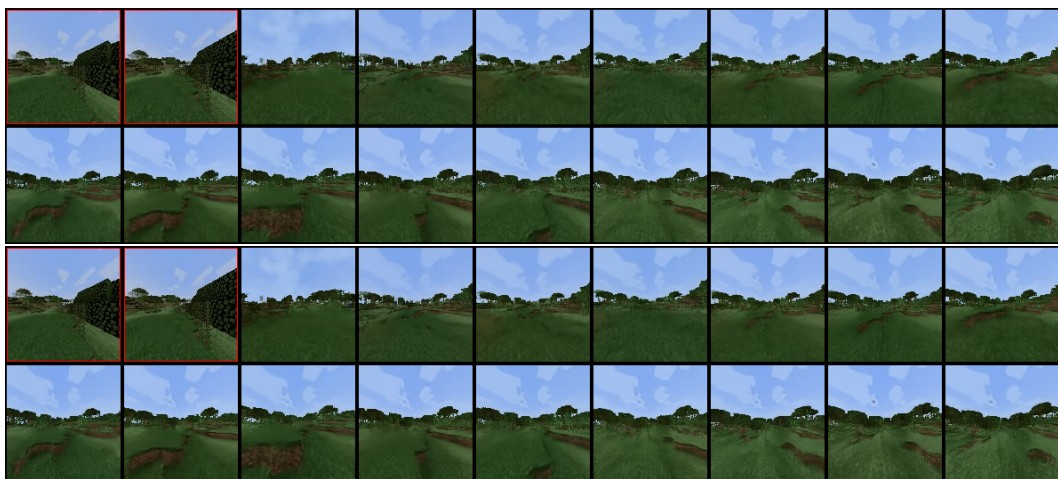

Figure 13: Additional non-cherry-picked video prediction results on the Minecraft dataset, generated within maximum training length. The first few frames marked in red are the ground truth images of the dataset used for conditioning.

also comes with 6 dof actions of the robot hand. During the data collection, since one successful demonstration will swap the position of two objects, its end configuration will naturally serve as the starting configuration of the other randomized location, which we leverage to save time.

Each demonstration comprises $500 - 600$ frames and actions. We train Diffusion Forcing on the entire sequence. However, since adjacent frames are visually close, we pad and downsample the videos to 40 frames where each frame is bundled with 15 actions.

## F.4   Dataset for time series

Table 3: Characteristics of the GluonTS datasets used to benchmark Diffusion Forcing in the domain of time series forecasting.

| Dataset | Dimension | Domain | Frequency | Steps | Prediction length |
|---|---|---|---|---|---|
| Exchange | 8 | $\mathbb{R}^+$ | BUSINESS DAY | 6,071 | 30 |
| Solar | 137 | $\mathbb{R}^+$ | HOUR | 7,009 | 24 |
| Electricity | 370 | $\mathbb{R}^+$ | HOUR | 5,833 | 24 |
| Traffic | 963 | (0,1) | HOUR | 4,001 | 24 |
| Taxi | 1,214 | $\mathbb{N}$ | 30-MIN | 1,488 | 24 |
| Wikipedia | 2,000 | $\mathbb{N}$ | DAY | 792 | 30 |

We use a set of time series datasets accessible via GluonTS [2], which are adopted from prior works like [71, 41, 55]. These datasets capture real-world data of high-dimensional dynamics like monetary exchange rates or the electricity grid. In Table 3, we provide a summary of the features of these datasets, such as the dimensionality, the domains, the sampling frequency, the length of the multivariate sequence in the training set, and the prediction length. We access the datasets in Table 3 via GluonTS and wrap the data processing functions implemented in GluonTS in our own dataloaders. Each dataset consists of one long multivariate sequence, which is the training split, and a set of short sequences that make up the test split. We construct a validation set of the same cardinality as the held-out test set as a randomly sampled subset of subsequences from the training set. All splits are normalized by the mean and the standard deviation of the features in the training split.

**Covariates** Often, statistical models that approximate (E.1) benefit from manually curated features as additional input to the observations. A sequence of covariates $C = \{\mathbf{c}_t\}_{t=1}^{T}$ can be constructed to help the model recognize seasonal patterns and other temporal dependencies. We follow the implementation in [50] to construct the covariate sequence as a function of the frequency of each dataset in Table 3. As such, our covariates are composed of lagged inputs, as well as learned embeddings and handcrafted temporal features that encode information such as the hour of the day or the day of the month, depending on the sampling rate of the particular time series that is being

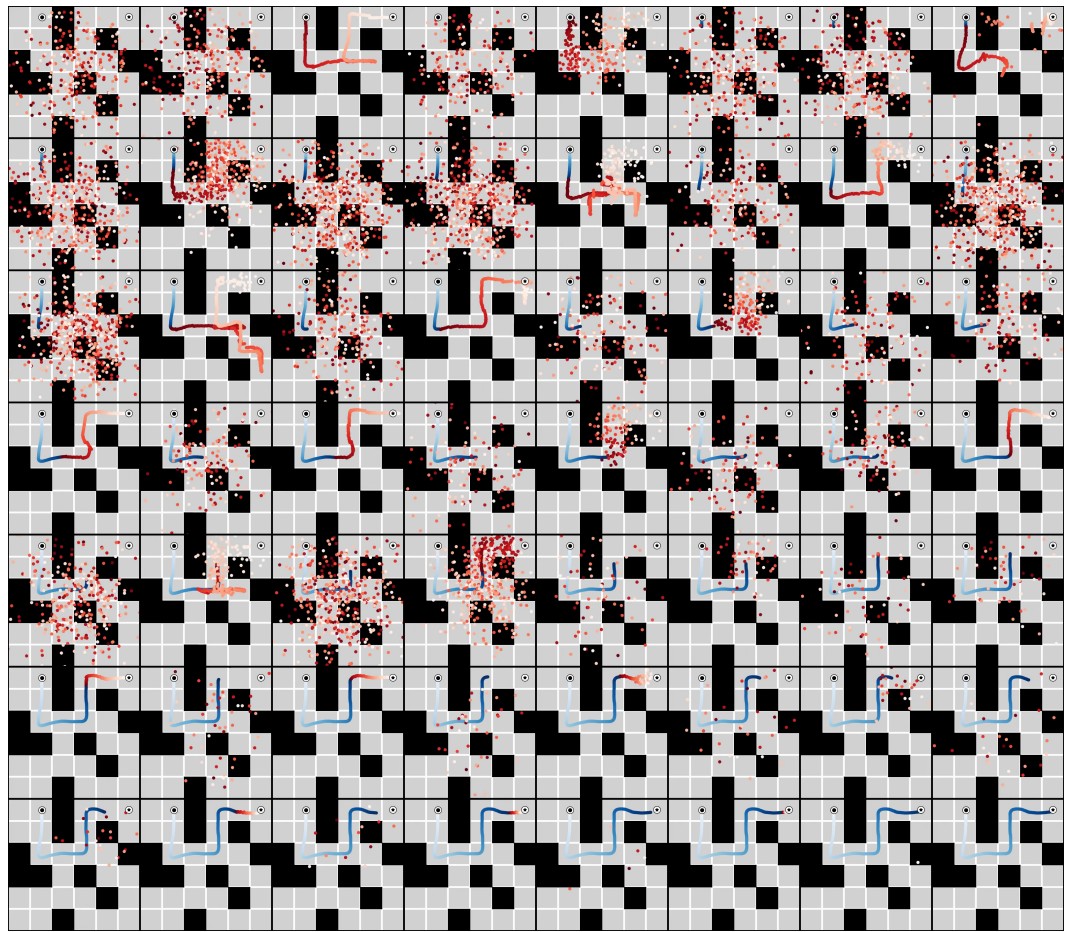

Figure 14: Example MPC planning for maze medium environment. Blue indicated trajectories actually executed already. Red is the plan.

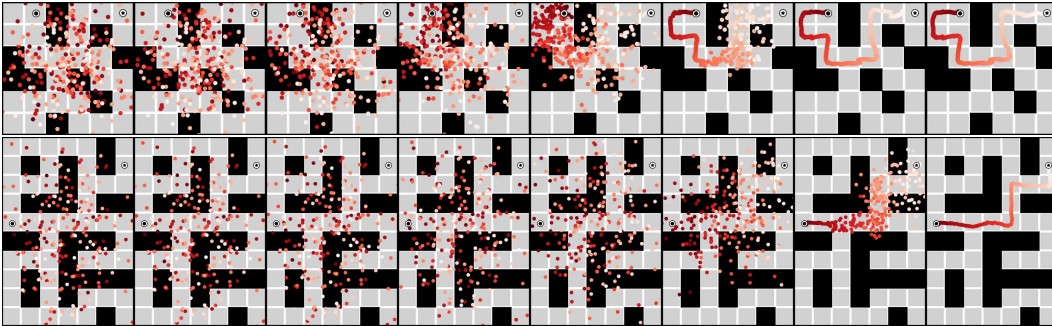

Figure 15: Example plans generated for maze medium (above) and maze large (below) environments.

modeled. Therefore, covariates are known for the entire interval $[1, T]$, even at inference. We can easily incorporate covariates into the probabilistic framework as

$$q(\mathbf{x}_{t_0:T} \mid \mathbf{x}_{1:t_0-1}, \mathbf{c}_{1:T}) := \prod_{t=t_0}^{T} q(\mathbf{x}_t \mid \mathbf{x}_{1:t_0-1}, \mathbf{c}_{1:T}). \qquad \text{(F.1)}$$

The benefit obtained from covariates is highly dependent on the characteristics of both the dataset and the model used, as well as the feature engineering practices followed.

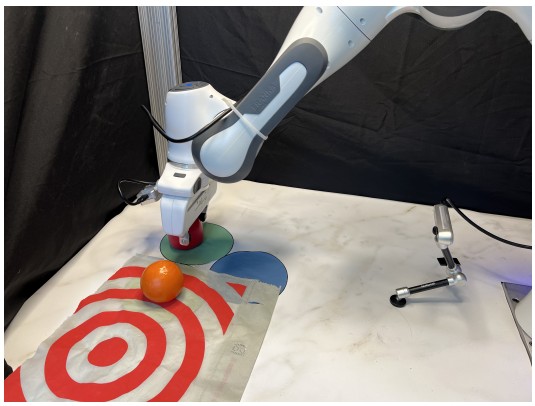

Figure 16: We randomly throw a target bag on the table as a strong visual distractor. Diffusion Forcing can be prompted to treat observation as corrupted rather than ground truth.

**Metric**  The Continuous Ranked Probability Score (CRPS) [45] is a scoring function that measures how well the forecast distribution matches the ground truth distribution:

$$\mathrm{CRPS}(F, x) = \int_{\mathbb{R}} \left(F(z) - \mathbb{I}\{x \le z\}\right)^2 \mathrm{d}z,$$

where $F(z)$ is the univariate cumulative distribution function (CDF) over the predicted value, $x$ is a ground truth observation, and $\mathbb{I}\{x \le z\}$ is the indicator function that is one if $x \le z$ and zero otherwise. By summing the $D$-dimensional time series along the feature dimension for simulated samples (resulting in $\hat{F}_{\mathrm{sum}}(t)$) and ground truth data (as $\sum_i x_{i,t}^0$), we can report the $\mathrm{CRPS}_{\mathrm{sum}}$

$$\mathrm{CRPS}_{\mathrm{sum}} = \mathbb{E}_{t \sim \mathcal{U}(t_0, T)} \left[ \mathrm{CRPS}\left( \hat{F}_{\mathrm{sum}}(t), \sum_i x_{i,t}^0 \right) \right]$$

as the average over the prediction window. The lower the $\mathrm{CRPS}_{\mathrm{sum}}$ value, the better the predicted distribution match the data distribution.

First, we manually sum the time series along the feature dimension and estimate the CDF $\hat{F}_{\mathrm{sum}}(t)$ via 19 quantile levels at each time step $t$ from 100 sampled trajectories. We then use the implementation in GluonTs [2] to compute the CRPS, which we report as $\mathrm{CRPS}_{\mathrm{sum}}$ in Table 2. While we aggregate the data manually, we verify that the numerical error relative to the GluonTS implementation remains orders of magnitude below the precision threshold of the reported metric.

