# OpenReview forum: "Diffusion Forcing: Next-token Prediction Meets Full-Sequence Diffusion"
_NeurIPS.cc/2024/Conference — NeurIPS 2024 poster_

### Official Review · Reviewer_2PA2 · 2024-06-26

**Soundness:** 3
**Presentation:** 3
**Contribution:** 3
**Rating:** 7
**Confidence:** 4

**Summary:**

This work presents Diffusion Forcing, a new framework for probabilistic sequence modeling that combines diffusion models with Bayesian filtering. This framework builds on state of the art approaches to sequence modeling using diffusion models, but has several novel contributions.
First, it allows the model to use *independent* noise levels per element in the sequence, which is a key factor for the stability of autoregressive generation and conditional, guided generation.
Second, this work casts the proposed method for sequential decision making, by defining a new guidance technique that allows the generation of the next toke by guidance on the full distribution of future tokens.
Third, the authors go at great length in demonstrating empirically that their proposed framework is general and can be applied beyond text generation, as opposed to related work.
Diffusion forcing relies on simple ideas: noising is understood as (partial) masking, and it is cast for sequential data, giving rise to a causal variant of diffusion forcing. In practical terms, we have a dynamical system modeled with a simple RNN, in which hidden states follow the Markovian principle: the next hidden state depends on the previous hidden state and a current observation. Previous, next and current are to be intended as indexes in the sequence. Observations are obtained by running a diffusion model with independent noise levels per sequence index, and noisy observations can be used to transition to the next hidden state. A connection with Bayesian filtering is made clear in the paper. Then, we end up with an observation model (for state transitions) and a denoising model (for the diffusion of the observations).
The authors provide a sound development of the training procedure and objective, by showing that their training algorithm optimizes a weighted ELBO on the expected log-likelihood.

**Strengths:**

* This work presents substantial improvement over the literature on the joint application of diffusion models and autoregressive models
* The proposed methodology is technically sound, and well supported by intuition, formal proofs and a wide range of experiments
* The experimental section expands over the literature by focusing on several domains including long-range video generation, planning, compositional generation and multivariate time series forecasting

**Weaknesses:**

* The intuition of the effects of noising on long-horizon generation (appendix B.2) is very similar to the ideas described in a related work AR-Diffusion [62]. This does not highlight the contribution of *independent* noise levels per sequence index
* Experiments do not compare (at least to the best of my understanding) Causal Diffusion Forcing to AR-Diffusion, which would be the natural competitor. Nevertheless, I understand that this would require considerable amount of adaptation work, since AR-Diffusion tackles language modeling mainly
* I liked Appendix B.6, but it is not referenced in the main, and I think this would be more helpful than the figures in sec 3.1

**Questions:**

Q.1: could you please provide a clear summary of why *independent* noise levels are key for your method, and substantiate the difference with respect to AR-Diffusion [62]? I have read Appendix B.2 and Appendix C, where you attempt at clarifying, but I think the benefits for stability, and conditioning on corrupted observations is not spelled out sufficiently

Q.2: is there a way to compare your work to AR-Diffusion that would not require a substantial re-factoring of their code, such that it can be applied to one (e.g. video) use case in your experiments? Another way to go would be to modify your CDF method and use linearly dependent noise levels, to ablate on the requirement for independent noise levels

Minor (no need to answer):
* typos: I could spot one typo in line 186: $[x_1^0, x_2^0, x_3^{K/2}]$
* please check the proofs in the appendix as there are some typos that slipped there, as well as the text in the appendix that has several grammar problems, missing verbs and the like


====== POST REBUTTAL MESSAGE ======

Thank you for the rebuttal. I have raised my score.

**Limitations:**

Yes

---

> ### Author Rebuttal · Authors · 2024-08-07
>
> > Independent noise vs AR-Diffusion
>
> We thank the reviewer for highlighting the need for a more explicit discussion relative to AR Diffusion.
>
> We would first like to clarify that the stabilization discussion in Appendix B.2 is orthogonal to AR-diffusion. The key to AR-Diffusion is training and sampling **future** tokens with linearly growing noise similar to the pyramid sampling of Diffusion Forcing. On the other hand, stabilization is a technique of assigning non-zero noise levels to **context** tokens (history) instead of future tokens. In fact, our stabilization experiment uses autoregressive diffusion, not pyramid noise like AR-Diffusion.
>
> Furthermore, AR-diffusion style pyramid sampling is just one of the possible sampling schemes supported by diffusion forcing **after training once**. In online decision-making like our robot experiment, we can use autoregressive diffusion to achieve the best speed, use pyramid sampling to achieve better guidance, or many other schemes. In **Figure 1 of rebuttal pdf**, we further show that non-causal diffusion forcing can do frame interpolation by assigning noise level 0 to keyframes at arbitrary time steps, while AR-diffusion’s non-causal variant, Rolling Diffusion lacks this flexibility.
>
> Finally, we found that Diffusion Forcing beats AR-Diffusion even using AR-Diffusion’s own sampling scheme as we show and analyze in our response to the next question. This shows that independent noise, just like standard techniques such as image pre-training, plays an important role in enhancing the underlying representation.
>
> > Experimental comparison to AR-Diffusion
>
> We are happy to report that we have re-implemented and benchmarked with AR-Diffusion for video generation, as well as a concurrent work named Rolling Diffusion[1] which is basically non-causal AR-diffusion. As shown in **Figure 4 of the rebuttal pdf**, both AR-diffusion and Rolling-diffusion performed worse in FVD metric compared to diffusion forcing. Our main insight is twofold: 1. while both use linearly growing noise levels along the time axis, both seem to be sensitive with respect to the slope of this linear growth. We spent a fair amount of time tuning the two baselines, but since one has to use the same slope during both training and testing, it requires us to re-train many times to find one slope that’s reasonable for our data. 2. We observe that training with linearly growing noise has too much redundancy. The trajectories tend to have a higher signal-to-noise ratio which made the task too easy. We tried the boundary condition in Rolling Diffusion[1] and multiple tunings to mitigate this, but still failed to match Diffusion Forcing. In fact, table 2 of Rolling Diffusion [1] shows that linearly growing noise isn’t necessarily better than normal diffusion in a large video dataset, aligning with our observation. In addition, a fixed slope at training time creates more inconvenience for DDIM sampling as well due to rounding, creating practical pain for any user who wants to tune DDIM steps.
>
>
> > Reference Appendix B.6 in the main paper
>
> Thank you for your suggestion - we will definitely reference Appendix B.6 in the main paper, and attempt to move it to the main paper entirely, space after incorporating all reviewer feedback permitting!
>
> > Typo
>
> Thank you for catching the typo, we have fixed it!
>
> > Grammar and spelling in appendix
>
> We have done a full pass over the appendix and polished grammar and spelling, thanks for pointing this out!

---

> > ### Comment · Reviewer_2PA2 · 2024-08-11
> > **Thank you for the rebuttal**
> >
> > Dear Authors,
> > thank you for the rebuttal, which answered all my questions clearly. Also, thank you for the additional experiments provided, as well as the thorough discussions with the other reviewers.
> >
> > For all these reasons, I will raise my score.

---

> ### Author Response · Authors · 2024-08-13
> **Thank you to the reviewers**
>
> Dear Reviewer, thank you for your insightful feedback! We will carefully revise our paper to incorporate your suggestions and the latest results from our discussion.

---

### Official Review · Reviewer_35x2 · 2024-07-02

**Soundness:** 4
**Presentation:** 4
**Contribution:** 4
**Rating:** 8
**Confidence:** 4

**Summary:**

The authors introduce Diffusion Forcing (DF), a method for diffusion of sequential data where the noise level at each token can be different (“independent”). The authors show that DF provides more flexible steerability properties and more stable rollouts compared to full-sequence diffusion and teacher forcing. Experimentally, these enable stable video prediction along several timesteps, improved performance on planning tasks, and more robust robotic visuomotor control, relative to the respective relevant baselines.

The following is a more detailed summary.

### Overview of the method

Diffusion Forcing (DF) denoises a sequence of noisy tokens $x^k\_1, \cdots, x^k\_T$ at noise level $k$, starting at $k=K$ (maximum noise) and finishing at $k = 0$. A sequence of hidden states $z\_1, \cdots, z\_T$ is also maintained throughout. Importantly, different tokens can be denoised by different amounts at each denoising step.

The architecture has two main components:
an encoder $p\_\theta(z\_t | z\_{t-1}, x^{k\_t}\_t, k\_t)$ mapping the previous hidden state $z\_{t-1}$, the current noisy token $x^{k\_t}\_t$ and the noise level $k$ to the new value of the current hidden state $z\_t$;
a denoiser $\epsilon\_\theta(z\_t, x^{k\_t}\_t, k\_t)$, which is used to denoise $x^{k\_t}\_t$.

At training time, the noise levels $(k\_t)\_{1 \leq t \leq T}$ are sampled independently, and the encoder and denoiser are trained jointly using the usual diffusion loss on the output of $\epsilon\_\theta$.

At inference time, the tokens are initialized with independent Gaussians. They are then denoised by first computing hidden states from left to right (via an RNN, in this case) using $p\_\theta$, and then by updating the values of the tokens using their current values and the hidden states.

The authors provide an ELBO interpretation for their loss function in the appendix.

### Features of Diffusion Forcing

- The authors highlight the following features of DF:
- It supports classifier guidance, like ordinary diffusion;
- It allows for keeping the noise level higher for future tokens. This makes intuitive sense in an auto-regressive setting, where future tokens depend on past tokens.
- It supports a flexible planning horizon, as tokens are denoised sequentially.
- It supports a more flexible form of classifier guidance (or reward guidance): past tokens can be guided by rewards that depend on future tokens, due to DF’s autoregressive architecture.

When doing reward guidance, the authors propose drawing many samples of possible future trajectories, and averaging their rewards, rather than using a single sample as in ordinary classifier guidance. They term this approach Monte Carlo Tree Guidance (MCTG).

### Overview of experimental findings

- The authors evaluate Diffusion Forcing on video prediction, planning and robotics tasks. Their findings can be summarized as:
- In video prediction (datasets: Minecraft gameplay and DMLab), DF provides more stable rollouts than full-sequence diffusion and teacher forcing. In particular, DF’s rollouts do not diverge as the number of tokens increases.
- In planning (environment: Maze2d from D4RL), DF produces more more consistent trajectories, and executing the generated actions indeed produces a trajectory similar to that given by the generated states. - In addition, DF with MCTG significantly outperforms Diffuser on Maze2d environments.
- In robotics, DF is robust to missing or noisy observations and can perform imitation learning with memory (as it maintains a hidden state, rather than directly mapping observations to actions).
In the appendix, the authors provide additional experiments on compositionality and time series prediction.

**Strengths:**

1. The authors propose an original and performant method combining strengths of diffusion (steerability, robustness to noise, high-quality gradual sample generation) and auto-regressive sequence modelling (flexible horizons, temporal causality, memory in the case of RNNs).

1. In addition, the authors provide a theoretical justification of their loss function in terms of an evidence lower bound (ELBO).

1. The paper is written clearly, providing a clear motivation for the authors’ approach, contextualizing DF relative to existing work (especially Diffuser, AR-Diffusion and Diffusion Policy), and highlighting the main contributions of the method conceptually and experimentally.

1. Trajectory inconsistency is a major limitation of Diffuser, which I have contended with in my own research. Mitigating this limitation is an important enabler of bringing the strengths of diffusion to bear in sequential decision making.

1. Monte Carlo Tree Guidance can be seen as maximizing an empirical estimate of the expected future reward. From a policy optimization perspective, this seems more principled than doing gradient ascent on the realized cumulative reward of a given trajectory, as is done in full-sequence diffusion (e.g. Diffuser). As the authors explain in Appendix B.3, this technique relies on the architecture of DF to be effective.

1. The results on video prediction, available in an anonymized project website provided in the abstract, are particularly impressive in terms of stability and 3D consistency. This, together with results on planning and robotics, indicates DF might contribute to advances in diffusion world models; a research area of established relevance that has received significant attention recently.

**Weaknesses:**

1. Clarification on classifier guidance term $\nabla\_x \log c (x^{\textrm{new}}\_{1:H})$: If this term is to be understood as the gradient of $x \mapsto \log c(x)$ evaluated at $x^{\textrm{new}}\_{1:H}$, then the gradients of $c$ on future tokens would not flow to previous tokens, as the inputs $x$ are “frozen” before being fed into $\log c$. It seems that what the authors mean to say is that future tokens are treated as a differentiable function of  past tokens when computing the gradients. It would strengthen the exposition if the authors either clarify this point in the paper, or update the notation to avoid confusion, as the current notation might lead the reader to believe that the gradients from future tokens do not flow into past ones.

1. The naming of Monte Carlo Tree Guidance seems to misleadingly suggest a similarity with Monte Carlo Tree Search (MCTS). However, the method consists of sampling several future trajectories independently and averaging their guidance gradients, which seems quite divorced from MCTS, which involves actual search on a tree of states and actions and backpropagation of rewards through this tree. As such, I believe naming the technique Monte Carlo Guidance would be more appropriate.

1. High-dimensional control evaluation: Janner et al. (2022) evaluate Diffuser on high-dimensional control locomotion tasks from D4RL. It would be interesting to see an evaluation of Diffusion Forcing in this setting, in particular regarding the consistency between states and actions. I recall from my own experience that executing longer plans from Diffuser in these locomotion environments in an open-loop fashion (i.e. no re-planning) led to trajectories diverging from the generated states, as noted by the authors. It would be interesting to see whether this is addressed by Diffusion Forcing on these higher-dimensional environments.

1. The compositional generation environment referenced in Section 4.3 is very similar (if not identical) to the one used by Janner et al. (2022) in Figure 1b of their paper. I believe it is likely worth mentioning this in Section 4.3.

1. Minor formatting problems

    1. Line 186: $x^{K/2\_3}$ -> $x^{K/2}\_3$

    1. Table 1 caption: “Diffusion Forcingkeeps” -> “Diffusion Forcing keeps”; “Diffusion Forcingachieves” -> “Diffusion Forcing achieves”

    1. Line 495: “in full abstraction” -> “in full generality”
    Line 503: “likelihood for likelihood of all” -> “likelihood for all”

    1. Equation A.3: superscript $k\_2$ on the LHS should be $k\_s$
    Line 516: revise the bracketing of the expression involving $p\_\theta$.

    1. Line 522: “under uniform levels” -> “under uniformly sampled levels”

    1. Line 524: “in the sequel” -> “in the following section”
    Equation A.5: $s \leq T$ -> $1 \leq s \leq T$

    1. Line 592: specify range for $s$ on the first expectation

    1. Line 598: correct superscripts $t\_k$ to $k\_t$

    1. Line 608: revise bracketing of the numerator inside the $\ln$

    1. Line 616: In the last and penultimate lines, replace $\frac{\ln p(...)}{q(...)}$ by $\ln \frac{p(...)}{q(...)}$

    1. Line 628: “we” -> “we have”

    1. Line 631: correct superscript of $x\_t$ on the second line

    1. Line 634: expression with $p\_\theta$ broken between lines

    1. Line 635: capitalize Dirac

    1. Equation B.1: include \left[ \right] in the brackets

    1. Line 664: “we are” -> “we use”

**Questions:**

1. Does the flexible planning horizon (line 209) of Diffusion Forcing not derive from the choice of an RNN as the architecture, rather than e.g. a UNet? Would implementing existing methods such as Diffuser (Janner et al. 2022) not allow for a similar property?

1. In the paragraph “Benefit of Modeling Causality”, the authors highlight that states and actions produced by DF are causally related, which does not hold in practice for Diffuser. Do the authors claim this is due to DF explicitly incorporating temporal structure into its architecture? Could it not also be due to the use of an observation model $p\_\theta(x^0\_t|z\_t)$ to predict the noise-free token $x^0\_t$ from a hidden state $z\_t$?

1. At first sight and in its current form, the method seems tailored to the use of an RNN architecture, rather than a Transformer. For example, the denoiser is applied token-wise, with the information from previous tokens affecting the current token only via the hidden states $z\_t$. How would the method have to be adapted, if at all, to work with transformers, in case one wants to scale up Diffusion Forcing?

1. Janner et al. (2022) showcase in Section 5.4 show how to apply Diffuser with a variable planning budget, and study how the resulting performance varies with the planning budget. Can Diffusion Forcing also be run with a variable planning budget, through warm-starting (as for Diffuser) or otherwise? If so, it would strengthen the paper if the authors described how, and included a similar budget vs. performance analysis, especially in planning and robotics tasks.

**Limitations:**

The authors address relevant limitations in Section 5 and social impacts in the checklist.

---

> ### Author Rebuttal · Authors · 2024-08-07
>
> We thank the reviewer for their in-depth review - we are particularly happy to be able to address some of the limitations of Diffuser that the reviewer had to contend with themselves in the past!
>
> > Clarification on Classifier Guidance Term
>
> Sorry about the confusion. The reviewer is exactly correct in that future tokens are a differentiable function of past tokens via the posterior model, and that this enables us to perform guidance. We adopted the notation for simplicity and to avoid clutter/accommodate space restrictions, but we will absolutely clarify the meaning of that gradient in the text of the “Guidance” section in our subsequent revision. On the other hand, in our transformer implementation of diffusion forcing, any token can attend to the entire history, allowing us to reproduce the guided planning results as well.
>
> > The naming of Monte Carlo Tree Guidance is misleading
>
> Thank you for your comment. Amusingly, we had an internal discussion among the authors on exactly this topic - we report that one of the parties feels highly vindicated by your comment. We will follow your advice and rename the method into Monte Carlo Guidance!
>
> > Consistency of action and state in high-dim control environments
>
> We tried CDF on one high-dimensional control environment (hopper-medium-expert) without guidance to exclude confounding factors. Unfortunately, the observation still diverges from the ground truth eventually without feedback as no model is perfect even in deterministic environments. However, it’s clearly better than Diffuser just like our visualizations in the 2d maze experiment. In the generic RL setting, the environment can be stochastic, and therefore open-loop execution of a plan is expected to diverge from the plan even with an oracle model. Therefore, we believe a thorough evaluation of dynamics better falls in the result in Section 4.5, which shows CDF is competitive for high-dimensional time-series data.
>
> > Similarity of compositional generation environment with Janner et al. (2022).
>
> Thank you for highlighting this relationship - we will absolutely add a reference to Janner et al. (2022) to Section 4.3!
>
> > Minor formatting problems
>
> We apologize for these mistakes and have fixed all of them for the camera-ready!
>
> > Does the flexible horizon property derive from RNN
>
> Flexible horizon planning doesn’t depend on RNN. We’ve reimplemented diffusion forcing with both transformer and 3D-Unet (with temporal attention), showing flexible horizon planning is fully reproducible with attention instead of RNN. For causal attention, this is obvious. For non-causal variants, diffusion forcing allows us to mark padded future entries as pure noise to achieve a flexible horizon. The only ability that depends on RNN is infinite rolling **without** sliding window because state space models have more invariance, although transformer can roll out longer **with** sliding window.
>
> > Could combining Diffuser with RNNs also lead flexible planning horizon?
>
> Diffuser’s maze planning result is strictly fixed horizon. While Diffuser made an argument about flexible horizon via 1D convolution, its maze planning results critically depend on the replacement technique we mentioned in Appendix B.4. That is, Diffuser fixed the last token to be at the goal throughout the diffusion process to generate a fixed length plan of a carefully chosen length. If the objective is to reach the goal asap, this objective would fail, as one doesn’t know which token shall be replaced as the goal. We implement this on Diffuser and cannot reproduce their numbers. We suspect that the mere reason why Diffuser got high scores is that their fixed length objective coincidentally encourages very slow plans that made the agent stay near the goal for longer, an out-of-distribution behavior that doesn’t exist in the dataset.
>
> > Source of causality in DF vs. Diffuser: architecture or observation model
>
> It’s most likely the architecture itself. Our transformer implementation is direct empirical evidence that a separate observation model is the key. It also follows theoretically: consider an alternative formulation of $x_t=[o_t, a_t, r_t]$ like diffuser instead of $x_t=[o_{t+1}, a_t, r_t]$ used in our paper. Even though the hidden state and observation model are untouched, causality is mathematically broken - at time step t, one wants to diffuse action $a_t$ given current $o_t$, but the process of diffusing $a_t$ samples another $o_t$, which might be different from the existing observation. Such weird inconsistency suggests the temporal structure is critical.
>
> > scale up Diffusion Forcing for transformers
>
> Between the initial draft and rebuttal, we’ve already reimplemented diffusion forcing with architectures like Transformers or 3D-Unet and reproduced results. We also find Diffusion Forcing to work well in latent diffusion settings for higher-resolution videos (**Figure 1 of rebuttal pdf**).
>
>
> > Can Diffusion Forcing, like the Diffuser, be run with a variable planning budget, through warm-starting or otherwise?
>
> Unlike Diffuser[1], diffusion forcing implements DDIM sampling which already achieves good speed without Diffuser’s warm-starting technique. Therefore, we provide an alternative planning budget analysis here in terms of frequency of replanning. In **Figure 2 of rebuttal pdf**, we show the performance when we replan 1x, 0.5x, 0.25x, 0.125x of episode length and for 50 steps. We found that diffusion forcing’s performance worsens as we replan less frequently. We’d like to clarify that this is partially due to the dataset itself - the maze dataset never contains the behavior of staying at the goal once reached, and the agent always walks away from the goal even after reaching it. Therefore, diffusion forcing is supposed to follow this suboptimal behavior if there is less replanning.
>
> [1] Michael Janner, Planning with Diffusion for Flexible Behavior Synthesis, 2022

---

> > ### Comment · Reviewer_35x2 · 2024-08-12
> > **Thank you to the authors**
> >
> > Thanks to the authors for taking my suggestions into consideration. I consider the achievements of this paper to be impressive and highly relevant for the line of work at the intersection of control and generative modeling. Hence, I maintain my position of strongly recommending acceptance.

---

> > > ### Author Response · Authors · 2024-08-13
> > > **Thank you to the reviewers**
> > >
> > > Dear Reviewer, thank you for your insightful feedback! We will carefully revise our paper to incorporate your suggestions and the latest results from our discussion.

---

### Official Review · Reviewer_tz82 · 2024-07-11

**Soundness:** 3
**Presentation:** 3
**Contribution:** 3
**Rating:** 7
**Confidence:** 4

**Summary:**

This paper proposes to augment autoregressive models with diffusion. Specifically, rather than generating every token in one shot (one neural network evaluation), the paper proposes to gradually denoise the tokens following an autoregressive order. That is, every token is given a different noise level (lower for former tokens and higher for latter ones), and the tokens are jointly denoised to generate better samples. Compared to pure autoregressive prediction, diffusion forcing allows the model to refine the samples through the diffusion process. Compared to diffusion models, the proposed model is capable of variable-length generation and extrapolation.

The authors also demonstrate additional potential generation tasks that can be done by diffusion-forcing models such as guided autoregressive sampling.

Empirical results demonstrate that diffusion forcing performs well on video prediction and various planning tasks.

**Strengths:**

This paper proposes an interesting combination of autoregressive models and diffusion models and demonstrates that the combination of both outperforms both individual models in terms of performance. Further, the diffusion-forcing paradigm offers many more applications that are otherwise impossible. For example, while doing variable-length generation, the model can leverage classifier-based/-free conditions. This provides much better flexibility to inference-demanding tasks such as planning and control.

The authors propose a training objective of diffusion forcing models based on noise prediction. The objective is proved to be a reweighted version of the evidence lower bound and thus is sound.

Diffusion forcing achieves much better performance compared to autoregressive models and diffusion models in long-horizon generation tasks.

**Weaknesses:**

A more detailed discussion of the noise schedule is desired to better understand the effectiveness of diffusion forcing. Is it necessary to use different noise schedules in different tasks to achieve good performance? Further, can we train the model with various/arbitrary noise schedules and at evaluation time find a good schedule? If any of these is possible it will greatly reduce the training complexity and extend diffusion forcing to more applications.

Theorem 3.1 states that the proposed objective is equivalent to a reweighting of the evidence lower bound. However, it is unclear how the noise schedule biases the reweighting since a very badly balanced ELBO can render the training process unstable.

How diffusion forcing balances efficiency and performance. In the extreme case where only one denoising step per token is allowed, diffusion forcing reduces to autoregressive generation. How much performance gain can we expect if we allow for more computation time?

**Questions:**

How easy or difficult can diffusion forcing be applied to non-autoregressive generation? Although diffusion forcing improves the performance in the autoregressive generation regime, some tasks (e.g., constrained text generation) require awareness of future tokens to generate the current ones. I wonder if diffusion forcing can be extended to this regime.

**Limitations:**

As discussed by the authors, one of the main limitations is that diffusion forcing is only tested with RNN base models but not other autoregressive models such as autoregressive transformers.

---

> ### Author Rebuttal · Authors · 2024-08-07
>
> Thank you for your helpful feedback. We respond to your comments below:
>
> > Can we train with arbitrary noise schedules and find a good schedule at evaluation time?
>
> Yes - at the core of Diffusion Forcing lies exactly the idea that by training with **arbitrary, random** noise schedules, we can experiment with arbitrary noise schedules at evaluation time to find the optimal schedule for each task! In practice, we found this to be very useful as users can try different noise schemes at sampling time for different applications after training once while the slightest tuning would require retraining for various baselines. The proof of the ELBO extends to arbitrary noise schedules as well.
>
> > Is it necessary to use different noise schedules in different tasks?
>
> In short, yes. For example, we found that pyramid sampling with causal diffusion forcing (CDF) is best for MCTG planning while autoregressive diffusion offers the best consistency for causal video. When conditioning on ground truth context frames, CDF works best without stabilization; In contrast, when context is previously generated, CDF works best with stabilization as shown in **Figure 3 of rebuttal pdf**. Furthermore, even if we commit to one specific schedule such as pyramid sampling, it is critical to tune the rate with which uncertainty grows with time via the noise schedule under pyramid sampling. DF allows us to do this easily after training once, while AR-Diffusion [2] (suggested by 35x2 and 2PA2) makes this difficult as this rate is a training hyperparameter and fixed at test time.
>
> We will use the extra page from the camera ready to include this discussion of noise schedules!
>
> > Reduce the training complexity
>
> Surprisingly, we found that random, independent noise schedules at training time do not significantly add to the training complexity. Due to limited space, please refer to the experiment in general response’s “Improved Model Performance” section which shows the overhead of independent noise is insignificant when considering the overall training compute.
>
> We speculate that random, independent noise levels at training time can be seen as a form of data augmentation[3], leading to a bigger “bang for the buck” of training data. Further, they might in expectation improve the quality of gradient flow backward in time.
>
> > Balance between performance and computation time
>
> The suggested sampling scheme reduces to auto-regressive diffusion with ddim_steps of 1, which mainly captures the low-frequency part of the data only. If we have more compute budget, we can sample auto-regressively with >1 ddim_steps and get much higher quality samples. As shown in **Figure 2 of rebuttal pdf**, the main video metric fvd decreases as the sampling time increases. A similar trend follows for maze planning.
>
> > reweighted ELBO can be poorly balanced
>
>
> Indeed, the formal version of Theorem 3.1 in the Appendix, Theorem A.1 demonstrates the weighting explicitly: that the ELBO on the expected log-likehoods, over the randomness of the noise schedule, corresponds to an objective which reweights the contribution of terms from denoising steps k by their step number k. Thus, the reweighting factor is by at most a factor of the total number of denoising steps, K. Note that this weighting by k is independent of the noise schedule (that is, of the variance of the noise at each step k), and depends only on the position k of the noising step (as well as our decision to select noise levels uniformly and independently). We will add this as a remark in the revision.
>
> Though we could choose to up weight gradients from earlier steps by more (as Thm A.1 makes the weights explicit), not reweighting the gradients according to the ELBO corresponds to effectively weighting gradients at smaller denoising steps k more, which may be desirable as these steps capture the “higher resolution” features of the model.
> Lastly, as noted in the remarks following Corollary A.2, our ELBO derivation is sufficiently tight that, given a sufficiently representative neural network, the optima of all (non-trivial) reweightings of our ELBO are also optima of the desired expected likelihood. Hence, the weighting does not meaningfully bias the training objective. We do acknowledge that the weighting may be salient for the dynamics of optimization, but empirically, we find that our decision not to compensate for the step-wise weighting of the ELBO leads to reliable training. We will clarify the remarks following Corollary A.2 to emphasize this point further.
>
> > Main result is RNN
>
> We’ve reimplemented diffusion forcing with both transformer and 3D-Unet architectures after the draft submission. Please see the general response for details due to limited space.
>
> > Apply diffusion forcing to non-autoregressive generation
>
> In the maze planning experiments, we already show how guidance can be used to achieve certain future outcomes. In fact, reconstruction guidance discussed in Appendix B.4 has been widely used to achieve consistency with context through guidance.  Alternatively, one simply uses a non-causal neural network architecture, like our implementation of 3D-Unet, and then - as discussed in the paper - varies the per-token noise level at training time. One can further mask out padded entries by marking them as full noise. We are actively pursuing this direction in follow-up work: Initial results in **Figure 1 of rebuttal pdf** showcase frame interpolation result of Diffusion Forcing without the expensive reconstruction guidance. However, a detailed discussion of this setting would exceed the capacity of the present paper, though we will discuss this direction in the discussion section.
>
> [1] Michael Janner, Planning with Diffusion for Flexible Behavior Synthesis, 2022
>
> [2] Tong Wu et al. “Ar-diffusion: Auto-regressive diffusion model for text generation”, Neurips 2023
>
> [3] Diederik P. Kingma, “Understanding Diffusion Objectives as the ELBO with Simple Data Augmentation”, Neurips 2023

---

> > ### Comment · Reviewer_tz82 · 2024-08-12
> >
> > I thank the reviewer for their detailed response and for adding more experiments and ablations, which strengthen the paper. Therefore, I will maintain my positive rating.

---

> > > ### Author Response · Authors · 2024-08-13
> > > **Thank you to the reviewers**
> > >
> > > Dear Reviewer, thank you for your insightful feedback! We will carefully revise our paper to incorporate your suggestions and the latest results from our discussion.

---

### Official Review · Reviewer_a6Fo · 2024-07-11

**Soundness:** 3
**Presentation:** 2
**Contribution:** 3
**Rating:** 6
**Confidence:** 4

**Summary:**

This paper introduces Diffusion Forcing, a novel training paradigm for sequential generative modeling using diffusion models. Diffusion Forcing learns from sequential tokens with varying independent noise levels, enabling more flexible sampling strategies and general capabilities such as guidance. The experimental results demonstrate that Diffusion Forcing outperforms existing methods, including full sequence diffusion and teacher forcing, across various tasks.

**Strengths:**

1. The proposed Diffusion Forcing method is general and flexible, making it applicable to various tasks.
2. The paper provides a comprehensive discussion on the capabilities of Diffusion Forcing.
3. The experiments are well-designed and effectively demonstrate the proposed method's effectiveness.

**Weaknesses:**

1. **Writing clarity and organization.**
   The writing style impacts readability, making the paper challenging to follow. It would benefit from a clearer organization. The paper primarily covers three points: (a) the proposed Diffusion Forcing (DF) method with independent noise levels and its theoretical analysis, (b) the capabilities of DF, including flexible sampling strategies, and (c) experimental results on various tasks. However, the current structure does not clearly present these points, particularly the DF method. Separating the design of DF and the intuitive explanation from the Bayesian filtering perspective, and listing the resulting capabilities in a separate section, would enhance clarity.

2. **Clarity of figures.**
   The figures are not well-explained and are difficult to understand without referring to the text. For instance, Figure 1 omits latent states in the sampling process for both Diffusion Forcing and Teacher Forcing, which is confusing.

3. **Minor issues and typos.**
   - Line 97: missing a ")"
   - Line 139: "nevel" should be "level"
   - Line 186: "$x^{K/2_3}$" should be "$x^{K/2}_3$"
   - Line 178, 184, etc.: paragraph titles are inconsistently formatted
   - Line 522: missing a "("

**Questions:**

1. **Consistency between training and sampling algorithms.**
   In Algorithms 1 and 2, there appear to be inconsistencies between the training and sampling algorithms. Can the authors provide an intuitive explanation for these inconsistencies? Specifically:
   - During training, the predicted noise $\hat{\epsilon}\_t$ is calculated using the latent from the previous step $z\_{t-1}$, whereas during sampling, $\hat{\epsilon}_t$ is calculated using the latent from the current step $z_t^{\text{new}}$.
   - Similarly, during training, $\hat{\epsilon}\_t = \epsilon\_\theta(z, x\_t\^{k\_t}, k\_t)$ uses the same noise level $k\_t$ of the noisy observation $x\_t\^{k\_t}$, but during sampling, $x\_t$ has a noise level $\mathcal{K}\_{m+1,t}$ instead of $k = \mathcal{K}_{m,t}$.

2. **Stabilizing auto-regressive generation.**
   The authors propose conditioning on the (latent of) slightly noisy previous tokens with a noise level $0 < k \ll K$ to stabilize the auto-regressive generation. How were the values of $k$ chosen in the experiments? Could the authors provide ablation studies on the impact of using this trick?

**I promise to raise the score once all the weaknesses/questions are solved.**

**Limitations:**

Yes.

---

> ### Author Rebuttal · Authors · 2024-08-07
>
> Thank you for your helpful feedback! We respond to your comments below:
>
> > Writing clarity and style & typos
>
> Sorry about the confusion. It’s true that we opted out of an independent “Method” section to introduce the intuitions first. With the extra 1 page for the camera-ready version, we promise to structure the paragraphs more clearly. In particular, following your advice, we will make (1) one “methods” section that discusses DF in one subsection as well as its intuition obtained from Bayesian Filtering in another subsection, (2) one “Capabilities” section discussing novel capabilities, and (3) one ” Results” section.  Thank you for your time on the list of typos, we will make sure to fix them in camera-ready.
>
> > Figure clarity
>
> Thank you for the constructive advice. We will make sure to caption the figures with more details with the 1 extra page limit of the camera-ready version so readers can easily understand them without referring to the main text. For the sampling scheme in Figure 2 (Assuming the figure index in your comment is a typo), we were trying to illustrate a scheme that illustrates both the transformer and RNN as mentioned in the caption. Therefore, we will likely instead remove latent from the training subfigure to make sure it’s consistent with the sampling figure and add a new figure in the appendix to illustrate RNN variant with latent state. We provide a preview of modification in **Figure 5 of the rebuttal pdf**.
>
> > Consistency between training and sampling algorithms.
>
> Thank you for pointing this out. We will definitely fix the algorithm box carefully: 1. $z_{t-1}$ in line 8 of the training algorithm should be changed to $z_{t}$ 2. In line 7 of the sampling algorithm, the $k$ should indeed be corrected to be $\mathcal{K}_{m+1,t}$. We will fix line 5 to reflect this. 3. Outside the two errors you pointed out, some $\alpha$ are accidentally indexed via $t$ or $k$, not $k_t$.
>
> > Details about stabilization and ablating $k$
>
> In our implementation, we choose $k=20$ for $K=1000$ but this value is largely dependent on the difficulty of the task. In **Figure 3(a)** of the rebuttal pdf, we present the result of the requested ablation of this stabilization value. We choose a slightly harder setting for this ablation, reducing the frame rate by half and using lower numbers of DDIM steps to make compounding error more obvious. One can observe that the FVD metric improves as we gradually increase the stabilization level from 0, reaching optimal around 100 before FVD starts to rise. We found stabilization to be extremely important for datasets with high stochastic like the BAIR robot pushing dataset shown in **Figure 3(b) of the rebuttal pdf**. We cannot achieve stable autoregressive rollout even with very high DDIM steps. Stabilization is a fundamental technique to get this dataset to work.

---

> > ### Comment · Reviewer_a6Fo · 2024-08-12
> >
> > Thank you for your response. I highly appreciate the contribution of your work, including novel frameworks, capabilities and extensive experiments, and I expect the writing could be refined in further versions. I raise my score to 6.

---

> > > ### Author Response · Authors · 2024-08-13
> > > **Thank you to the reviewers**
> > >
> > > Dear Reviewer, thank you for your insightful feedback! We will carefully revise our paper to incorporate your suggestions and the latest results from our discussion.

---

### Author Rebuttal · Authors · 2024-08-07

## General Response
We thank the reviewers for their comments and suggestions. We are pleased that the reviewers find our paper original & interesting (Reviewers 35x2,tz82), general & flexible (Reviewers a6Fo, tz82), that it has great performance (Reviewers a6Fo, tz82,35x2) with substantial improvement over prior methods (Reviewers 2PA2) backed by well-designed experiments (Reviewers a6Fo,2PA2) and theoretical justification (Reviewers tz82,35x2,2PA2).

The outstanding concerns center around comparison to prior methods, scaling up beyond RNNs (Reviewers tz82,35x2), the need for independent noise (Reviewers tz82, 2PA2), and more ablations to help understanding (Reviewers a6Fo, tz82,35x2,2PA2). We present further discussion, supported by additional experimental results, to address these concerns.

> Comparison to new baselines and variants

We reimplemented AR-diffusion[1] and its non-causal variant, Rolling-diffusion[2] with multiple rounds of tuning. As we detailed in Appendix C, AR-Diffusion uses linearly growing noise during training and sampling. Their sampling scheme closely resembles Diffusion Forcing’s pyramid sampling so we also present a result of diffusion forcing but using AR-diffusion’s own pyramid sampling scheme. **In Table 4(a) of the rebuttal pdf**, we find that Diffusion Forcing has a clear advantage over all baselines, causal or non-causal. Furthermore, the fact that Diffusion Forcing can beat new baselines with their own sampling schemes further highlights the importance of independent noise, as we detail below.

>  The need for independent noise

We’ve included additional investigative experiments that demonstrate the importance of independent noise, both for **improved model performance** as well as **numerous model capabilities**. We summarize them here and provide detailed discussions in the individual responses.

### Improved Model Performance
- In **Table 4(a) of rebuttal pdf** We trained with Diffusion Forcing but using AR- and Rolling-diffusion’s sampling schemes for inference: this yields improvements over those baselines, suggesting the benefit lies in the training via independent noise.
- We present experimental evidence that, when adopting a standard technique for video diffusion models - image pre-training - the added complexity of independent noise is well warranted: we pretrain Diffusion Forcing and all baselines with images with no temporal structure, before training on video data. While this pretraining improves the performance of all baselines, Diffusion Forcing still (1) performs the best, and (2) its performance in earlier training iterations (20k) is superior to that of baselines at convergence. This shows that the overhead of independent noise is justified when considering the overall training compute (equivalent to 100k video training steps)
- We see a strong practical benefit of independent noise when tuning hyperparameters such as uncertainty rate in pyramid sample, which is painfully inconvenient in AR-Diffusion since it requires retraining.

### Model Capabilities
- As noted in the main text, the use of independent noise confers a number of additional capabilities in our model, including stabilization of autoregressive rollout (Sec. 4.1), modeling causal uncertainty (Sec. 4.2), and removing the need for expensive reconstruction guidance when conditioning on context  (Appendix B.4). None of these capabilities can be achieved by full-sequence diffusion and AR-diffusion can only achieve the first and third one. To demonstrate yet another capability, we present preliminary results on using Diffusion Forcing for frame interpolation without reconstruction guidance in video prediction **Figure 1 of rebuttal pdf**.

> Further ablations

We’ve added the following ablations.
1. An ablation of stabilization level k for our ability to stabilize autoregressive generation. As shown in **Figure 3(a) of the rebuttal pdf**, we found that video metrics monotonically improve as one goes from no stabilization to a certain level and monotonically worsen as we further increase the value.  We further present a qualitative visualization in **Figure 3(b) of the rebuttal pdf** for a more stochastic dataset and found stabilization is an indispensable technique to prevent blowing up.
2. Multiple ablations of performance vs computing budget across domains. In **Figure 2 of the rebuttal pdf**, we found that diffusion forcing can trade off speed for better quality by varying DDIM sampling steps for video and replan frequency for planning.

> Scaling up beyond RNNs

It is straightforward to adapt Diffusion Forcing to transformers, and no changes to the proposed framework have to be made. One simply uses any architecture - causal or non-causal, and then - as discussed in the paper - varies the per-token noise level at training time.

After the initial draft, we’ve further reimplemented Diffusion Forcing with alternative architectures such as the transformer and 3D-UNet with temporal attention. We found that our approach similarly works. In addition, we’ve integrated diffusion forcing with modern techniques such as latent diffusion, allowing it to scale up to much longer video sequences (300) or those with a much higher resolution (512x512). In **Figure 1 of the rebuttal pdf**, we present non-cherry-picked samples of Diffusion Forcing on RE10K, a hard dataset that’s high-resolution, with latent diffusion and temporal attention. We’ve successfully reproduced maze planning results using transformers as well. These results show positive signs of scaling up diffusion forcing. However, we find that a detailed quantitative evaluation of these new variants would exceed the capacity of the present paper, though we will discuss this direction and interesting implications in the discussion section.



[1] Tong Wu et al. “Ar-diffusion: Auto-regressive diffusion model for text generation”. Neurips 2023.

[2]​​ David Ruhe et al. “Rolling Diffusion Models”. Arxiv 2024

---

### Decision · Program_Chairs · 2024-09-25

**Decision:**

Accept (poster)

**Comment:**

The paper presents a new algorithm, called Diffusion Forcing, that attempts
to achieve the strengths of both purely autoregressive generation and
full-sequence generation.  The algorithm is well motivated by the
limitations of those approaches.  The results of the algorithm are
impressive both in terms of numerical performance on control benchmarks and
in the stability of longer range video prediction.

The authors were able to address some concerns about comparisons and
ablations during the rebuttal period.